# High-resolution mapping of global winter-triticeae crops using a sample-free identification method

Yangyang Fu [1], Xiuzhi Chen[1], Chaoqing Song[1], Xiaojuan Huang[2], Jie Dong[3], Qiongyan Peng[1], Wenping Yuan[1]

[1] International Research Center of Big Data for Sustainable Development Goals; School of Atmospheric Sciences, Sun Yat-sen University, Zhuhai, Guangdong, 519082, China
[2] School of Earth Sciences, Chengdu University of Technology, Chengdu, Sichuan, 610059, China
[3] College of Geomatics & Municipal Engineering, Zhejiang University of Water Resources and Electric Power, Hangzhou, Zhejiang, 310018, China

*Correspondence to*: Wenping Yuan (yuanwp3@mail.sysu.edu.cn)

**Abstract.** Winter-triticeae crops, such as winter wheat, winter barley, winter rye, and triticale, are important in human diets and planted worldwide, and thus accurate spatial distribution information of winter-triticeae crops is crucial for monitoring crop production and food security. However, there is still a lack of global high-resolution maps of winter-triticeae crops because of the reliance of existing crop mapping methods on training samples, which limits their application at the global scale. In this study, we propose a new method based on the Winter-Triticeae Crops Index (WTCI) for global winter-triticeae crops mapping. This is a new sample-free method for identifying winter-triticeae crops based on differences in their normalized difference vegetation index (NDVI) characteristics from the heading to the harvesting stages and those of other types of vegetation. We considered state (or province) or country as an identification unit and employed WTCI to produce the first global 30 m resolution distribution maps of winter-triticeae crops from 2017 to 2022 using Landsat and Sentinel images. Validation using field survey samples and visual interpretation samples from Google Earth images indicated that the method exhibited satisfying performance and stable spatiotemporal transferability, with producer's accuracy, user's accuracy and overall accuracy of 81.12%, 87.85% and 87.7%, respectively. Moreover, compared with the Cropland Data Layer (CDL) and the EuroCrops datasets, the overall accuracy and F1 score in most regions of the United States and Europe were more than 80% and 75%. The identified area of winter-triticeae crops was consistent with the agricultural statistical area in almost all investigated counties or regions, and the correlation coefficient ($R^2$) between the identified area and the statistical area was over 0.6, while the relative mean absolute error (RMAE) was less than 30% in all six years. Overall, this study provides a reliable and automatic identification method for winter-triticeae crops without any training samples. The high-resolution distribution maps of global winter-triticeae crops are expected to support multiple agricultural applications. The distribution maps can be obtained at https://doi.org/10.57760/sciencedb.12361 (Fu et al., 2023a).

## 1 Introduction

Crop mapping can provide detailed location and can be used to analyse spatiotemporal dynamics of crops (Skakun et al., 2017). As one of the important types of grain in the world, the planting area and production of winter-triticeae crops (such as winter wheat, winter barley, winter rye, and triticale) in 2020 accounted for approximately 30% and 41% of global grain area and production, respectively (https://www.fao.org/faostat/en/#data), playing a crucial role in global food production and trade. Closely monitoring the spatial distribution of winter-triticeae crops is therefore beneficial for evaluating yield, optimizing land use, and assessing food security (Fu et al., 2021; Nelson and Burchfield, 2021; Wardlow et al., 2007).

Previous studies have mainly focused on mapping winter-triticeae crops distribution in limited regions rather than at the global scale (Gella et al., 2021; Zhang et al., 2019; Zhang et al., 2021). Few studies have attempted to produce global triticeae crop maps (You et al., 2014), but efforts have been limited to coarse resolutions. For example, Monfreda et al. (2008) combined census statistics with global cropland data (Ramankutty et al., 2008) to generate a global distribution map of crops (including barley, rye, triticale, wheat) for the year 2000, with a spatial resolution of 10 km. A recent study produced circa 2015 annual crop harvested area for 26 crops (including barley and wheat) worldwide at 5-min resolution based on a crop production system (irrigated and rainfed) (Grogan et al., 2022). The coarse spatial resolution of these datasets highly limits their applications (Luo et al., 2022). The WorldCereal project proposed by European Space Agency (ESA) has released a global crop map with a spatial resolution of 10 m for 2021, addressing the limitations of spatial resolution in global-scale crop mapping (Van Tricht., 2023). However, this product is currently only available for one year, which will affect the demand for continuous years. At present, the available long-term and high-spatial resolution distribution maps of winter-triticeae crops are mainly at small or national scales (Dong et al., 2020; Huang et al., 2022; He et al., 2019; Zhang et al., 2019), with the most well-known being the Cropland Data Layer (CDL) product in the United States, which is updated annually and has an accuracy greater than 90% for winter-triticeae crops (Boryan et al., 2011). However, in most countries where winter-triticeae crops are planted widely, such maps are still in short supply. Therefore, it is necessary to produce distribution maps of winter-triticeae crops with high-spatial resolution and continuous years for these countries.

The greatest challenge in global crop mapping is the need for substantial field samples for algorithm training. Several methods have been proposed to address this problem when there are only a few or even no ground samples in the target year. Some studies developed a cross-region classifier transfer method (Macdonald and Hall, 1980; Xu et al., 2020). For example, Ge et al. (2021) combined Landsat images with the CDL production of Arkansas to train a classifier and then assessed the spatial transferability of the classifier in California, USA, and Liaoning, China. Other studies proposed a temporal transfer method to alleviate the limitation of insufficient ground samples, i.e., training a classifier based on historical crop samples and then applying it to a target year (Cai et al., 2018; Konduri et al., 2020; Yaramasu et al., 2020). Such as, a previous study used the NDVI features extracted from 2013 crop samples to establish classification rule, and then transferred this rule to identify the crop types for 2011-2013 (Liu et al., 2016). Nevertheless, the accuracy of these methods is relatively low due to the fact that

the trained classifier focuses on a specific region and year, while neglecting the differences in crop phenology in different regions and across years (Zhang et al., 2019).

This study aims to develop a new sample-free method, i.e., Winter-Triticeae Crops Index (WTCI), to identify global winter-
triticeae crops based on Landsat 7, Landsat 8, Sentinel-1 and Sentinel-2 satellite data. The main goals are to (1) assess the accuracy and spatiotemporal transferability of the new method using field survey samples, visual interpretation samples from high-resolution images on Google Earth, CDL dataset, EuroCrops dataset and agricultural statistical data, (2) produce 30 m spatial resolution distribution maps of winter-triticeae crops in 66 countries worldwide from 2017 to 2022 to fill such product gaps, providing a data basis for yield estimation and crop management.

**2 Data and method**

**2.1 Study area**

The study area covers 66 countries, including 36 European countries, 15 Asian countries, eight African countries, two North American country, four South American countries, and one Oceania country (Fig.1). The harvested area of global triticeae crops (including spring and winter varieties) is 278.87 million ha in 2020 (https://www.fao.org/faostat/en/#data), with winter-
triticeae crops accounting for about 75% (i.e., 209.15 million ha) of the global triticeae crops harvested area (Zhao et al., 2018). According to the statistics of the winter-triticeae crops area provided on official websites of various countries (Table S1), the total harvested area of winter-triticeae crops in our study area in 2020 is 207.45 million ha, occupying 99.19% of the global winter-triticeae crops harvested area. The study area features an intricate interweaving of plains and mountains, resulting in a complex and varied agricultural landscape and different tillage systems. In addition, the study area has a diverse climate
dominated by temperate and subtropical conditions. Winter-triticeae crops are usually sown in the autumn of the previous year and harvested in the summer of the following year.

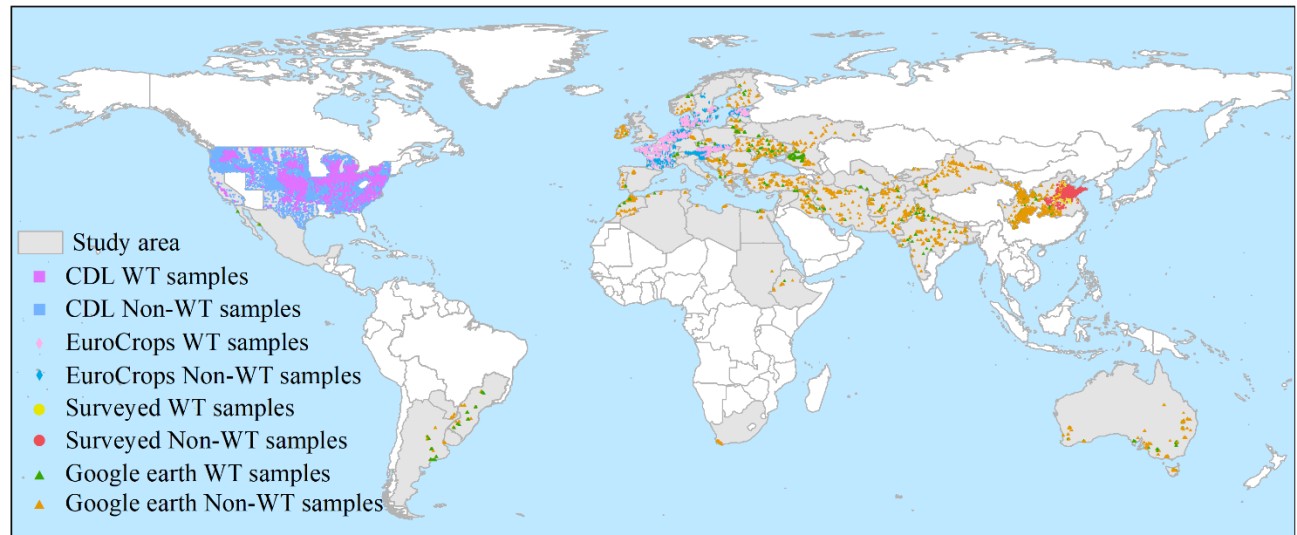

**Figure 1: Distribution of the study area and validation samples. The study area is the region covered in grey; The legend indicates the winter-triticeae (WT) crops samples and non-winter-triticeae (Non-WT) crops samples from Cropland Data Layer (CDL) dataset of the United States, the EuroCrops dataset of Europe, and field survey in China, as well as visual interpretation base on Google Earth images, respectively.**

## 2.2 Data

The data used in this study included: (1) reflectance data from Landsat 7, Landsat 8 and Sentinel-2; (2) Synthetic Aperture Radar (SAR) data from Sentinel-1; (3) field survey samples, visual interpretation samples, CDL and EuroCrops datasets; (4) agricultural statistical data. Reflectance data and SAR data were used to generate winter-triticeae crops maps; field survey samples, visual interpretation samples, and CDL and EuroCrops datasets, as well as agricultural statistical data were used to assess the performance of the proposed method.

### 2.2.1 Satellite data

In this study, we used all available Landsat 7 collection 2 data (USGS Landsat 7 Level 2, Collection 2, Tier 1) and Landsat 8 collection 2 data (USGS Landsat 8 Level 2, Collection 2, Tier 1), as well as Sentinel-2 data (Harmonized Sentinel-2 MSI: MultiSpectral Instrument, Level-2A) on the Google Earth Engine (GEE) platform to obtain NDVI from 2016 to 2022, all of which were surface reflectance (SR) products and have undergone atmospheric correction. The SR products of Landsat 7 and Landsat 8 have a spatial resolution of 30 m and a temporal resolution of 16 days. The spatial and temporal resolution of Sentinel-2 is 10 m and 5 days, respectively. We choose Landsat 7 satellite to obtain more available data although a malfunction in its scan line corrector. To ensure the data quantity and quality, we first removed the pixels with clouds. The quality band BQA was used to remove pixels with clouds from Landsat 7 and Landsat 8, and the quality band QA60 was used to remove pixels contaminated by clouds from Sentinel-2. Then, based on nearest neighbour method, we resampled the NDVI of Sentinel-2 to 30 m to keep the same spatial resolution as Landsat data. Furthermore, we obtained NDVI of all cloud-free pixels, and

chose the maximum values of monthly composites with 30 m spatial resolution, which has been proven effective for crop mapping and displaying crop growth stage (Huang et al., 2022). Last, we used linear interpolation and the Savitzky-Golay filter methods (Chen et al., 2004) to fill the missing values and smooth the NDVI series to reduce the contamination from cloud, rain and snow (Zheng et al., 2022). The above processes were run on the GEE platform.

The VH band with 10 m spatial resolution from SAR of Sentinel-1 was employed to distinguish winter-triticeae crops from other winter crops (i.e., winter rapeseed) (Dong et al., 2020). The data provided on GEE platform has undergone thermal noise removal, radiometric calibration, and terrain correction. We applied a refined Lee filter (Abramov et al., 2017) to alleviate the impact of speckle noise caused by the interferences between adjacent backscatter returns, and finally obtained the monthly maximum composite values of VH from 2016 to 2022 and resampled them to 30 m using the nearest neighbour method to keep consistency with NDVI. These operations were also run on the GEE platform.

### 2.2.2 Validation samples

The validation samples were obtained from: (1) field surveys, (2) Google Earth images, (3) CDL dataset and (4) EuroCrops dataset. We conducted field surveys in Hebei, Henan, Shandong, Anhui, and Jiangsu provinces in China in 2019 and 2020. The survey routes were pre-planned based on prior knowledge of the spatial distribution of winter-triticeae crops and transportation accessibility. In the fieldwork, we only selected large winter-triticeae crops fields with an area greater than 900 $m^2$, and used GPS (G120, UniStrong, Beijing, China) (Fu et al., 2023b) to mark the locations inside the fields. For non-winter-triticeae crops samples, we randomly selected large areas of non-winter-triticeae crops fields, forests, and grasslands around the pre-planned routes, and also used GPS to mark their locations. Finally, we processed these samples using Acrmap10.2 to maintain the same spatial projection as the identification map in China, resulting in a total of 3,054 winter-triticeae crops samples and 4,088 non-winter-triticeae crops samples. For other provinces in China and other countries (except US), we relied on high-resolution images from Google Earth from 2019 to 2020 for visual interpretation, which is a compensatory and effective method when ground truth samples cannot be obtained (Huang et al., 2022; Zheng et al., 2022). We first chose regions with available images during the growing season of winter-triticeae crops (section 2.3.3), and selected samples from these regions based on the texture features and colours. Winter-triticeae crops have deeper colour or stronger texture than winter rapeseed and grassland, and their roughness is lower than that of forest, which can be used to distinguish winter-triticeae crops from other land cover types (Fig .3a). The images of wetland and shrub show obvious differences from those of winter-triticeae crops. Wetland have dual characteristics of water and vegetation, and without regular texture features. Shrub have lower vegetation coverage and stronger graininess. These features make them easy to distinguish from winter-triticeae crops (Fig .3a). Crops with different growing season (such as maize, rice, and soybean) will not affect the visual interpretation. To ensure the accuracy of the samples, we then validated the selected samples on GEE platform by checking whether the NDVI temporal features of these samples matched the characteristics of winter-triticeae crops, and finally obtained 7,029 winter-triticeae crops samples and 8,897 non-winter-triticeae crops samples (Fig. 1).

In addition, we used CDL and EuroCrops datasets to further evaluate the performance of WTCI method. The CDL released annually has high accuracy in capturing crop distribution in US and has been widely used as a base map for crop dynamic monitoring and production estimation (Wang et al, 2019; Xu et al., 2023). We thus treated CDL labels as ground truth to validate the accuracy of our identification map in the US. Specifically, we first used Acrmap10.2 to randomly select samples

from pixels labelled with winter-triticeae crops, including winter wheat, double crop winter wheat/soybeans, winter wheat/corn, winter wheat/sorghum and winter wheat/cotton. Non-winter-triticeae crops samples were randomly generated in the remaining pixels, including other crops pixels in cultivated land and non-cultivated land pixels. Then we converted these samples into the same spatial projection as the identification map in the US. We finally obtained 7,500 winter-triticeae crops samples and 12,500 non-winter-triticeae crops samples in 2020 (Fig. 1). The EuroCrops dataset, supported by the German Space Agency

at DLR on behalf of the Federal Ministry for Economic Affairs and Climate Action (BMWK), is combines all publicly available self-declared crop reporting datasets from countries of the European Union. Importantly, this dataset utilizes a new version of Hierarchical Crop and Agriculture Taxonomy (HCAT) to provide a unified hierarchical representation scheme for all crops within the European Union (Schneider et al., 2023). We collected 10 countries (Austria, Belgium, Germany, Denmark, Estonia, France, Netherlands, Slovakia, Slovenia and Sweden) with winter-triticeae crops clearly labelled in EuroCrops dataset,

including winter spelt, winter barley, winter durum hard wheat, winter common soft wheat, winter triticale, winter rye and winter oats (https://zenodo.org/records/10118572), and these data cover the period from 2018 to 2021. We first convert the polygon file into point file using Acrmap 10.2, then randomly extracted winter-triticeae crops samples from the point file labelled with winter-triticeae crops in each country, and selected non-winter-triticeae crops samples from other land cover types, such as forest, grassland or other crops. We then transformed the spatial projection of these samples to be consistent

with the European identification map, and ultimately obtained 2,000 winter-triticeae crops samples and 3,000 non-winter-triticeae crops samples to assess the result of WTCI method in Europe (Fig. 1).

### 2.2.3 Agricultural statistical data

To evaluate the rationality in spatial distribution of winter-triticeae crops maps produced by the WTCI method, we thus collected the agricultural statistical data of winter-triticeae crops from 2017 to 2022 through the official websites of all

160 countries (Table S1) to compare its consistence with the identified area. Overall, we obtained the total statistical area data of winter-triticeae crops in each country and the statistical area data at the state (or province) or municipal or county level in 34 countries.

### 2.3 Method

The workflow for identifying winter-triticeae crops (Fig. 2) mainly includes four steps after pre-processing satellite data: (1)

selecting pixels with a maximum NDVI value greater than 0.4 during the winter-triticeae crops growing season as potential pixels; (2) developing the WTCI based on the unique characteristics of NDVI time series of winter-triticeae crops compared

with other land cover types; (3) calculating the WTCI value of potential pixels to quantify their similarity with winter-triticeae crops, and using thresholds to obtain the distribution maps of winter-triticeae crops; (4) evaluating the performance of WTCI method based on validation data.

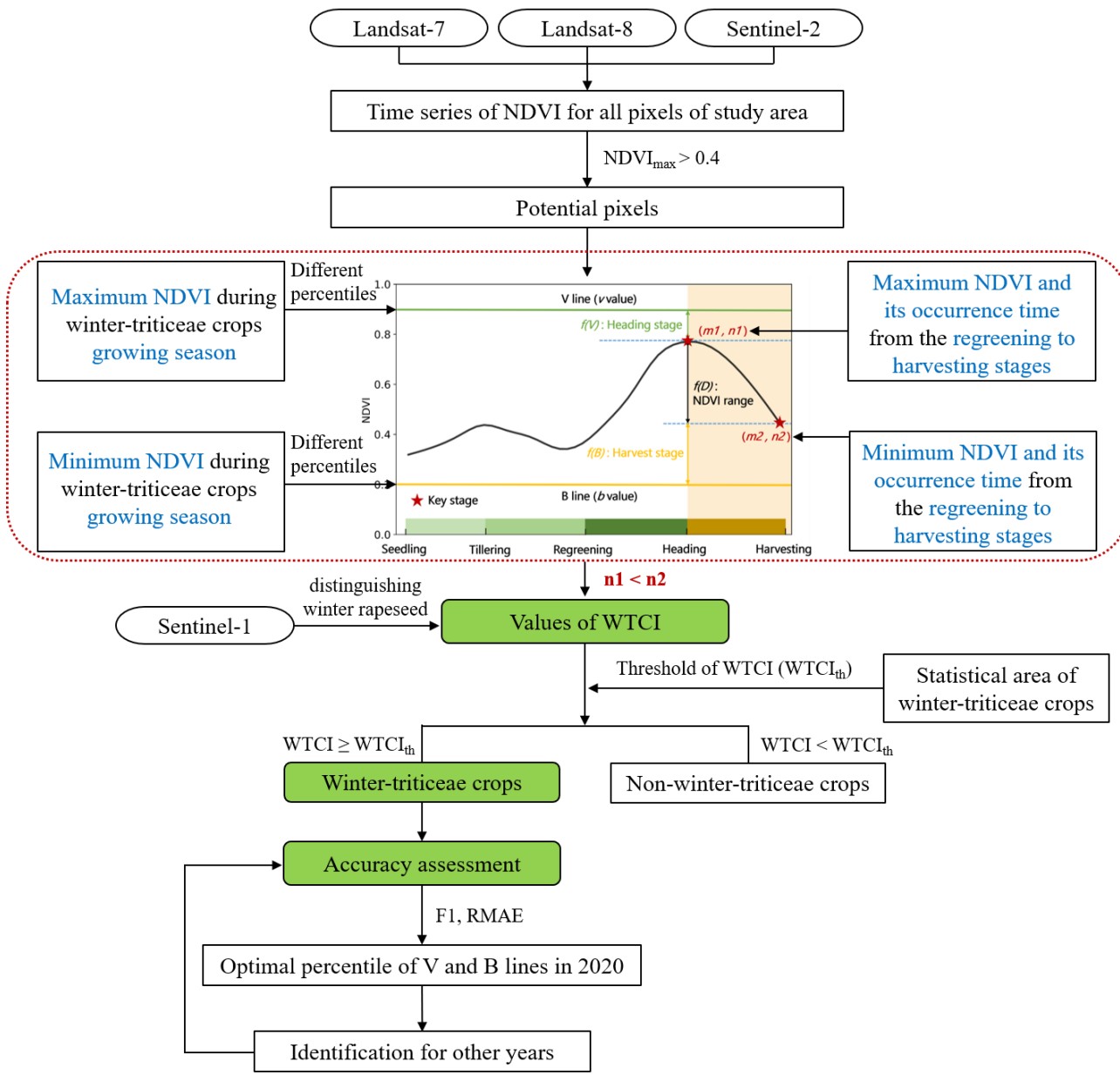

**Figure 2: The flowchart of identifying winter-triticeae crops using the WTCI method.**

### 2.3.1 Time series characteristics of NDVI for different land cover types

The design of the Winter-Triticeae Crops Index (WTCI) is based on the analysis of NDVI time series for different land cover types. Specifically, we first selected the NDVI time series of each pixel during the growth season (i.e., autumn to summer of the following year) of winter-triticeae crops. Some regional and global scale studies have reported that NDVI greater than 0.4 usually indicates vegetation cover (Ma et al., 2022; Peng et al., 2019; Xu et al., 2023; Yang et al., 2024; Yang et al., 2024). Therefore, pixels with a maximum NDVI greater than 0.4 during the selected growth period were retained as the potential pixels. After applying these steps, the main remaining land cover types in the potential pixels were forest, grassland, cultivated land, wetland and shrub.

There are significant differences in the temporal variations of NDVI between winter-triticeae crops and natural vegetation types (i.e., deciduous forest, evergreen forest, shrub and grassland) as well as wetland during the growing season of winter-triticeae crops (Fig. 3b). Specifically, in the period from seedling to tillering stages, winter-triticeae crops are in a state of slow growth, with their NDVI gradually increasing. In contrast, natural vegetation types are in the deciduous stage and exhibit a continuous decrease in NDVI during this period, and wetland also exhibit the similar characteristics (Fig. 3b). From the regreening to the heading stages, the NDVI of winter-triticeae crops rapidly increases and reaches its maximum value, while the increase of NDVI of natural vegetation types and wetland tends to lag behind that of winter-triticeae crops (Fig. 3b). Furthermore, the NDVI of winter-triticeae crops show a downward trend and reach their lowest value during the harvesting stage. However, the NDVI values of natural vegetations and wetland rapidly increase at this time (Fig. 3b). Additionally, except for winter rapeseed, there are significant differences in the growth season of maize, rice, and soybean compared to that of winter-triticeae crops. Therefore, these crops will not interfere with the identification of winter-triticeae crops, even if they have similarities in the NDVI time series characteristics with winter-triticeae crops.

Based on the above analysis, there are two periods that can be used to distinguish between winter-triticeae crops and other land cover types, i.e., the seedling to tillering stages and the heading to harvesting stages (Fig. 3b), during which the NDVI of winter-triticeae crops and other land cover types showed opposite temporal variations. Compared with the period from seedling to tillering, the NDVI characteristics of winter-triticeae crops from heading to harvesting stages are more stable, and more significantly different from those of other land cover types. A previous study on the relatively weak growth and not obvious increase of NDVI of winter-triticeae crops from seedling to tillering stages (Wang et al., 2015) further supports our finding. Therefore, this study used the NDVI time series characteristics of winter-triticeae crops from heading to harvesting stages to design the WTCI.

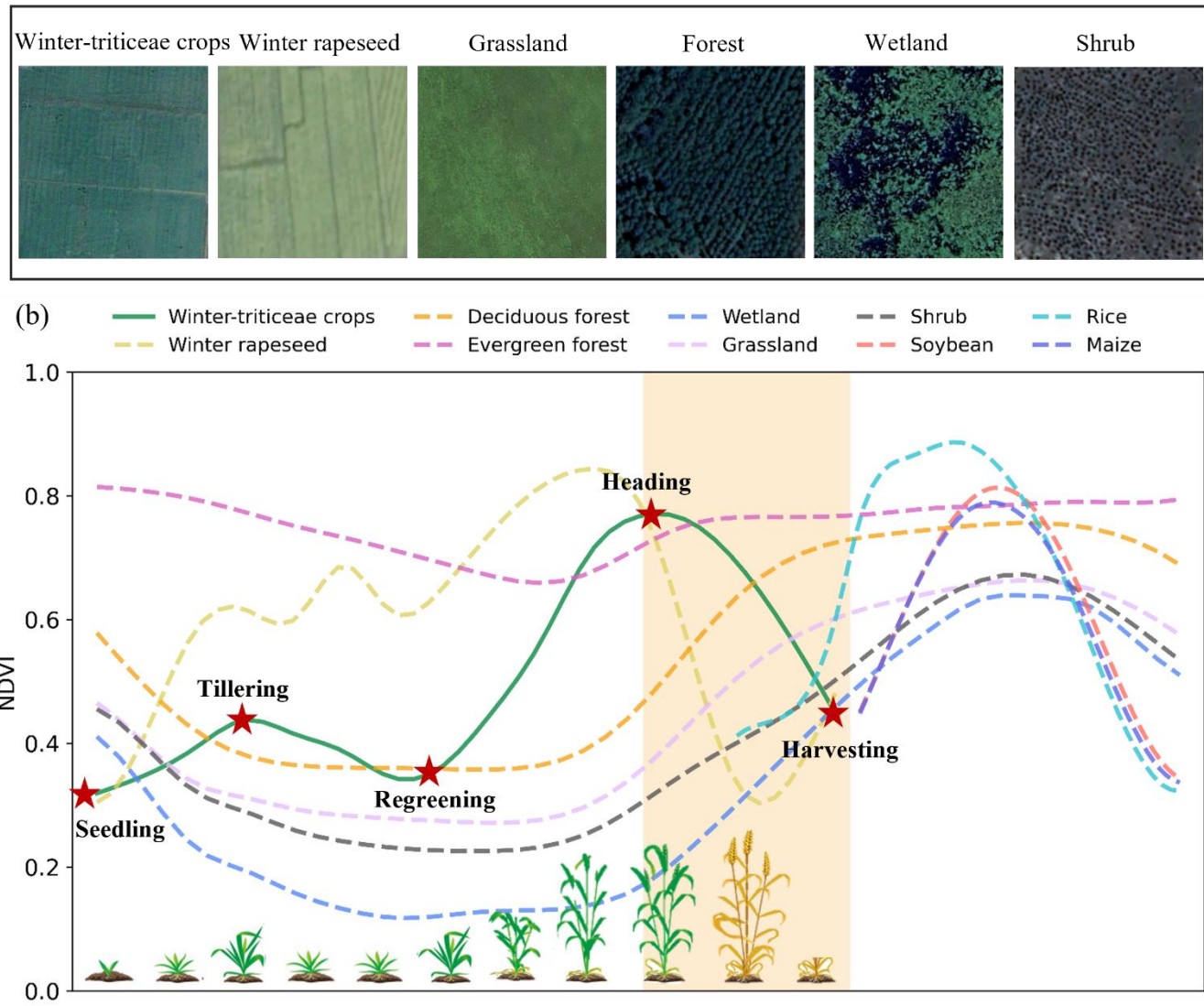

(a)

Winter-triticeae crops | Winter rapeseed | Grassland | Forest | Wetland | Shrub

(b)

**Figure 3: Example of the (a) textures and colours on the high-resolution images from © Google Earth and (b) NDVI time series characteristics of different land cover types. The red five-pointed stars represent the different phenological stages of winter-triticeae crops.**

### 2.3.2 Development of the Winter-Triticeae Crops Index

Based on the comparison of the NDVI time series characteristics of winter-triticeae crops with other land cover types, the unique characteristics of winter-triticeae crops during the growing season can be summarized as: (1) the NDVI of winter-triticeae crops peaks at the heading stage, which is close to the maximum value of natural vegetation during its growing season; (2) winter-triticeae crops have low NDVI values during the harvesting stage, when the surface tends to be close to bare land

after crop removal. On the contrary, the NDVI of natural vegetation approaches its peak in a year. To quantity the above

210 characteristics, this study set an upper boundary to denote vegetation (V line) and a lower boundary to indicate bare land (B line) (Fig. 4). Then, three indicators, $f(D)$, $f(V)$, and $f(B)$, were constructed to represent the unique NDVI characteristics of winter-triticeae crops from the heading to the harvesting stages (Fig. 4), and their integrate (i.e., WTCI) were employed to determine whether the potential pixel is winter-triticeae crops:

$$WTCI = f(D) \times f(V) \times f(B), n1 < n2 , \tag{1}$$

where $n1$ and $n2$ represent the time when the maximum and minimum NDVI appear, respectively (Fig. 4). It should be noticed that Eq. (1) was used to identify the winter-triticeae crops only when $n1<n2$, i.e., the maximum NDVI should appear before the minimum NDVI.

Specifically, $f(D)$, $f(V)$, and $f(B)$ were designed as follows:

$$f(D) = \frac{1}{1+e^{\left(\frac{v-b}{2}-D\right)}}, D = m1 - m2 , \tag{2}$$

$$f(V) = 1 - V^2, V = \begin{cases} 1, & m1 \le b \\ \frac{v-m1}{v-b}, & b < m1 \le v \\ 0, & m1 > v \end{cases} , \tag{3}$$

$$f(B) = 1 - B^2, B = \begin{cases} 1, & m2 \ge v \\ \frac{m2-b}{v-b}, & b \le m2 < v \\ 0, & m2 < b \end{cases} , \tag{4}$$

where $v$ and $b$ represent the NDVI corresponding to the V and B lines, respectively. $m1$ and $m2$ represent the maximum and minimum NDVI of the potential pixel from the heading to harvesting stages (Fig. 4), respectively. $f(D)$ quantifies the proximity of the range of NDVI variation between the potential pixels and those of winter-triticeae crops. Given a pixel with $D$ (i.e., $m1$

$- m2$) closer to the value of $v - b$, the higher the value of $f(D)$, the higher the likelihood that it represents a winter-triticeae crops. $f(V)$ quantifies the proximity of the maximum NDVI ($m1$) of the potential pixels with that of vegetation. The pixels closer to the V line at the $n1$ period (i.e., $m1$ approaches $v$) are more likely to be winter-triticeae crops. Additionally, $f(B)$ quantifies the proximity of the minimum NDVI ($m2$) of the potential pixel with that of bare land. Pixels closer to the B line at the $n2$ period (i.e., $m2$ approaches $b$) have a greater likelihood of being winter-triticeae crops. The algorithms of $f(D)$, $f(V)$,

and $f(B)$ reported by Xu et al. (2023) were used in this study.

Winter-triticeae crops should simultaneously have all the above three characteristics, this means that the WTCI should be designed to integrate these three indicators. The values of $f(D)$, $f(V)$, and $f(B)$ range from 0 to 1. Therefore, WTCI varies between 0 and 1, and pixels with higher WTCI have a greater probability of being winter-triticeae crops. In addition, this study

uses agricultural statistical area of winter-triticeae crops to determine the threshold of WTCI. The potential pixels (*N*th) with

high WTCI values are considered winter-triticeae crops in a given identification unit, and the total area of all *N* potential pixels should be equal to the agricultural statistical area of the identification unit.

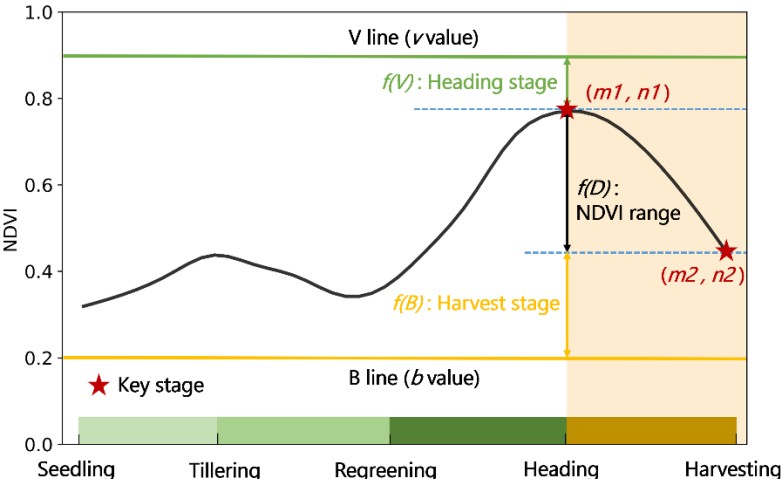

**Figure 4: Characteristics of NDVI time series for designing the Winter-Triticeae Crops Index. The black solid line represents the**
**NDVI time series of winter-triticeae crops. The green and orange solid lines represent the V line and the B line, respectively; The red five-pointed stars indicate the heading and harvesting stages of winter-triticeae crops; *m1* and *n1* represent the maximum value of NDVI and the time when the maximum value occurs during the study period; *m2* and *n2* represent the minimum value of NDVI and the time when the minimum value occurs during the study period.**

### 2.3.3 WTCI-based winter-triticeae crops identification

In this study, we considered each state (or province) as an identification unit in China, Brazil, India, Australia and US, and the threshold of WTCI was determined based on statistical area at state (or province) scale. For the remaining countries, we treated each country as an identification unit, and the threshold of WTCI was calculated relied on statistical area at national scale. The annual statistical area was used to determine the threshold of WTCI for each identification unit in the current year. Furthermore, given the diversity and complexity of land cover types and agricultural planting structures in the study area, we used different

percentile combinations of the V and B lines. Specifically, this study referred to crop calendar data provided by the United States Department Agriculture (USDA) (https://ipad.fas.usda.gov/ogamaps/cropcalendar.aspx) to determine the growth season of winter-triticeae crops in each country. Then, we extracted the maximum and minimum NDVI of all potential pixels in each identification unit during the growing season of winter-triticeae crops. We further obtained different percentiles (5%, 20%, 40%, 60%, 80%, and 95%) of the maximum and minimum NDVI for each identification unit, respectively, corresponding to

*v* and *b* in the Eq. (2), Eq. (3) and Eq. (4). In addition, the *m1* and *m2* were automatically searched in the NDVI curve between the regreening and harvesting stages of winter-triticeae crops. In this study, the regreening stage was based on the start time of spring in the northern (March) and southern (September) hemispheres (Ren et al., 2019), and the harvesting stage referred to the crop calendar provided by USDA. We first determined the *m1* and *n1* of each potential pixel, then we looked for the *m2* in

the period after $n1$, and further calculated WTCI. Pixels that do not meet this condition (i.e., $n1<n2$) are identified as non-winter-triticeae crops. In addition, we determined the optimal combination of V and B lines in each identification unit according to the identification accuracy at the pixel scale (F1 score) and the relative mean absolute error (RMAE) between identified and agricultural statistical areas. For countries lacking agricultural statistical data, the optimal combination was decided solely based on the F1 score. Based on the optimal combination of V and B lines of each identification unit in 2020, winter-triticeae crops from 2017 to 2019 and 2021 to 2022 were identified to evaluate the temporal transferability of the WTCI.

The identification of winter-triticeae crops in the study area may be affected by winter rapeseed and garlic, as these crops have similar growth season and spectral characteristics with winter-triticeae crops (Fu et al., 2023b; Tian et al., 2021). Winter rapeseed is mainly distributed in China, India and parts of Europe. The planting area of winter rapeseed in some states (or provinces) of China and India is equivalent to or even higher than that of winter-triticeae crops, while the planting area in countries such as France, Germany, Poland, Britain, Hungary, and Ukraine accounts for 17%-32% of the planting area of winter-triticeae crops. Winter garlic is mainly distributed in some provinces of China, Spain, and Ukraine. However, the planting area of winter garlic is very small compared to that of winter-triticeae crops and winter rapeseed. For example, the planting area of winter garlic in China, the largest planting country, only accounted for about 2% of the winter crops (http://data.stats.gov.cn/). Therefore, this study only distinguished between winter rapeseed and winter-triticeae crops (Fig. S1). The NDVI time series of winter rapeseed shows a downward trend from the heading to harvest stages of winter-triticeae crops, which is resemble winter-triticeae crops (Fig. 3b). Tao et al. (2023) have also demonstrated that winter rapeseed and winter-triticeae crops have similar NDVI characteristics, making it difficult to distinguish them only based on optical images (Veloso et al., 2017). Fortunately, previous studies have indicated that the VH (vertical transmit and horizontal receive) band can effectively eliminate the interference from winter rapeseed in the identification of winter-triticeae crops in China and Europe (Dong et al., 2020; Huang et al., 2022). Therefore, we distinguished winter rapeseed and winter-triticeae crops based on the methods of these studies. Specifically, the VH threshold set by Dong et al. (2020a), which was obtained by comparing winter-triticeae crops and winter rapeseed filed samples, was employed in this study. In regions of India where winter rapeseed is planted, we calculated the VH values from Sentinel-1 images in March considering the lower latitude and earlier harvest period of these regions. In other Asian regions where winter rapeseed is grown, this study obtained VH values for April. This study identified these pixels with VH values greater than -15.5 in March or April as non-winter-triticeae crops. Similarly, in some European countries, we calculated VH values for May, and considered that pixels with VH values greater than -15.5 were non-winter-triticeae crops (Huang et al., 2022).

## 2.4 Accuracy assessment

This study evaluated the accuracy at both pixel and regional scales. The producer's accuracy (PA), user's accuracy (UA), overall accuracy (OA) and F1 score (Congalton, 1991; Hripcsak and Rothschild, 2005; Lin et al., 2022) were employed to validate the identification accuracy at the pixel scale. At the regional scale, we obtained the identified areas of winter-triticeae

crops based on the total pixel area of winter-triticeae crops on the identification maps. In China, Brazil, India, Australia and the US, we used the statistical area at low-level administration, such as municipal or county scale, to validate the accuracy of identified area at state (or province) scale. For other counties, the statistical area of all states or provinces or municipalities or counties included in each country was used to evaluate the accuracy at national scale. The correlation coefficient ($R^2$) and relative mean absolute error (RMAE) were used to examine the consistency between the identified area and the statistical area (Shen et al., 2023; Zheng et al., 2022).

## 3 Results

### 3.1 The spatial transferability of the WTCI method

The spatial distribution map of winter-triticeae crops in 66 countries in 2020 was first produced based on the WTCI method (Fig. 5), which effectively presented the distribution of winter-triticeae crops in the study area. Specifically, the winter-triticeae crops were mainly distributed in most European countries and Asian plains (Fig. 5b and 5c). To display the detailed information of the map of winter-triticeae crops, we selected twelve typical areas in different countries to zoom in and compared them with high-resolution images from Google Earth (Fig. 6). In general, despite some noise, the identification map clearly displays the fields planted with winter-triticeae crops and effectively distinguishes roads and rivers between the fields. In addition, we compared the spatial distribution map of winter-triticeae crops in this study with some existing products in Europe (Huang et al., 2022) and China (Dong et al., 2020), which also have a spatial resolution of 30 m. The spatial distribution of winter-triticeae crops fields in the maps produced in this study was similar to other studies, and the maps generated by WTCI method had less noise and clearer boundaries of roads and rivers (Fig. S3).

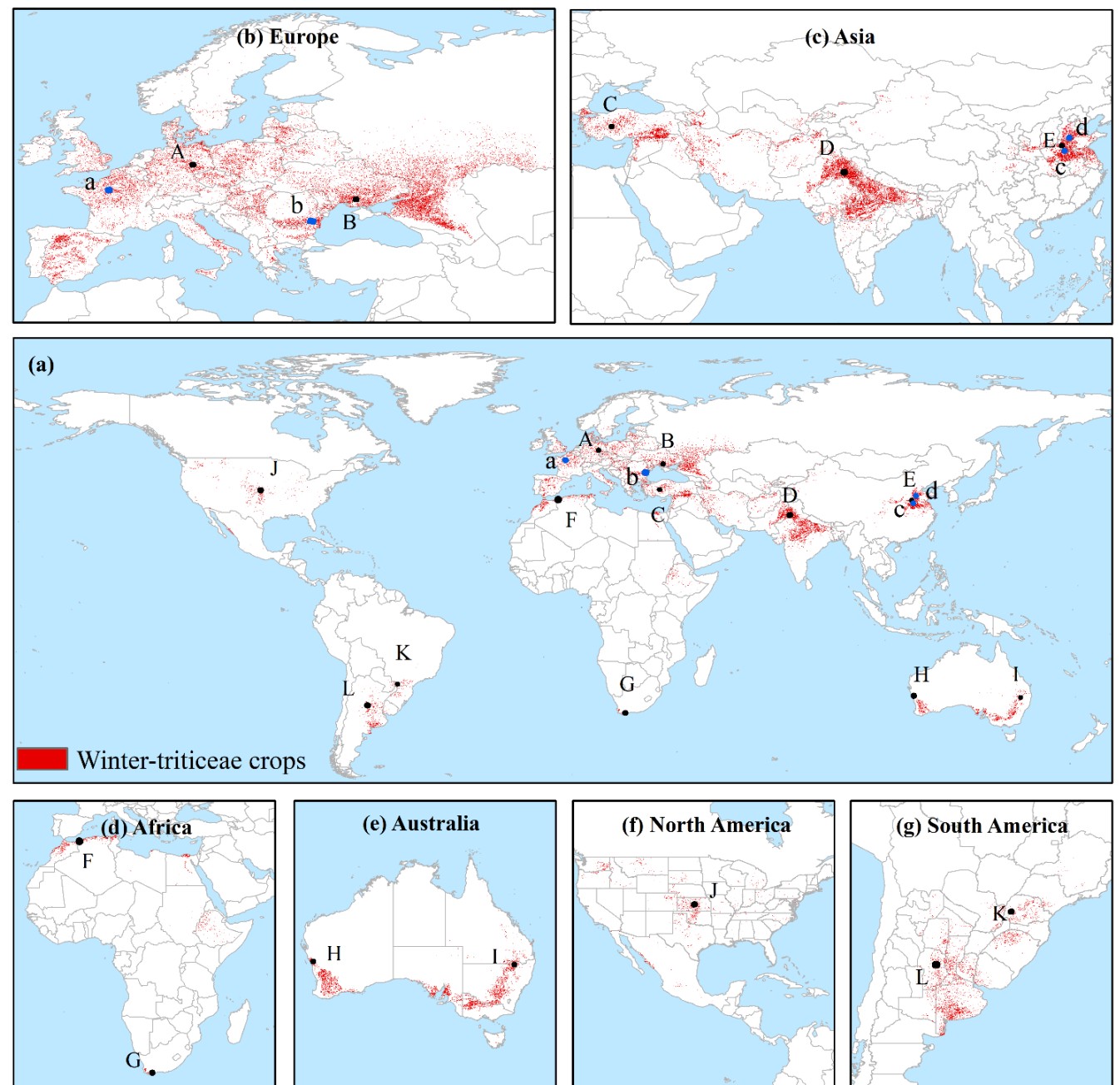

**Figure 5: Spatial distribution of winter-triticeae crops in the study area in 2020.** (a) shows the distribution of winter-triticeae crops in 66 countries; (b-g) show the zoomed-in maps of Europe, Asia, Africa, Australia, North America and South America, respectively.

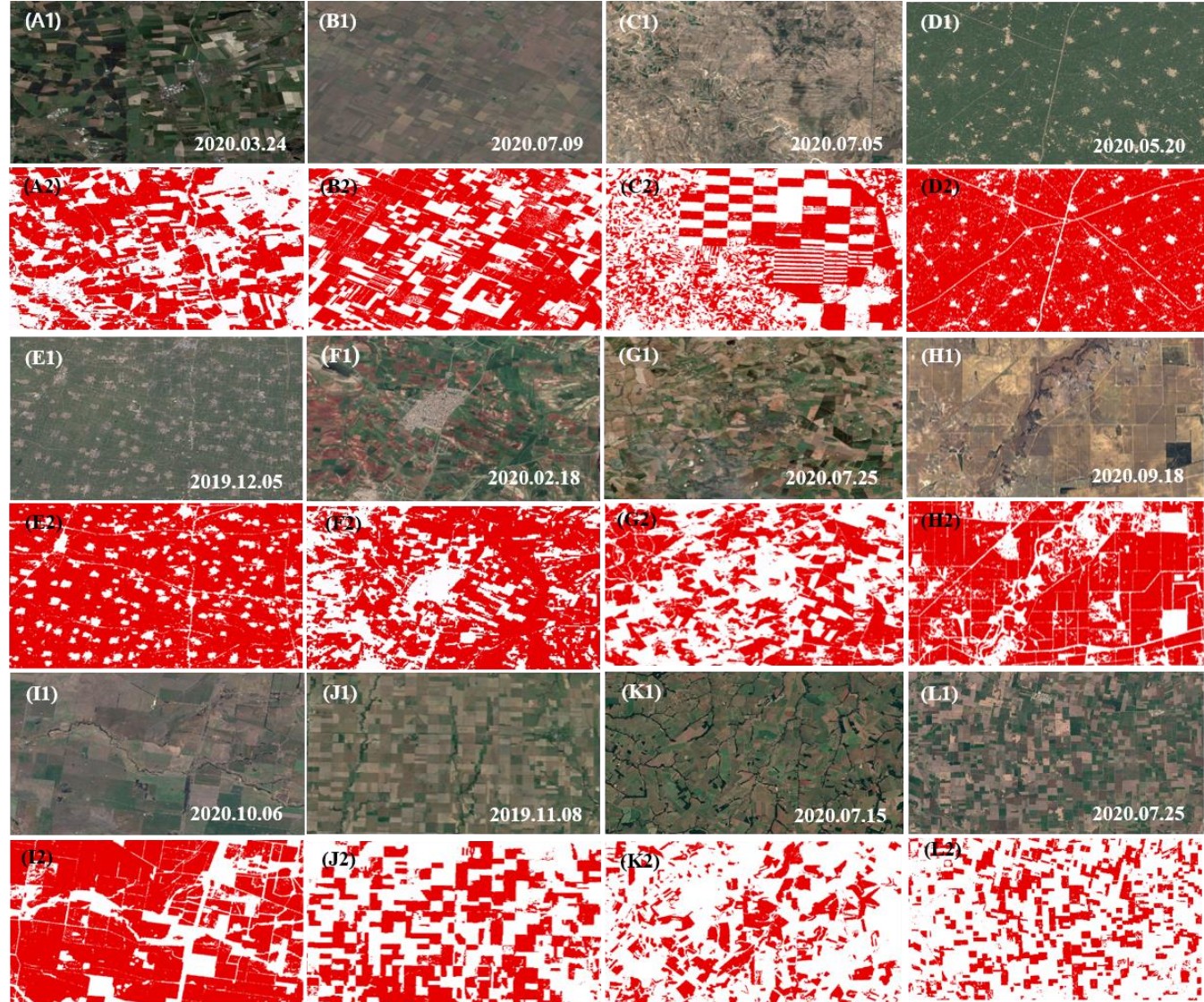

**Figure 6: Comparison between the identification maps of winter-triticeae crops and high-resolution images from © Google Earth in the study area. (A1-L1) represent the high-resolution images from Google Earth of different regions; (A2-L2) represent the zoomed-in maps of area A-L in Figure 5.**

Based on the field survey samples and visual interpretation samples, the overall accuracy (OA), producer's accuracy (PA), and user's accuracy (UA) of the winter-triticeae crops identification maps in 65 countries (except US) were 87.7%, 81.12% and 87.85%, respectively, and the F1 score was 84.04% (Fig. 7). PA and UA varied between 52% and 97.73%, 63.64% and 97.83% over the various countries, and OA and F1 ranged from 70.86% to 96.05% and 65.63% to 96.09%, respectively. At state (province) scale, the variation range of OA and F1 score in China were 77.68% to 95.9% and 71.79% to 94.47%, respectively (Fig. 8a). In Brazil, the OA and F1 score were in the range of 76.99%-94.74% and 78.26%-96.24% (Fig. 8b). The OA in India

was between 67.53% and 92.07%, and the F1 score was between 65.24% and 92.05% (Fig. 8c). The OA and F1 score in Australia lied in the range of 79.21% to 91.67% and 69.23% to 91% (Fig. 8d). In general, the F1 score in most of the identification units was greater than 75%, indicating that the WTCI method shows satisfactory accuracy in identifying winter-triticeae crops. The regions with F1 scores less than 75% were mainly found in small winter-triticeae crops planting areas and complex winter crop types, such as Croatia (HRV), Albania (ALB), Sichuan (SC) province in China, and Bihar (BR) state in India. On the contrary, the identification accuracy of regions with larger planting areas of winter-triticeae crops was significantly higher than that of regions with smaller planting areas.

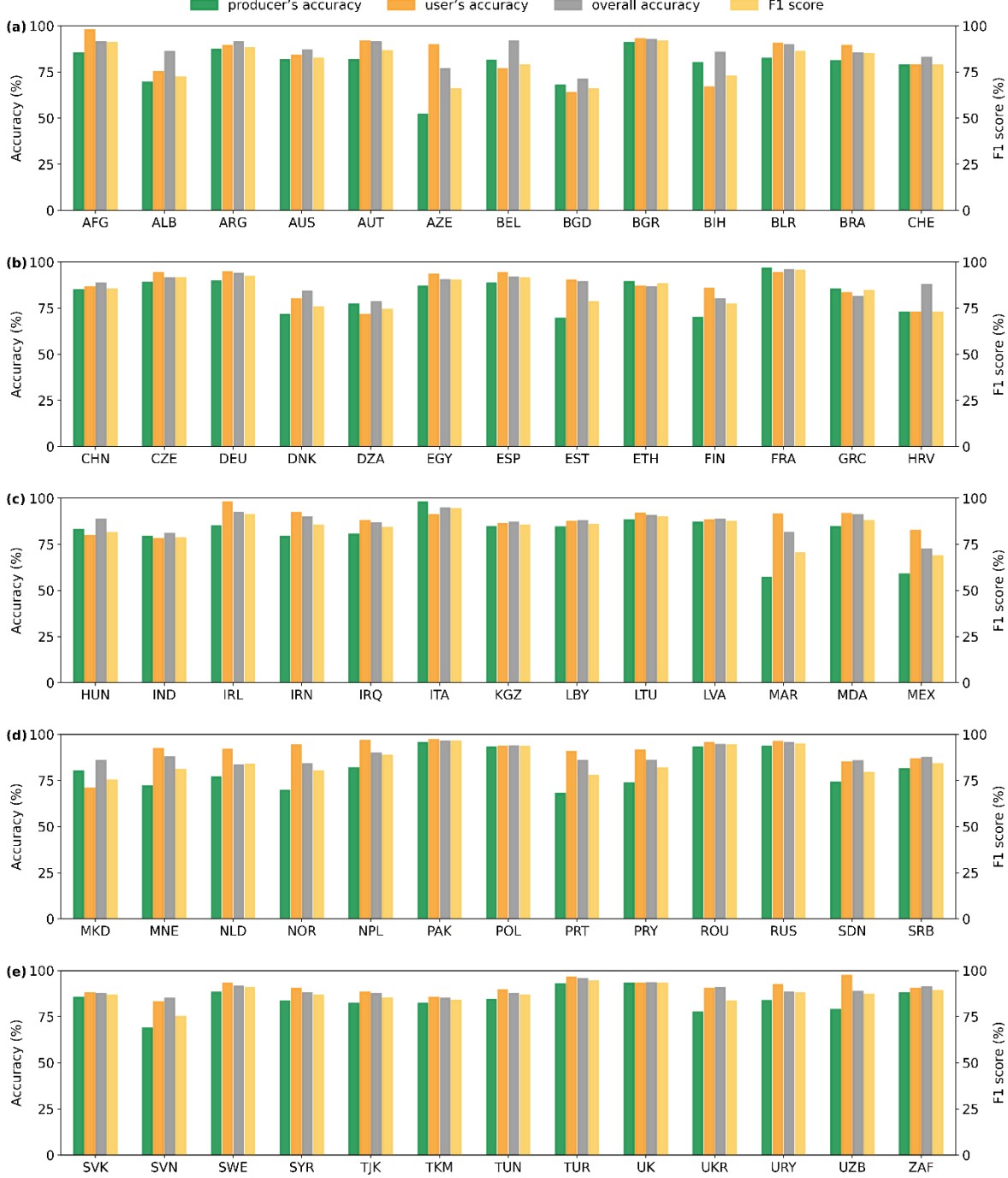

Figure 7: The producer's accuracy (PA), user's accuracy (UA), overall accuracy (OA) and F1 score of the identification maps of winter-triticeae crops at national scale in 2020. The abbreviations of countries are shown in Table S2 in the supplement.

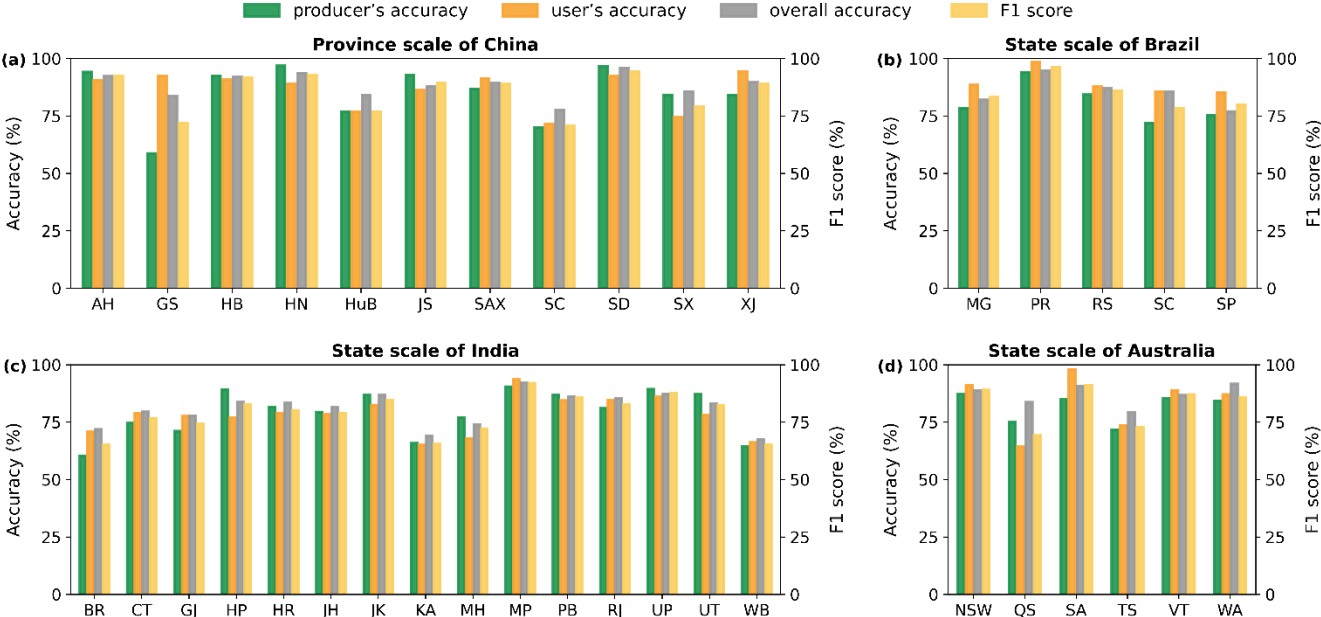

**Figure 8: The producer's accuracy (PA), user's accuracy (UA), overall accuracy (OA) and F1 score of the identification maps of winter-triticeae crops at state (province) scale in 2020. (a-d) represent the identification accuracy at state (province) scale in China, Brazil, India and Australia, respectively. The abbreviations of states (provinces) are shown in Table S3 in the supplement.**

In addition, compared to the agricultural statistical area in different administrative units in 2020, the WTCI method can effectively estimate the area of winter-triticeae crops. At national scale, the $R^2$ between the identified and the statistical areas of winter-triticeae crops ranged from 0.62 to 1, with an RMAE of 8.47% to 38.51% (Fig. 9a and 9b). At state (province) scale, the $R^2$ and RMAE between identified and statistical areas in China were between 0.75-0.99 and 12.64%-45.1%, respectively (Fig. 10a1 and 10a2). In Brazil, the $R^2$ was in the range of 0.84 to 0.91, with RMAE of 36.04% to 48.02% (Fig. 10b1 and 10b2). The $R^2$ and RMAE of 15 states in India ranged from 0.58 to 0.98 and 6.12% to 47.61%, respectively (Fig. 10c1 and 10c2). The $R^2$ and RMAE in Australia varied from 0.79 to 0.98 and 23.61% to 38.43%, respectively (Fig. 10d1 and 10d2). Overall, all of these results demonstrate that the WTCI method exhibits reliable spatial applicability in identifying winter-triticeae crops.

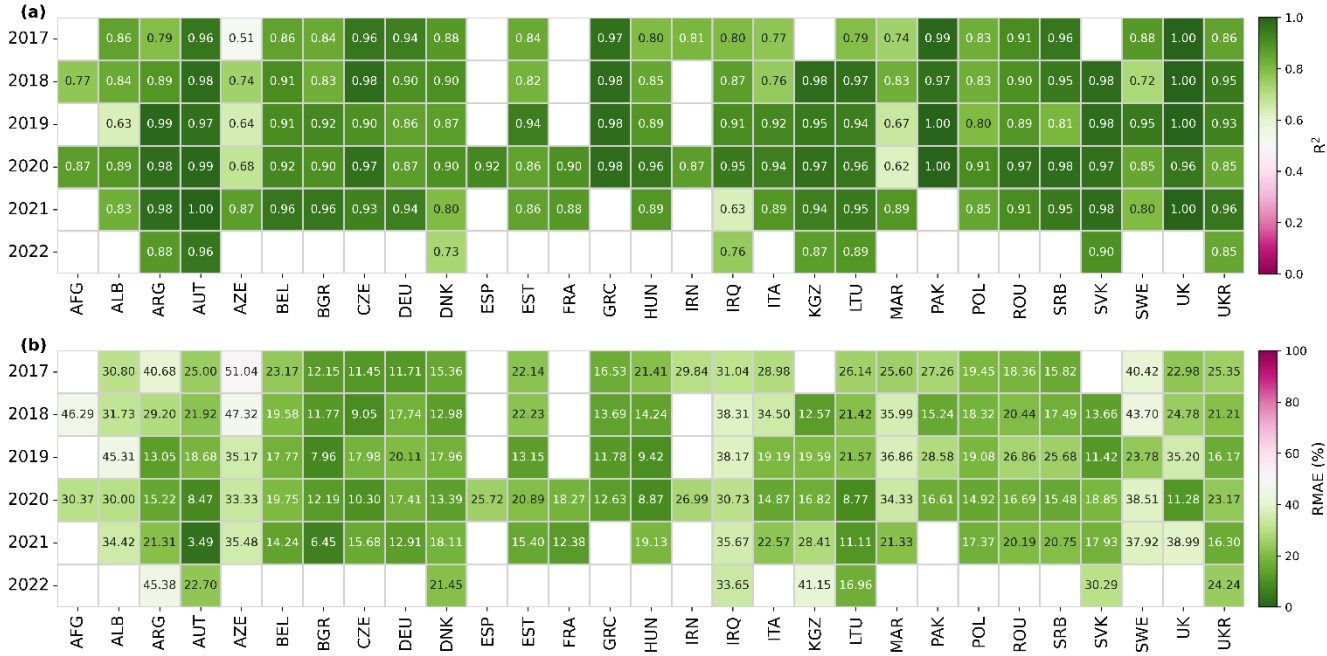

**Figure 9:** Comparison between identified and statistical areas of winter-triticeae crops at national scale from 2017 to 2022. (a) and (b) show the correlation coefficient and RMAE between identified and statistical areas, respectively.

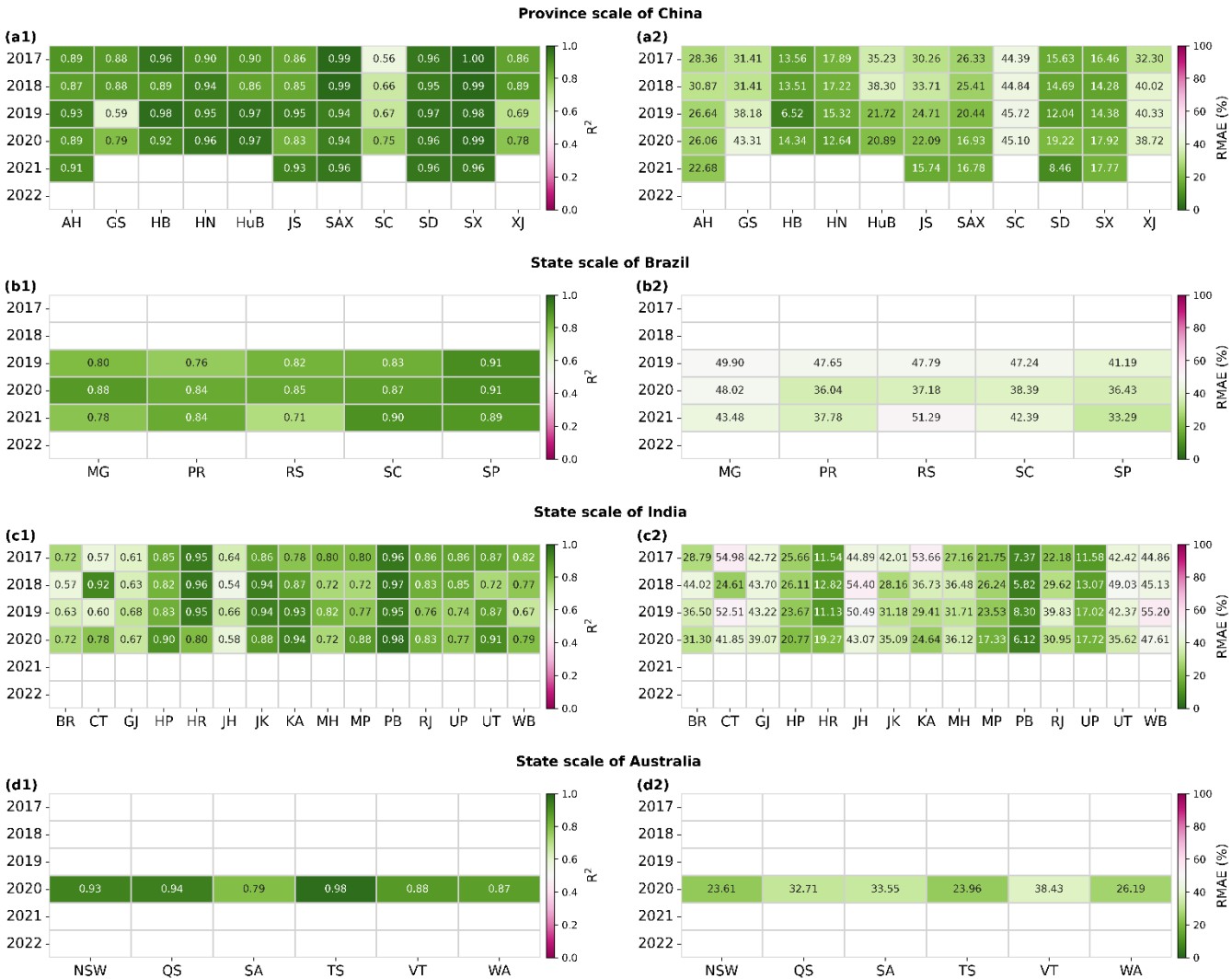

**Figure 10: Comparison between identified and statistical areas of winter-triticeae crops at state (province) scale from 2017 to 2022. (a1-d1) represent the correlation coefficient at state (province) scale in China, Brazil, India, and Australia, respectively; (a2-d2) represent the RMAE at state (province) scale in China, Brazil, India, and Australia, respectively.**

### 3.2 The temporal transferability of the WTCI method

The comparison between the identified and statistical areas of winter-triticeae crops indicates that the WTCI method can be effectively applied to other years. At national scale, the $R^2$ between identified and statistical areas of winter-triticeae crops in all years was between 0.51-1, and RMAE was between 3.49%-51.04% (Fig. 9a and 9b). At state (province) scale, the $R^2$ and RMAE ranged from 0.56 to 0.99 and 6.52% to 45.72% in China, respectively (Fig. 10a1 and 10a2). In Brazil, the range of these two metrics was from 0.71 to 0.91 and 33.29% to 51.29%, respectively (Fig. 10b1 and 10b2). In India, they varied from 0.54 to 0.97 and 5.82% to 55.2%, respectively (Fig. 10c1 and 10c2). The $R^2$ in most identification units were more than 0.6,

and RMAE was less than 30%. These results illustrate that there is good consistency between the identified and statistical areas of winter-triticeae crops, confirming the stable temporal transferability of the proposed method.

### 3.3 The performance of the WTCI method validated using CDL and EuroCrops datasets

The distribution map of winter-triticeae crops exhibited high consistency with CDL and EuroCrops datasets. In 2020, the OA and F1 score in the US were 86.84% and 82.09%, respectively, and the PA and UA were 76.96% and 88.13%, respectively (Fig. 11 and Table S4). The performance of the WTCI method varied by state. For all states planting winter-triticeae crops, the OA varied from 70.42% to 94.24%, and the F1 score ranged from 66.67% to 91.01% (Fig. 11a-11c and Table S4). In major planting states, such as Kansas, Oklahoma and Texas, the planting area of winter-triticeae crops accounted for approximately 50% of the total area of winter-triticeae crops in the US, with OA and F1 score over 92% and 85%, respectively (Fig. 11 and Table S4). The identified area by WTCI method also exhibited good consistency with the US official statistical data. At national scale, the $R^2$ and RMAE were 0.89 and 28.9%, respectively (Fig. 12a). At state scale, the $R^2$ varied between 0.52 to 0.96, and the RMAE was in 9.01%-57.84% (Fig. 12b-12w). Among the 10 European countries from EuroCrops datasets, the OA, F1 score, PA and UA ranged from 71.22% to 94.79%, 67.67% to 90.14%, 63.68% to 84.77% and 71.43% to 96.24%, with the mean value of 83.88%, 78.87%, 73.18% and 86% (Fig. 11d and Table S5), respectively. In general, the OA and F1 score in most of regions of US and Europe were higher than 80% and 75%, implying that the WTCI method exhibited satisfactory performance compared to the CDL and EuroCrops datasets. Additionally, we further presented spatial details of the identification map produced by the WTCI method in US and Europe. The results indicate that the identification map can effectively capture the field distribution of winter-triticeae crops in CDL and EuroCrops datasets (Fig. 13).

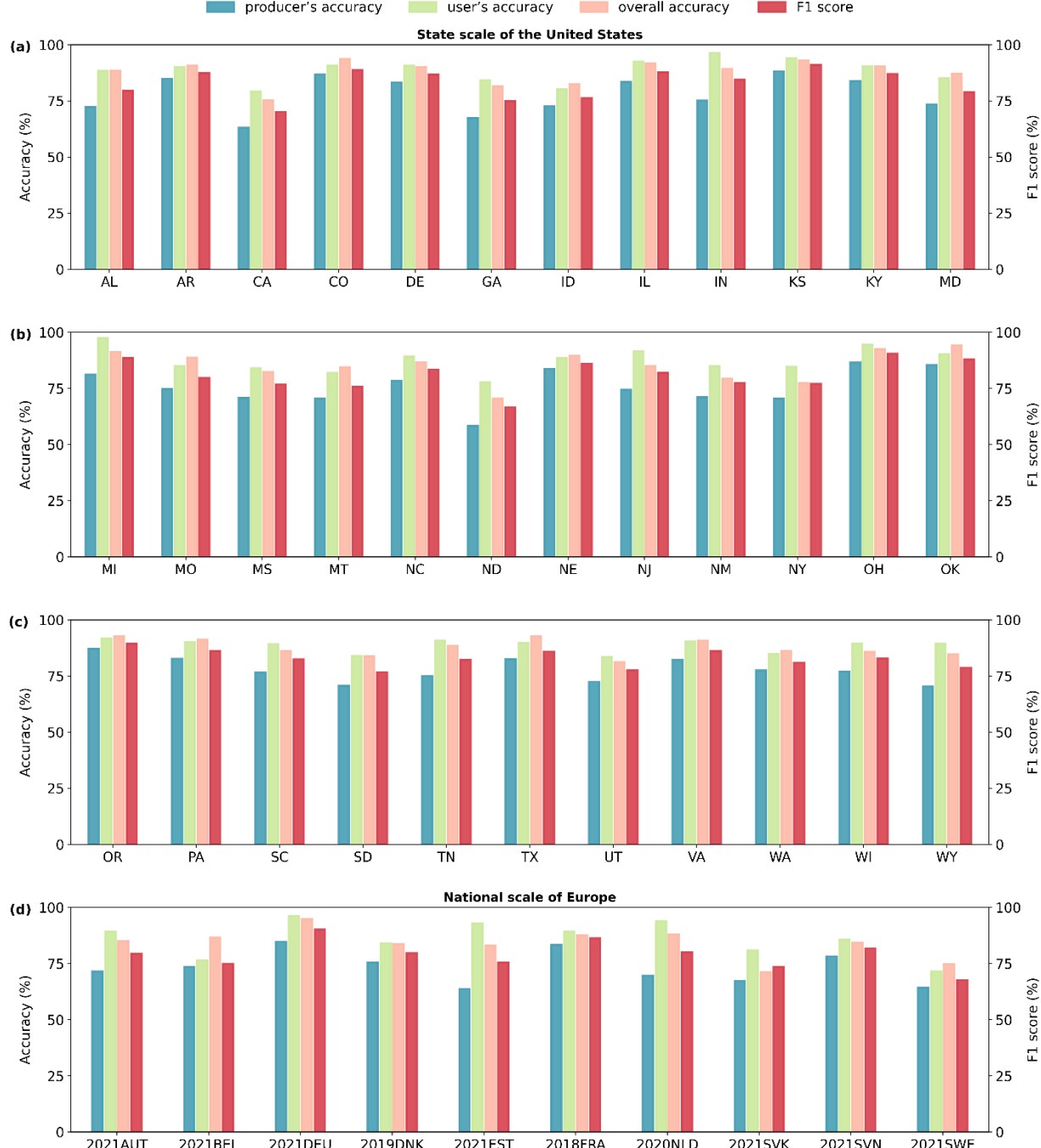

**Figure 11: The producer's accuracy (PA), user's accuracy (UA), overall accuracy (OA) and F1 score of the identification maps of winter-triticeae crops in the US and Europe. The abbreviations of countries and states are shown in Table S2 and S3 in the supplement. 2018FRA indicates the identification accuracy of the country in 2018.**

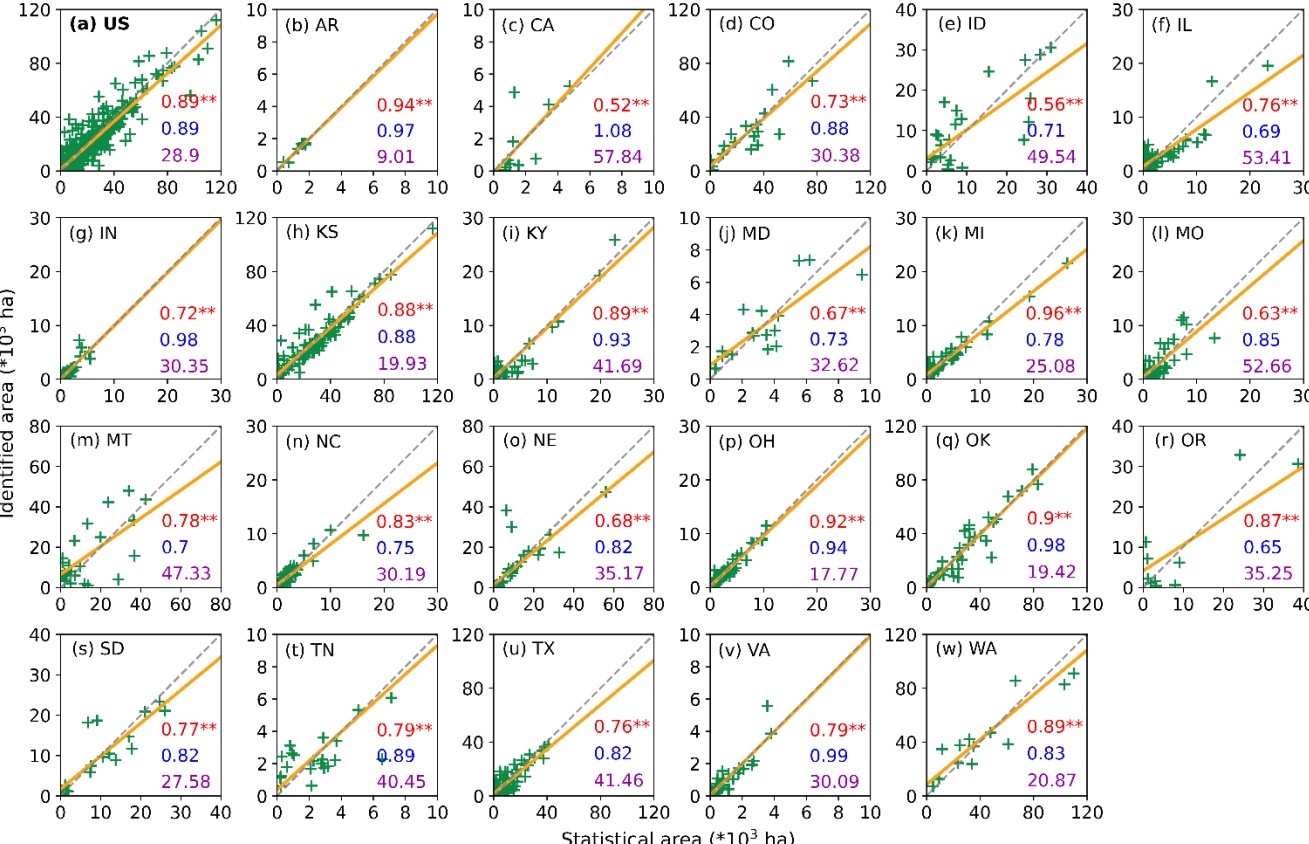

**Figure 12: Comparison between identified and statistical areas of winter-triticeae crops in 2020 in the US. (a) show the results between identified and statistical areas at national scale; (b-w) show the results between identified and statistical areas for each state, respectively. The green symbols represent the counties of each state. The yellow solid lines are the regression lines, and the grey short-dashed lines are the 1:1 lines. The red, blue and purple numbers represent R², slope and RMAE values between identified and statistical areas, respectively.**

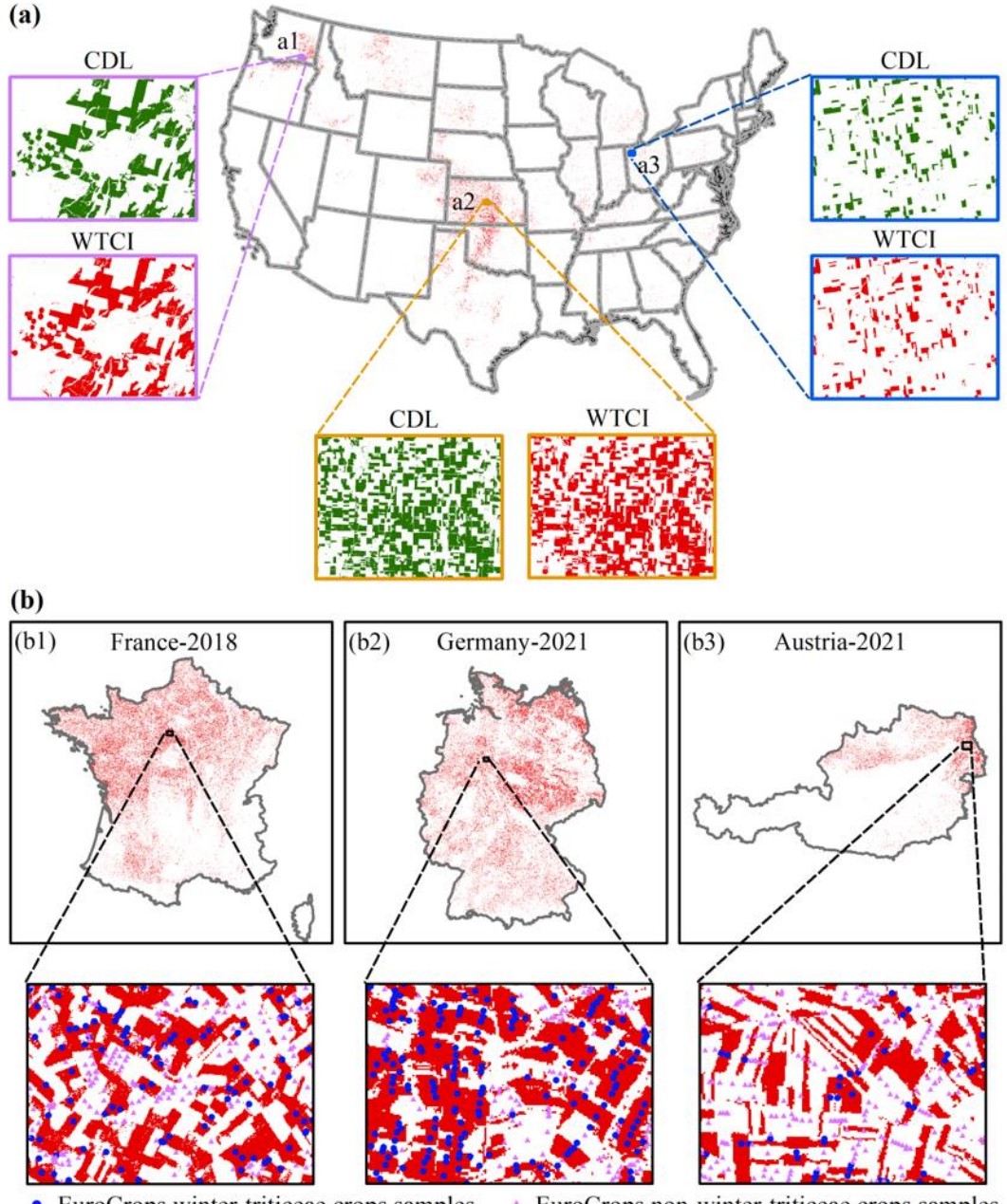

**Figure 13: Comparison of the identification maps of winter-triticeae crops with CDL and EuroCrops datasets. (a) shows the comparison results between the identification maps and CDL dataset in the US; (b) shows the comparison results between the identification maps and EuroCrops samples in Europe.**

## 3.4 Harvest time of global winter-triticeae crops

We finally calculated the harvest time of winter-triticeae crops in the study area in 2020 based on the time when the minimum NDVI occurred during the harvesting stage. Overall, the harvest time of winter-triticeae crops is delayed with increasing latitude (Fig. 14). In the Northern Hemisphere, winter-triticeae crops in East and South Asia were harvested in May and June (Fig. 14c), and the harvested area accounted for about 35.64% of the total harvested area in the study area (Fig. 15). The harvest time in Central Asia, Europe, North Africa and North America was concentrated between July and August (Fig. 14b, 14c 14d and 14f), and the proportion of harvested area to the total area was around 47.05% (Fig. 15). The regions with harvest time in September were mainly distributed in high latitude areas of Russia (Fig. 14b). In the Southern Hemisphere, the harvest time of winter-triticeae crops was mainly from November to January of the following year (Fig. 14e and 14g), with the harvested area accounting for 13.7% of the total harvested area (Fig. 15). These areas with the harvest time occurring from November to January were mainly located in high latitude regions of Australia and South America (Fig. 14e and 14g), and the harvest time in October only occurred in some areas of low latitude regions of South America (Fig. 14g).

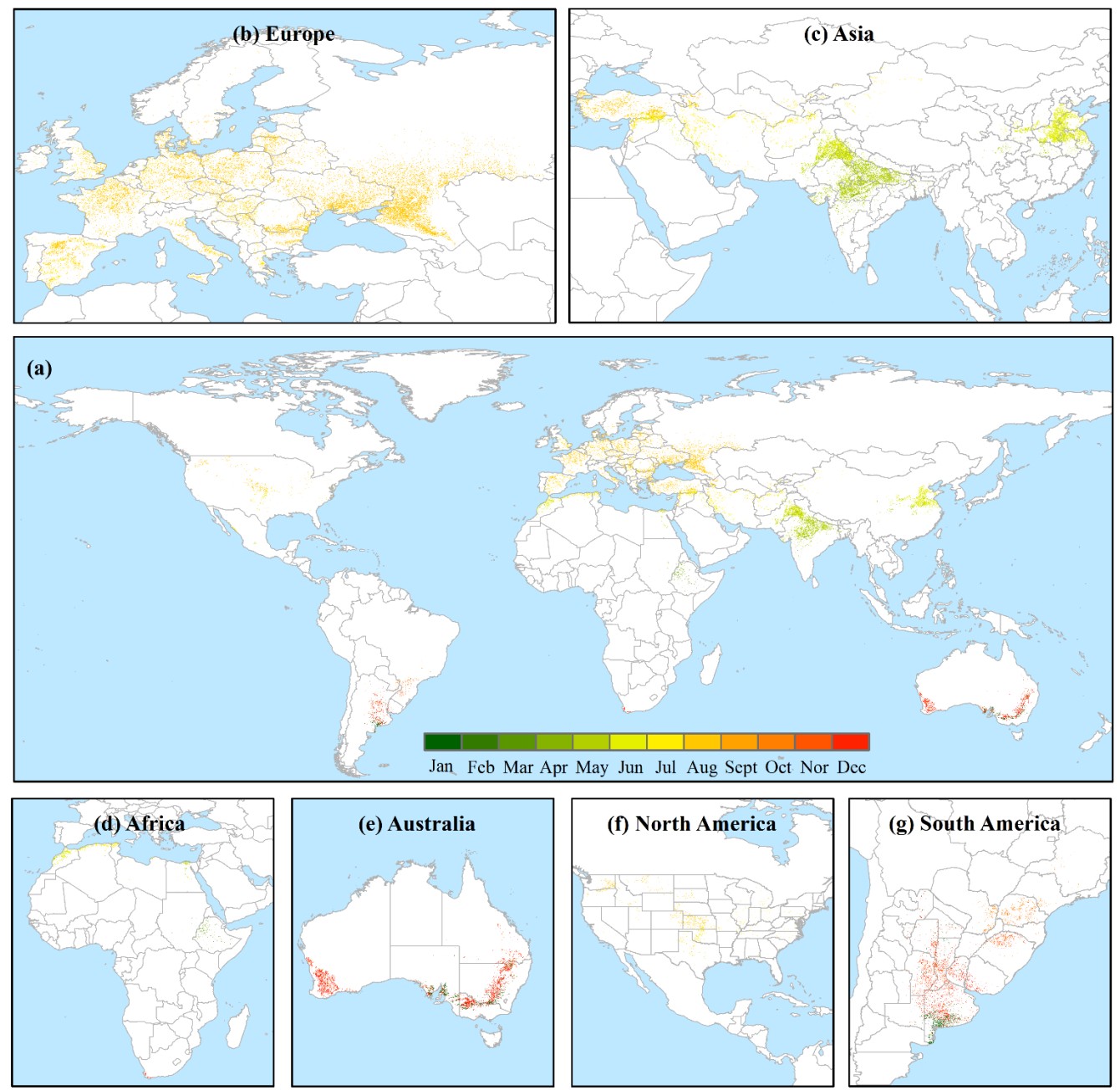

**Figure 14: Harvest time of winter-triticeae crops in the study area in 2020.**

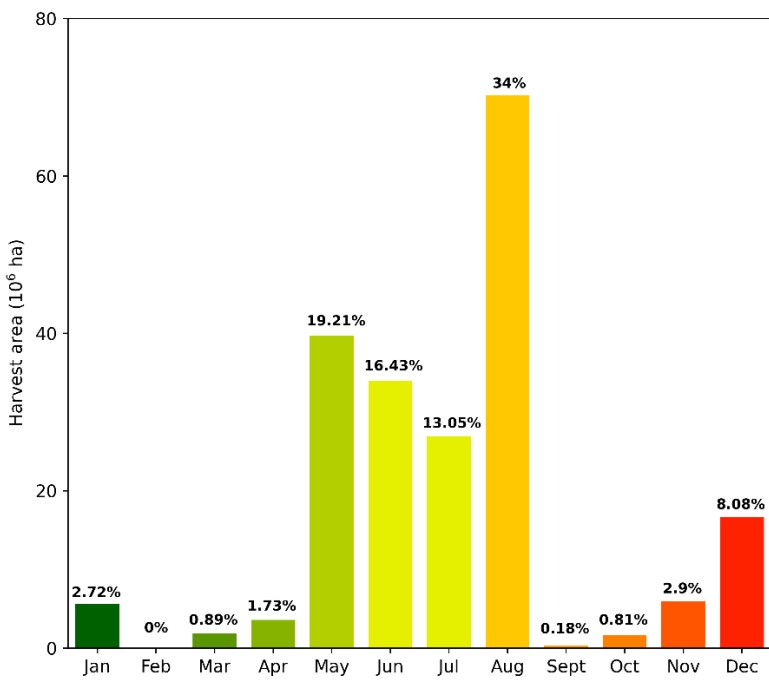

**Figure 15: Harvested area and proportion of winter-triticeae crops in the study area in 2020.**

## 4 Discussion

Winter-triticeae crops are among the most important grain crops in the world. Therefore, the ability to efficiently capture the distribution information about these crops is critical for monitoring crop growth and drafting grain subsidy policies (Liu et al., 410  2018). To our knowledge, there is currently a lack of a global distribution map for winter-triticeae crops at high resolution. Although there have been previous studies focusing on global triticeae crops mapping (Monfreda et al., 2008; Portmann et al., 2010; You et al., 2014), they resulted in maps for single or discontinuous years and with coarse spatial resolution, which may include large amounts of mixed pixels and have limited applications. For example, Lou et al. (2022) used inflection- and threshold-based methods to produce the global wheat map at a spatial resolution of 4 km, but the accuracy was low due to 415  mixed pixel problems in medium and small fields of South America. The available high-resolution maps of winter-triticeae crops with wide coverage can display more accurate information on planting location, such as the CDL in the US, winter wheat maps in China (Dong et al, 2020a), and winter cereals maps in Europe (Huang et al., 2022), but they are not currently available globally. In this study, we produced the first distribution maps of winter-triticeae crops with 30 m spatial resolution for 66 countries from 2017 to 2022 (2020 for US) based on the new WTCI method, filling the gap in the lack of global continuous 420  years and high-resolution winter-triticeae crops maps.

In addition, the method proposed in this study has the following advantages. First, $f(V)$ and $f(B)$ were incorporated in WTCI to alleviate errors and uncertainties in determining crop types based only on the part of features. Most previous studies only considered the differences between the maximum and minimum values of vegetation indices at key crop phenological stages (Atzberger et al., 2013; Chu et al., 2016; Manfron et al., 2017; Qiu et al., 2017). For example, Qu et al. (2021) set rules to determine the maximum and minimum NDVI before and after the over-wintering stage, respectively, and designed the winter wheat index (WWI) using the product of the differences between maximum and minimum NDVI. However, in some regions, the maximum NDVI values are not easy to determine before over-wintering, either due to the crop varieties or climate, resulting in very small differences between the maximum and minimum NDVI before over-wintering, which increases omission errors. Similar to this study, Xu et al. (2023) developed a spectral index for rice identification based on SAR data and tested the differences using partial features and three features. The results showed that considering three features simultaneously could better distinguish between rice and other crops, as well as other land cover types, and achieved the highest accuracy.

Second, all parameters of the WTCI are determined automatically. For example, based on the NDVI of each identification unit, the V and B lines are automatically generated to adapt to the differences in climate and land cover types between different regions, making the WTCI method more stable. This study selected two representative regions to test the sensitivity of identification accuracy to different percentile combinations of the V line ($v$) and the B line ($b$) (Fig. 16). The results demonstrate that the identification accuracy is insensitive to the percentiles of the V ($v$) and B lines ($b$) where winter-triticeae crops are the dominant crops (Fig. 16a). However, where winter-triticeae crops are not dominant, the identification accuracy is sensitive to the percentiles of the V ($v$) and B lines ($b$) (Fig. 16b). Overall, we achieved promising results in each identification unit, indicating that the WTCI method can be flexibly applied to different regions. Users can choose the appropriate percentile based on the local situation. Besides, the maximum and minimum NDVI values are automatically searched between the regreening and harvesting stages of winter-triticeae crops, avoiding the limitations caused by the use of a large number of constraints (Bazzi et al., 2019; Cai et al., 2019). Manfron et al. (2017) set multiple conditions based on expert knowledge to search for NDVI characteristics of key phenological stages to identify winter wheat. Although high identification accuracy was achieved in the study area, the application of the method was limited due to the proposed conditions in specific areas.

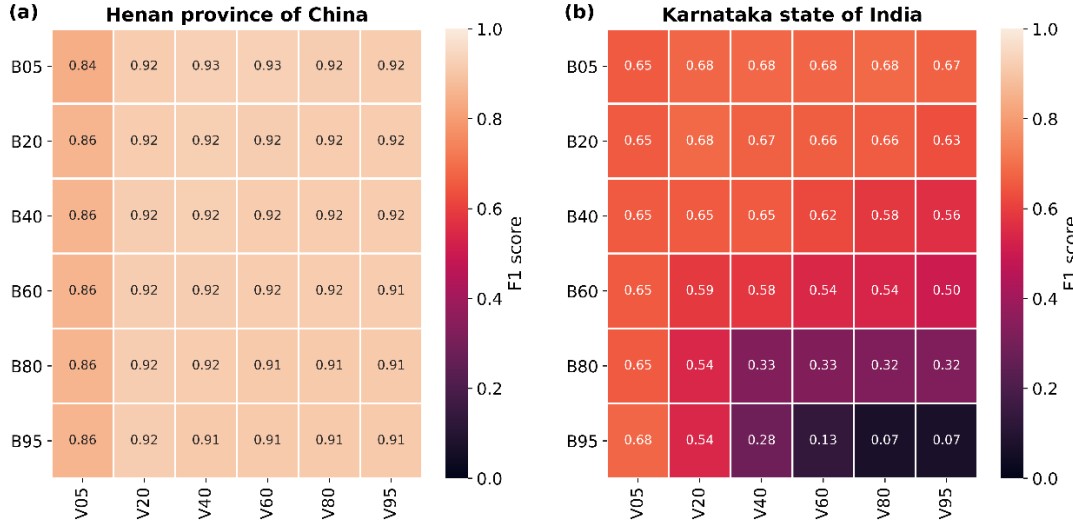

**Figure 16: Identification accuracy under different percentile combinations of V line (*v*) and B line (*b*). V05 and B05 represent the 5% percentile of the V and B lines, respectively.**

Finally, the WTCI method is not limited by samples and has strong transferability in time and space, making it suitable for mapping winter-triticeae crops in large regions. Supervised classification algorithms can extract information features from training samples and achieve high identification accuracy in specific years or regions (Brown and Pervez, 2014; Yin et al., 2020). However, the accuracy is often affected by insufficient training samples (Petitjean et al., 2012) or classification rules and regional limitations of parameters (Zhong et al., 2014) when the trained model is transferred to other years or regions, which makes it difficult to apply them on a large scale. The WTCI method does not require training samples and has achieved accurate results in most of the countries, with the OA values of 88.35% and 88.97% in China and Europe, respectively, which are comparable to the results of previous studies (Dong et al., 2020; Huang et al., 2022). Moreover, the satisfactory performance in capturing the field distribution of winter-triticeae crops in CDL and EuroCrops datasets supports the reliability and applicability of the WTCI method.

Despite the advantages, our study also suffers from some uncertainties. First, the commission error is higher in regions where winter-triticeae crops are not dominant crops, such as Sichuan (SC) province of China, West Bengal (WB), Bihar (BR), Karnataka (KA) and few countries in Mediterranean Sea region, indicating that here non-winter-triticeae crops are misclassified as winter-triticeae crops. One potential reason is the quantity and quality of the satellite data. Although we used synthetized images from Landsat and Sentinel productions to increase the amount of effective data and conducted linear interpolation and the Savitzky-Golay filter to further improve data quality, there are still differences in the quantity and quality of satellite data among the study area. A previous study highlighted that the availability of effective data greatly affected crop identification accuracy (Dong et al., 2015). Second, due to the scan line corrector failed of the Landsat 7 sensor, the striping

issues and reduced data availability may also impact the accuracy of NDVI time series (Ju and Roy., 2008), resulting in the errors in identification results. In our study, there were some striping issues in the distribution map of winter-triticeae crops in a few regions (Fig. S4a), which may lead to errors in winter-triticeae crops identification and the differences in identification results between different years (Fig. S4). Additionally, the wavelength difference between Sentinel-2 and Landsat sensors may affect the quality of synthesized NDVI. It is still a challenge to completely eliminate the impact from this difference (He et al., 2018). Besides, this study ignored the internal differences between winter wheat, winter barley, winter rye and triticale due to their similar NDVI time series and phenological characteristics (Huang et al., 2022; Xu et al., 2017), which may affect the identification accuracy. We referred to previous studies (Dong et al., 2020; Huang et al., 2022) on winter crop mapping and only distinguished winter rapeseed to reduce its impact on the identification of winter-triticeae crops. Other winter crops with smaller planting area that have not been discovered or overlooked may also interfere with the identification and lead to errors in the identification map. In the future, identifying useful bands or vegetation indexes that eliminate interferences from other land covers, further subdividing each winter-triticeae crop, as well as increasing the availability and quality of satellite data, will further promote the performance of the WTCI method.

## 5 Data availability

The 30 m resolution distribution maps of winter-triticeae crops in 66 countries worldwide from 2017 to 2022 (2020 for the US) are available at https://doi.org/10.57760/sciencedb.12361 (Fu et al., 2023a). The product is provided in GeoTIFF format with pixel values of 1 for winter-triticeae crops and 0 for other land covers.

## 6 Conclusions

This study proposed a new sample-free method (WTCI) for mapping winter-triticeae crops and examined its performance in 66 countries worldwide. The new method exhibits high accuracy and strong spatiotemporal transferability by comparing the produced maps with field survey samples and visual interpretation samples from Google Earth images, the CDL and EuroCrops datasets, and agricultural statistical data. Overall, the OA and F1 score were more than 80% and 75% in most of identification units, respectively. The $R^2$ between identified and statistical areas in most of regions was greater than 0.6 in all years, and RMAE less than 30%. These satisfactory results indicate that the WTCI method can be used for long-term and large-scale crop mapping. At the same time, the first 30 m spatial resolution distribution maps of winter-triticeae crops from 2017 to 2022 produced by the WTCI method fills the current product gaps, which can be further served for the harvest area monitoring, yield estimation and agricultural management.

**Author contributions.** WY and YF designed the research and developed the method. YF, XH, JD, and QP performed the investigation. YF wrote the manuscript, WY, XC, and CS revised the manuscript.

**Competing interests.** The contact author has declared that none of the authors has any competing interests.

**Disclaimer.** Publisher's note: Copernicus Publications remains neutral with regard to jurisdictional claims in published maps and institutional affiliations.

**Acknowledgements.** The authors would like to thank the editors and reviewers for their constructive comments to our manuscript.

**Financial support.** This work was supported by the Open Research Program of the International Research Center of Big Data for Sustainable Development Goals, Grant No. CBAS2023ORP02.

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
