# Peer review of "High-resolution mapping of global winter-triticeae crops using a sample-free identification method"

_Earth System Science Data, 2023_

## Author Response (AR1)

*Earth System Science Data*

Paper # essd-2023-432

Apr. 9, 2024

Dear editor and reviewers:

We are grateful to you for your constructive comments and suggested amendments on our manuscript entitled: "**High-resolution mapping of global winter-triticeae crops using a sample-free identification method**" (essd-2023-432). Your comments provide valuable insights for improving the contents and analysis. We have carefully studied the comments and revised our manuscript accordingly.

Here are our detailed responses to your comments. Please note that the comments from you are in **bold font** followed by our responses in regular font, changes/additions to the manuscript are underlined.

Sincerely yours,

Wenping Yuan on behalf of all co-authors

Corresponding author: Wenping Yuan, Ph.D., Professor

School of Atmospheric Sciences,

Sun Yat-sen University

135 West Xingang Road, Guangzhou 510275, China

E-mail address: yuanwp3@mail.sysu.edu.cn

**Detailed responses to reviewers' comments**

**Reviewer #1**

**This is needed and important research. However, some of the used methods are not clear enough, and in fact, the accuracy assessment may be not reliable. This makes the manuscript not appropriate for publication in ESSD in its current form.**

Response: Thanks for your comments. We deeply appreciate your time for reviewing the manuscript. Your suggestions are very useful for us to improve our manuscript. We have revised our manuscript according to your comments, and we also attached a point-by point letter to you. The detailed responses are listed below.

**My main concerns:**

**1). National datasets used for validation are not described at all, and I am not sure if they are reliable sources. The questions arise if these datasets are robust and/or detailed enough to perform accuracy assessment for the presented map? Did you only compare the area reported by country statistics and areas obtained in your maps? If so, that is not enough. Maybe, as a validation dataset it would be better to include USA CDL dataset.**

Response: Thanks for your suggestion. We have added a table (Table S1) to display the sources of agricultural statistical data for each country. This study used state (or province) scale statistical area to determine the WTCI thresholds for China, Brazil, India, Australia, and the United States, and evaluated the accuracy of each state (or province) using municipal or county scale statistical area. A state (or province) can contain dozens or hundreds of municipalities or counties. The national scale statistical area was used to determine the WTCI thresholds for other counties, and the statistical area of all states or provinces or municipalities or counties included in each country was used to evaluate accuracy. We hope that this comparison can be used to verify the spatial distribution of winter-triticeae crops map. We have added some details to describe the use of statistical area for accuracy assessment in the revised manuscript:

"In this study, we considered each state (or province) as an identification unit in China, Brazil, India, Australia and US, and the threshold of WTCI was determined based on statistical

area at state (or province) scale. For the remaining countries, we treated each country as an identification unit, and the threshold of WTCI was calculated relied on statistical area at national scale."

"At the regional scale, we obtained the identified areas of winter-triticeae crops based on the total pixel area of winter-triticeae crops on the identification maps. In China, Brazil, India, Australia and the US, we used the statistical area at municipal or county scale to validate the accuracy of identified area at state (or province) scale. For other counties, the statistical area of all states or provinces or municipalities or counties included in each country was used to evaluate the accuracy at national scale."

Additionally, we added validation data from USA CDL dataset and the Land Parcel Identification System (LPIS) dataset. The details are as follows:

"In addition, we used CDL and LPIS datasets to further evaluate the performance of WTCI method. The CDL released annually has high accuracy in capturing crop distribution in US and has been widely used as a base map for crop dynamic monitoring and production estimation. We thus treated CDL labels as ground truth and randomly selected 7,500 winter-triticeae crops samples and 12,500 non-winter-triticeae crops samples in 2020 to validate the accuracy of our method in US (Fig. 1). The LPIS dataset produced by European Union, accurately records and describes field geometry and landcover in EU countries. We thus collected and selected 10 countries with data clearly labelled with winter-triticeae crops, including winter spelt, winter barley, winter durum hard wheat, winter common soft wheat, winter triticale, winter rye and winter oats (https://zenodo.org/records/10118572). These data cover the period from 2018 to 2021, from which we randomly extracted 2,000 winter-triticeae crops samples and 3,000 non-winter-triticeae crops samples to assess the result of WTCI method in Europe (Fig. 1)."

The validation results of the WTCI method using CDL and LPIS datasets can be seen in 3.3 section in the revised manuscript.

**2). Another dataset for comparison/validation comes from Google Earth imagery. However, how is it possible to check or distinguish if there are winter crops indeed if for some years only single image is available, and may be not acquired during the time when**

**it is possible to assess?**

Response: We guess that the winter crops mentioned by the reviewer may refer to winter-triticeae crops in our study. The purpose of selecting samples based on Google Earth imagery is to assess the identification accuracy of WTCI method. Therefore, we select samples in the regions where there are available images to ensure the exactitude of the samples. At the same time, most of the planting areas of winter-triticeae crops are not perennial rainy areas, and the proportion of effective satellite observation is relatively high. Based on the actual situation of the selected samples, we can obtain a suitable number of samples in various winter-triticeae crops planting areas around the world. Here, we have added some content to explain how we selected samples from Google Earth imagery, the details are as follows:

"For other provinces in China and other countries (except US), we relied on high-resolution images from Google Earth from 2019 to 2020 for visual interpretation. We first chose regions with available images during the study period and selected samples from these regions based on the texture features. In order to ensure the accuracy of the samples, we then validated the selected samples on GEE platform by checking whether the NDVI temporal features of these samples matched the characteristics of winter-triticeae crops, and finally obtained 7,029 winter-triticeae crops samples and 8,897 non-winter-triticeae crops samples (Fig. 1)."

Previous studies (Yang et al., 2017; Zheng et al., 2022) have also adopted the approach that selecting samples from visual interpretation of high-resolution images when ground truth samples cannot be obtained. To increase the reliability of our methods and results, we further collected samples from CDL and LPIS datasets to validate the performance of our method. Detailed information can be seen in 2.2.2 and 3.3 sections in the revised manuscript, also can be found in response to Q1.

References:

Yang, D., Chen, J., Zhou, Y., Chen, X., Chen, X., Cao, X.: Mapping plastic greenhouse with medium spatial resolution satellite data: Development of a new spectral index, ISPRS J. Photogramm. Remote Sens., 128, 47–60, https://doi.org/10.1016/j.isprsjprs.2017.03.002, 2017.

Zheng, Y., dos Santos Luciano, A. C., Dong, J., Yuan, W. P.: High-resolution map of sugarcane

cultivation in Brazil using a phenology-based method, Earth Syst. Sci. Data., 14, 2065–2080, https://doi.org/10.5194/essd-14-2065-2022, 2022.

**3). The methodology is sometimes not clear. And what is also important, the data should be described firstly, before the methods used! For example, the methodology behind integration of Sentinel-2 and Landsat imagery is not clear. Do you used any harmonization techniques, which are needed in such combination between two satellite sources?**

Response: Thanks for your suggestion. We have moved the data section to the front of the method section, and we have added a detailed description of the integration of Sentinel-2 and Landsat imagery in the Data section of the revised manuscript:

"In this study, we used Landsat 7 collection 2 data and Landsat 8 collection 2 data, as well as Sentinel-2 data on the Google Earth Engine (GEE) platform to obtain NDVI from 2016 to 2022, all of which were surface reflectance (SR) products and have undergone atmospheric correction. The SR products of Landsat 7 and Landsat 8 have a spatial resolution of 30 m and a temporal resolution of 16 days. The spatial and temporal resolution of Sentinel-2 is 10 m and 5 days, respectively. To reduce the impact of clouds and ensure the quantity and quality of effective observation data, we first removed the pixels with clouds. The quality band BQA was used to remove pixels with clouds from Landsat 7 and Landsat 8, and the quality band QA60 was used to remove pixels contaminated by clouds from Sentinel-2. Then, based on nearest neighbour method, we resampled the NDVI of Sentinel-2 to 30 m to keep the same spatial resolution as Landsat data. Furthermore, we obtained NDVI of all cloud-free pixels, and chose the maximum values of monthly composites with 30 m spatial resolution, which has been proven effective for crop mapping and displaying crop growth stage (Huang et al., 2022). Last, we used linear interpolation and the Savitzky-Golay filter methods (Chen et al., 2004) to fill the missing values and smooth the NDVI series to reduce the contamination from cloud, rain and snow (Zheng et al., 2022). The above processes were run on the GEE platform."

We did not use harmonization techniques to combine Landsat and Sentinel data. There are differences in band wavelengths among different sensors of Sentinel-2 and Landsat, but the difference between NDVI calculated by Landsat and Sentinel products is small (Claverie et al.,

2018). Moreover, some studies (You and Dong., 2020; Dong et al., 2020) have successfully classified different crops using unharmonized vegetation index from Landsat and Sentinel products.

Here, we have also added some contents to discuss the difference in the Discussion section:

"Besides, the wavelength difference between Sentinel-2 and Landsat sensors may affect the quality of synthesized NDVI. It is still a challenge to completely eliminate the impact from this difference (He et al., 2018)."

References:

Claverie, M., Ju, J., Masek, J. G., Dungan, J. L., Vermote, E. F., Roger, J. C., Skakun, S. V., Justice, C.: The Harmonized Landsat and Sentinel-2 surface reflectance data set, Remote Sens. Environ., 219, 145–161, https://doi.org/10.1016/j.rse.2018.09.002, 2018.

He, M., Kimball, J. S., Maneta, M. P., Maxwell, B. D., Moreno, A., Beguería, S., Wu, X.: Regional crop gross primary productivity and yield estimation using fused landsat-MODIS data, Remote Sens., 10(3), 372, https://doi.org/10.3390/rs10030372, 2018.

You, N., Dong, J.: Examining earliest identifiable timing of crops using all available Sentinel 1 / 2 imagery and Google Earth Engine, ISPRS J. Photogramm. Remote Sens., 161, 109–123, https://doi.org/10.1016/j.isprsjprs.2020.01.001, 2020.

**4). Checking the dataset for my country shows that a large part is in fact located in the agricultural areas (however I cannot say if these are winter, not winter or not triticeae crops). However, there are also large parts located in the forests, and large areas with "stripes" probably related to not proper processing of Landsat 7 imagery. This should be for sure addressed in future, and methods should be refined. I also checked the area of the Mediterranean Sea, where many areas of maquis /shrublands were indicated as winter crops.**

Response: Thanks for your comments and deep thought. Overall, our sample-free WTCI method performs well in the main winter-triticeae crops planting regions, while the accuracy of the regions with complex crop planting types needs to be improved. The principle of the WTCI method is to use the NDVI characteristics of winter-triticeae crops from heading to harvesting

stages to distinguish other land covers. These stages occur in spring and summer, the other land covers are basically in the growing season with increased NDVI while the NDVI of winter-triticeae crops shows an obvious downward trend (Figure 2 in the revised manuscript). There should be significant differences and distinctions in theory. Therefore, we supposed that the poor performance might be related to the quality of data, which cannot effectively reflect the characteristics of land cover types, leading to misclassification of maquis/shrublands as winter-triticeae crops. We have discussed these questions in the Discussion section of the revised manuscript:

"Second, although we used synthetized images from Landsat and Sentinel productions to increase the amount of effective data, there are still large differences in the available images among the study area. A previous study highlighted that the availability of effective data greatly affected crop identification accuracy (Dong et al., 2015). In this study, the error between the identified area and statistical area of winter-triticeae crops was relatively high in the south of China and in some regions of India and South America, where the RMAE was greater than 35%. One potential reason for this is the quality of the satellite data. For example, cloud and rain contaminations introduce noise in the NDVI data and consequently dampen the winter-triticeae crops detection signal (Song et al., 2017; Xiao et al., 2014). Additionally, due to the scan line corrector failed of the Landsat 7 sensor, the striping issues and reduced data availability may also impact the accuracy of NDVI time series (Ju and Roy., 2008), leading the errors in identification results. Besides, the wavelength difference between Sentinel-2 and Landsat sensors may affect the quality of synthesized NDVI. It is still a challenge to completely eliminate the impact from this difference (He et al., 2018). In the future, identifying useful bands or vegetation indexes that eliminate interferences from other land covers, as well as increasing the availability and quality of satellite data, will further promote the performance of the WTCI method."

**Some other comments related to specific lines:**
**5). Line 28 – this sentence should be rephrased, mapping cannot monitor something**
Line 28: Crop mapping can monitor crop information by providing detailed location and nearreal time crop area (Skakun et al., 2017).

Response: Thank you for your suggestion. We have revised this sentence as follows:

"Crop mapping can provide detailed location and analyse spatiotemporal dynamics of crops (Skakun et al., 2017)."

**6). Line 58 – add information which satellite imagery did you use.**

Line 58: Here, this study developed the Winter-Triticeae Crops Index (WTCI), a sample-free method for identifying the global distribution of winter-triticeae crops.

Response: Thank you for your suggestion. We have added the used satellite information:

"Here, based on Landsat 7, Landsat 8 and Sentinel-2 satellite data, this study developed the Winter-Triticeae Crops Index (WTCI), a sample-free method for identifying the global distribution of winter-triticeae crops."

**7). Figure 1 – samples should have different, more distinguishable colours**

Response: Thank you for your advice. We have revised Figure 1:

[Figure]

**Figure 1: Distribution of the study area and validation samples. The study area is the region covered in green; The legend indicates the winter-triticeae (WT) crops samples and non-winter-triticeae (Non-WT) crops samples from Cropland Data Layer (CDL) dataset of the United States, the Land Parcel Identification System (LPIS) dataset of Europe, and field survey in China, as well as visual interpretation base on Google Earth images, respectively.**

**8). Line 80 – As mentioned above, data should be described first, before methodology.**

Response: Yes, we have moved the data section to the front of the method section.

**9). Line 88 – 91 – what about evergreen forests? They are not described or shown on Figure 2, while they are usually also characterized by high values during winter, for example. I think they should be taken into consideration when determining thresholds/methodology. Also, the vegetation in, for example, Mediterranean zones such as maquis may also be examined. Furthermore, what about the snow impact on the indices values?**

Line 88: There are significant differences in the temporal variations of NDVI among winter-triticeae crops, forest, and grassland.

Response: Thank you for your suggestion. First, we have modified Figure 2 and added NDVI time series of other land cover types (including evergreen forests) for comparison with winter-triticeae crops. This study determined the WTCI method based on the characteristic that the NDVI of winter-triticeae crops declines from heading to harvesting stage, which occurs between spring and summer. Although evergreen forests have high NDVI in winter, the NDVI values do not decrease between the heading and harvesting stages of winter-triticeae crops. Therefore, our method can accurately differentiate evergreen forest from winter-triticeae crops. Meanwhile, we have also made improvements to the Method section of the revised manuscript. The details are as follows:

"There are significant differences in the temporal variations of NDVI between winter-triticeae crops and natural vegetation types (i.e., deciduous forest, evergreen forest, and grassland) during the growing season of winter-triticeae crops (Fig. 2). Specifically, in the period from seedling to tillering stages, winter-triticeae crops are in a state of slow growth, with their NDVI gradually increasing. In contrast, natural vegetation types are in the deciduous stage, and exhibit a continuous decrease in NDVI during this period (Fig. 2). From the regreening to the heading stages, the NDVI of winter-triticeae crops rapidly increases and reaches its maximum value, while the NDVI increase of natural vegetation types tends to lag behind that

of winter-triticeae crops (Fig. 2). Furthermore, winter-triticeae crops show a downward trend and reach their lowest value during the harvesting stage. However, natural vegetations enter their growth season at this time, and their NDVI values rapidly increase (Fig. 2). Additionally, except for winter rapeseed, there are significant differences in the growth season of maize, rice, and soybean compared to that of winter-triticeae crops. Although the NDVI time series characteristics of these crops share similarities with winter-triticeae crops, they do not interfere with the identification of winter-triticeae crops."

[Figure]

**Figure 2: NDVI time series characteristics of different land cover types. The red five-pointed stars represent the different phenological stages of winter-triticeae crops.**

Second, thank you for your reminder, we did not consider the maquis in Mediterranean zones when analyzing the NDVI differences of different land cover types, which may lead to errors in the identification results. As mentioned in the Discussion section, we will work to solve the uncertainties of WTCI method and improve the identification accuracy in the future. In addition, we used the maximum values of monthly composites to obtain NDVI, and further employed linear interpolation and the Savitzky-Golay filter methods to fill the missing values and smooth the NDVI series to reduce the contaminations from cloud, rain and snow. Moreover, we used NDVI time series from spring to summer to identify winter-triticeae crops, therefore,

the impact of snow on indicator values is very small.

**10). Figure 2 – what about southern hemisphere?**

Response: Thank you for your reminder. We have modified and improved Figure 2, which can be found in response to Q9. There are seasonal differences between northern and southern hemispheres, but crops and land cover types are basically the same. Therefore, we used the phenological period of winter-triticeae crops to represent temporal variation to make the figure more concise.

**11). Lines 172-176 – use of SAR VH-derived thresholds is not clear.**

Lines 172-176: Therefore, the VH thresholds set by these studies were further employed to distinguish winter rapeseed and winter-triticeae crops. Specifically, in regions of Asia where winter rapeseed is planted, this study provides smaller WTCI values for pixels with VH values greater than -15.5 in March or April. In some European countries, pixels with VH values greater than -15.5 in May were assigned smaller WTCI values to reduce their probability of becoming winter-triticeae crops.

Response: We have clarified the content regarding the use of SAR VH-derived thresholds, and the details are as follows:

"Therefore, we distinguished winter rapeseed and winter-triticeae crops based on the methods of these studies, and the VH threshold set by Dong et al. (2020a), which was obtained by comparing filed samples, was employed in this study. Specifically, in regions of India where winter rapeseed is planted, we calculated the VH values from Sentinel-1 images in March considering the lower latitude and earlier harvest period of these regions. In other Asian regions where winter rapeseed is grown, this study obtained VH values for April. Then this study identified these pixels with VH values greater than -15.5 in March or April as non-winter-triticeae crops. Similarly, in some European countries, we calculated VH values for May, and considered that pixels with VH values greater than -15.5 were non-winter-triticeae crops (Huang et al., 2022)."

**12). Line 184 and further – what method for harmonizing the Sentinel-2 and Landsat data did you use? What collection from GEE were utilized? How did you remove pixels with clouds?**

Line 183-187: In this study, we obtained NDVI for 2016 – 2022 from reflectance data of Landsat 7, Landsat 8 and Sentinel-2 images on the Google Earth Engine (GEE) platform. To reduce the impact of clouds and ensure the quantity and quality of effective observation data, we first removed the pixels with clouds and acquired the maximum values of monthly composites with 30 m spatial resolution. Then, we used linear interpolation and the Savitzky-Golay filter methods (Chen et al., 2004) to fill the missing values and smooth the NDVI series (Zheng et al., 2022). The above processes were run on the GEE platform.

Response: We have added details information to reply these questions and the details can be found in response to Q3.

**13). Line 200 – how did you distinguish winter crops based on Google Earth imagery?**

Line 200: we relied on high-resolution images from Google Earth from 2019 to 2020 for visual interpretation and obtained 7,029 winter-triticeae crops samples and 8,897 non-winter-triticeae crops samples (orange triangles in Fig. 1).

Response: We have added some content to descript how we distinguish winter crops based on Google Earth imagery. The details can be found in 2.2.2 section in the revised manuscript, also can be seen in response to Q2.

**14). Equations 5-9 are redundant; they are commonly used and well-known.**

Response: Yes, we have deleted these equations.

**Reviewer #2**

**This manuscript deals with the important challenge of mapping winter triticeae crops at a global scale, a group of crops which is crucial for ensuring global food security. Conventional methods typically require a large number of reference samples to train supervised models that learn to map these crops. In many regions around the world, the availability of such samples is limited to non-existent. The proposed approach works independently from any reference data and therefore does not suffer from this drawback, providing an interesting alternative. The method is based on the temporal behavior of NDVI, where winter triticeae crops are said to be having unique characteristics that allows them to be mapped out against other crops or land cover.**

**While this is an attractive idea, the authors do not provide sufficient proof of the ability of such a simple approach to really result in high-quality maps at the global scale. There are several major shortcomings and lack of methodological details based on which I cannot recommend the manuscript for publication in its present form. Given its submission to ESSD, I would also expect more attention to the published data itself.**

Response: Thanks for your comments. We deeply appreciate your time for reviewing the manuscript. Your suggestions are very useful for us to improve our manuscript. We have revised our manuscript according to your comments, and we also attached a point-by point letter to you. The detailed responses are listed below.

**Major comments:**

**1). The main methodology is based on NDVI values of bare land vs. vegetation and the timing of these events. Looking at equations (2), (3) and (4), it seems that the only timing-related requirement is that the max NDVI should occur before the min NDVI. In Fig. 2 the winter triticeae temporal behavior is only compared to natural vegetation such as forest and grass. The most competing classes to map out from winter triticeae crops are of course other crop types! Why were these omitted from Fig. 2? How much of the reasoning still holds when compared to other crops? For example maize would be slightly**

**delayed wrt winter cereals in the Northern Hemisphere, but as far as I can tell from the provided equations, maize pixels would also have a high WTCI because their max NDVI occurs before the min NDVI and those values will be similar to vegetation and bare signals, respectively. What am I missing here?**

Response: Thanks for your comments. We have added NDVI time series of other land cover types in Figure 2 for comparison with winter-triticeae crops, and we have also made modifications and improvements to the corresponding content:

"There are significant differences in the temporal variations of NDVI between winter-triticeae crops and natural vegetation types (i.e., deciduous forest, evergreen forest, and grassland) during the growing season of winter-triticeae crops (Fig. 2). Specifically, in the period from seedling to tillering stages, winter-triticeae crops are in a state of slow growth, with their NDVI gradually increasing. In contrast, natural vegetation types are in the deciduous stage, and exhibit a continuous decrease in NDVI during this period (Fig. 2). From the regreening to the heading stages, the NDVI of winter-triticeae crops rapidly increases and reaches its maximum value, while the NDVI increase of natural vegetation types tends to lag behind that of winter-triticeae crops (Fig. 2). Furthermore, winter-triticeae crops show a downward trend and reach their lowest value during the harvesting stage. However, natural vegetations enter their growth season at this time, and their NDVI values rapidly increase (Fig. 2). Additionally, except for winter rapeseed, there are significant differences in the growth season of maize, rice, and soybean compared to that of winter-triticeae crops. Although the NDVI time series characteristics of these crops share similarities with winter-triticeae crops, they do not interfere with the identification of winter-triticeae crops."

[Figure]

**Figure 2: NDVI time series characteristics of different land cover types. The red five-pointed stars represent the different phenological stages of winter-triticeae crops.**

"The NDVI time series of winter rapeseed shows a downward trend from the heading to harvest stages of winter-triticeae crops, which is resemble winter-triticeae crops (Fig. 2). Tao et al. (2023) have also demonstrated that winter rapeseed and winter-triticeae crops have similar NDVI characteristics, making it difficult to distinguish them only based on optical images (Veloso et al., 2017). Fortunately, previous studies have indicated that the VH (vertical transmit and horizontal receive) band can effectively eliminate the interference from winter rapeseed in the identification of winter-triticeae crops in China and Europe (Dong et al., 2020a; Huang et al., 2022). Therefore, we distinguished winter rapeseed and winter-triticeae crops based on the methods of these studies, and the VH threshold set by Dong et al. (2020a), which was obtained by comparing filed samples, was employed in this study. Specifically, in regions of India where winter rapeseed is planted, we calculated the VH values from Sentinel-1 images in March considering the lower latitude and earlier harvest period of these regions. In other Asian regions where winter rapeseed is grown, this study obtained VH values for April. Then this study identified these pixels with VH values greater than -15.5 in March or April as non-winter-triticeae crops. Similarly, in some European countries, we calculated VH values for May, and

considered that pixels with VH values greater than -15.5 were non-winter-triticeae crops (Huang et al., 2022)."

In addition, it should be noted that although there are similarities in the NDVI time series characteristics between corn and winter-triticeae crops, the growing seasons of these crops do not overlap. We mentioned in section 2.2.1 of the original manuscript that our method is based on the time range from heading to harvesting stages of winter-triticeae crops.

Reference:

Tao, J. B., Zhang X. Y., Wu, Q. F., Wang, Y.: Mapping winter rapeseed in South China using Sentinel-2 data based on a novel separability index, J Integr Agric., 22(6), 1645-1657, https://doi.org/10.1016/j.jia.2022.10.008, 2023.

**2). The method relies on max NDVI occurring before min NDVI. But how do you decide on the reference period for which to analyse the curve? This should be different for northern and southern hemisphere at least. This reference period is a crucial choice for the outcome of the method.**

Response: Thank you for your reminder. We added some contents to clarify these questions, and the details are as follows:

"Specifically, this study referred to crop calendar data provided by the United States Department Agriculture (USDA) (https://ipad.fas.usda.gov/ogamaps/cropcalendar.aspx) to determine the growth season of winter-triticeae crops in each country. Then, we extracted the maximum and minimum NDVI of all potentially identified pixels during the growing season of winter-triticeae crops. Meanwhile, different percentiles (5%, 20%, 40%, 60%, 80%, and 95%) of all maximum and minimum NDVI were collected, and the values corresponding to the percentile of the maximum and minimum NDVI were chosen as $v$ and $b$, respectively. After the above steps, we conducted winter-triticeae crops identification for all countries in 2020 based on the calculated WTCI, with each identification unit having its corresponding V line ($v$) and B line ($b$). When calculating WTCI, we searched for the maximum and minimum NDVI values between the regreening and harvesting stages of winter-triticeae crops. In this study, the regreening stage was based on the start time of spring in the northern (March) and southern

(September) hemispheres (Ren et al., 2019), and the harvesting stage referred to the crop calendar provided by USDA. We first determined the maximum NDVI value and its occurrence time of each potentially identified pixel, then looked for the minimum NDVI value in the period after the maximum NDVI appears, and further calculated WTCI. Pixels that do not meet this condition are identified as non-winter-triticeae crops."

Reference:

Ren, S. L., Qin, Q. M., Ren, H. Z.: Contrasting wheat phenological responses to climate change in global scale, Sci Total Environ., 665, 620–631, https://doi.org/10.1016/j.scitotenv.2019.01.394, 2019.

**3). This study uses agricultural statistical data to determine the threshold of WTCI where statistical data is available. This way, mapping is tuned towards matching these statistical numbers. In the validation results, comparisons are made between the resulting maps and the same statistical data, where correlation coefficients are reported between mapped area and the reported area. This is not an independent analysis and if thresholds were tuned to match statistical numbers, high correlation coefficients with these same numbers seem obvious. In addition, how are planted areas computed from the resulting maps? Area estimates from maps have to be done carefully to avoid biased estimates. This is not discussed here.**

Response: Thanks for your suggestion. First, the statistical data used to determine the threshold and the statistical data used for accuracy validation are independent of each other. Specifically, this study used state (or province) scale statistical area to determine the WTCI thresholds for China, Brazil, India, Australia, and the United States, and evaluated the accuracy of each state (or province) using municipal or county scale statistical area. A state (or province) can contain dozens or hundreds of municipalities or counties. The national scale statistical area was used to determine the WTCI thresholds for other counties, and the statistical area of all states or provinces or municipalities or counties included in each country was used to evaluate accuracy. We hope that this comparison can be used to verify the spatial distribution of winter-triticeae crop map. Second, given that the spatial resolution of each pixel is 30 m ×30 m under the

projection of Albers Equal Area Conic, we calculated the sum of the pixel area of winter-triticeae crops on the resulting map to obtain the identified area. We have added some details in the revised manuscript:

"In this study, we considered each state (or province) as an identification unit in China, Brazil, India, Australia and US, and the threshold of WTCI was determined based on statistical area at state (or province) scale. For the remaining countries, we treated each country as an identification unit, and the threshold of WTCI was calculated relied on statistical area at national scale."

"At the regional scale, we obtained the identified areas of winter-triticeae crops based on the total pixel area of winter-triticeae crops on the identification maps. In China, Brazil, India, Australia and the US, we used the statistical area at municipal or county scale to validate the accuracy of identified area at state (or province) scale. For other counties, the statistical area of all states or provinces or municipalities or counties included in each country was used to evaluate the accuracy at national scale."

In addition to verifying the identified area at regional scale through agricultural statistical area, we also used Google Earth samples for accuracy evaluation at the pixel scale. Besides, we have added the Cropland Data Layer (CDL) dataset and Land Parcel Identification System (LPIS) data to perform independent validation for our results. The details are as follows:

"In addition, we used CDL and LPIS datasets to further evaluate the performance of WTCI method. The CDL released annually has high accuracy in capturing crop distribution in US and has been widely used as a base map for crop dynamic monitoring and production estimation. We thus treated CDL labels as ground truth and randomly selected 7,500 winter-triticeae crops samples and 12,500 non-winter-triticeae crops samples in 2020 to validate the accuracy of our method in US (Fig. 1). The LPIS dataset produced by European Union, accurately records and describes field geometry and landcover in EU countries. We thus collected and selected 10 countries with data clearly labelled with winter-triticeae crops, including winter spelt, winter barley, winter durum hard wheat, winter common soft wheat, winter triticale, winter rye and winter oats (https://zenodo.org/records/10118572). These data cover the period from 2018 to 2021, from which we randomly extracted 2,000 winter-triticeae crops samples and 3,000 nonwinter-triticeae crops samples to assess the result of WTCI method in Europe (Fig. 1)."

The validation results of the WTCI method using CDL and LPIS datasets can be seen in 3.3 section in the revised manuscript or in the response to Q4.

**4). USA is excluded from the analysis because "highly accurate and annually CDL" is already available. This seems odd. A study aiming for global mapping should include USA for completeness and consistency of the maps as well. In fact, USA could be excellent to compare your results to the CDL and report agreement and differences. Also with respect to proving your method is not triggered by other crops than winter triticeae.**

Line 69-70: However, due to its high accuracy and annually updated CDL, the study area did not include the United States.

Response: Thank you for your suggestion. We have deleted this sentence and added the US as our study area, and further validated our results using CDL dataset. The details are as follows:

"**3.3 The performance of the WTCI method validated using CDL and LPIS datasets**

Based on CDL and LPIS datasets, we further validated the performance of the WTCI method in the US and Europe. In 2020, the OA and F1 score in the US were 86.84% and 0.82, respectively, and the PA and UA were 76.96% and 88.13%, respectively (Fig. 10). The performance of the WTCI method varied by state. For all states planting winter-triticeae crops, the OA varied from 70.42% to 94.24%, and the F1 score ranged from 0.67 to 0.91 (Fig. 10a-10c). In major planting states, such as Kansas, Oklahoma and Texas, the planting area of winter-triticeae crops account for approximately 50% of the total area of winter-triticeae crops in the US, displaying high accuracy with OA and F1 score over 92% and 0.85, respectively (Fig. 10a). The identified area of WTCI method exhibited good consistency with official statistical data. At national scale, the $R^2$ and RMAE were 0.89 and 28.9%, respectively (Fig. 11a). At state scale, the $R^2$ varied between 0.52 to 1, and the RMAE was in 9.01%-57.84% (Fig. 11b-11w). In Europe, the PA, UA and OA in 10 countries were 73.18%, 86% and 83.88%, respectively, and the F1 score was 0.79 (Fig. 10d). The PA and UA ranged from 63.68% to 84.77%, and 71.43% to 96.24% over the various countries, respectively. The OA and F1 score varied from 71.22% to 94.79% and 0.68 to 0.9, respectively (Fig. 10d). In general, the OA and F1 score in most of

regions of US and Europe were higher than 80% and 0.75, implying that the WTCI method exhibited satisfactory results compared to the CDL and LPIS datasets. Additionally, we presented spatial detail information of the identification map produced by the WTCI method in US and Europe for comparison with CDL and LPIS datasets (Fig. 12). The results indicate that the identification map and can effectively capture the field distribution of winter-triticeae crops in the US and Europe (Fig. 12)."

[Figure]

**Figure 10: The producer's accuracy (PA), user's accuracy (UA), overall accuracy (OA) and F1 score of the identification maps of winter-triticeae crops in the US and Europe. The abbreviations of countries and states are shown in Table S2 and S3 in the supplement. 2018FRA indicates the identification accuracy of the country in 2018.**

[Figure]

**Figure 11: Comparison between identified and statistical areas of winter-triticeae crops in 2020 in the US. (a) show the results between identified and statistical areas at national scale; (b-w) show the results between identified and statistical areas for each state, respectively. The green symbols represent the counties of each state. The yellow solid lines are the regression lines, and the grey short-dashed lines are the 1:1 lines. The red, blue and purple numbers represent R², slope and RMAE values between identified and statistical areas, respectively.**

[Figure]

**Figure 12: Comparison of the identification maps of winter-triticeae crops with CDL and LPIS datasets. (a) shows the comparison results between the identification maps and CDL dataset in the US; (b) shows the comparison results between the identification maps and LPIS samples in Europe.**

5). The study discusses (also in the title) global mapping, while the study area actually contains just 65 countries (Fig. 1). It is stated that 99% of the global winter triticeae crops are covered referring to FAO 2020 which does not appear in the reference list. How did the authors determine the 99% in the first place? Russia is not included while it grows a

**major part of the global wheat production. I would recommend in any case to be more careful with the "global" terminology.**

Response: Thank you for your suggestion. We have listed the official website and reference of the data sources, and revised some content:

"The study area covers 66 countries, including 36 European countries, 15 Asian countries, 8 African countries, 2 North American country, 4 South American countries, and 1 Oceania country (Fig.1). The area of global triticeae crops (including spring and winter varieties) is 278.87 million ha in 2020 (https://www.fao.org/faostat/en/#data), with winter-triticeae crops accounting for about 75% (i.e., 209.15 million ha) of the global triticeae crops area (Zhao et al., 2018). According to the statistics of the winter-triticeae area provided on official websites of various countries (Table S1), the total area of winter-triticeae in our study area in 2020 is 207,45 million ha, occupying 99.19% of the global winter-triticeae crops area."

In addition, thank you for your reminder about the terminology of global. In order to accurately show the scope of our study area, we only display the planting regions of winter-triticeae crops in Russia, China, India and Brazil rather than the whole country in Figure 1 in the original manuscript. Besides, we have also added US as our study area. In the revised manuscript, the winter-triticeae crops area in our study area accounts for 99.19 % of the global winter-triticeae crops area in 2020. Thus, we used global in the manuscript.

Reference:

Zhao, G. C., Chang, X. H., Wang, D. M., Tao, Z. Q., Wang, Y. J., Yang, Y. S., Zhu, Y. J.: General Situation and Development of Wheat Production, Crops., 4, 1-7, doi: 10.16035/j.issn.1001-7283.2018.04.001, 2018.

**6). The validation section is insufficient. A field survey was conducted in China but for all other countries, visual interpretation of Google Earth images was performed. How can the latter be done reliably? How do you identify winter triticeae from a Google Earth picture? Why can't this be another crop? In regions such as Europe, many LPIS datasets are freely available, providing an excellent validation resource for the maps. I strongly suggest the authors to compare to such data instead of interpreting Google Earth pictures,**

**especially where such high quality data is available. This is crucial to prove that the method doesn't detect other crops.**

Response: Thank you for your suggestion. We have added some content to explain how we selected samples from Google Earth imagery, the details are as follows:

"For other provinces in China and other countries (except US), we relied on high-resolution images from Google Earth from 2019 to 2020 for visual interpretation. We first chose regions with available images during the study period and selected samples from these regions based on the texture features. In order to ensure the accuracy of the samples, we then validated the selected samples on GEE platform by checking whether the NDVI temporal features of these samples matched the characteristics of winter-triticeae crops, and finally obtained 7,029 winter-triticeae crops samples and 8,897 non-winter-triticeae crops samples (Fig. 1)."

Previous studies (Yang et al., 2017; Zheng et al., 2022) have also adopted the approach that selecting samples from visual interpretation of high-resolution images when ground truth samples cannot be obtained. To increase the reliability of our methods and results, we further collected samples from CDL and LPIS datasets to validate the performance of our method. Detailed information can be seen in 2.2.2 and 3.3 sections in the revised manuscript, also can be found in responses to Q3 and Q4.

References:

Yang, D., Chen, J., Zhou, Y., Chen, X., Chen, X., Cao, X.: Mapping plastic greenhouse with medium spatial resolution satellite data: Development of a new spectral index, ISPRS J. Photogramm. Remote Sens., 128, 47–60, https://doi.org/10.1016/j.isprsjprs.2017.03.002, 2017.

Zheng, Y., dos Santos Luciano, A. C., Dong, J., Yuan, W. P.: High-resolution map of sugarcane cultivation in Brazil using a phenology-based method, Earth Syst. Sci. Data., 14, 2065–2080, https://doi.org/10.5194/essd-14-2065-2022, 2022.

**7). Satellite data is hardly described. Why is Landsat 7 still part of the analysis knowing its striping issues and knowing that for the temporal range of the study it's not essential? How were reflectance data from the different sensors harmonized? Is this an existing**

**collection? If not, more detail is required here. Based on what were clouds masked for the different sensors?**

Response: Thank you for your advices. First, in order to obtain more observation data, we used Landsat 7 satellite and performed linear interpolation and the Savitzky-Golay filter to improve data quality. However, there are still inevitable striping issues in a few regions, which affect our identification results. This is also the part that we need to make improvements and enhancements in our future work, and we have discussed this issue in the discussion section of the revised manuscript:

"Additionally, due to the scan line corrector failed of the Landsat 7 sensor, the striping issues and reduced data availability may also impact the accuracy of NDVI time series (Ju and Roy., 2008), leading the errors in identification results."

Second, we have added some details to display the process of integrating Landsat 7, Landsat 8 and Sentinel-2 data in the Data section:

"In this study, we used Landsat 7 collection 2 data and Landsat 8 collection 2 data, as well as Sentinel-2 data on the Google Earth Engine (GEE) platform to obtain NDVI from 2016 to 2022, all of which were surface reflectance (SR) products and have undergone atmospheric correction. The SR products of Landsat 7 and Landsat 8 have a spatial resolution of 30 m and a temporal resolution of 16 days. The spatial and temporal resolution of Sentinel-2 is 10 m and 5 days, respectively. To reduce the impact of clouds and ensure the quantity and quality of effective observation data, we first removed the pixels with clouds. The quality band BQA was used to remove pixels with clouds from Landsat 7 and Landsat 8, and the quality band QA60 was used to remove pixels contaminated by clouds from Sentinel-2. Then, based on nearest neighbour method, we resampled the NDVI of Sentinel-2 to 30 m to keep the same spatial resolution as Landsat data. Furthermore, we obtained NDVI of all cloud-free pixels, and chose the maximum values of monthly composites with 30 m spatial resolution, which has been proven effective for crop mapping and displaying crop growth stage (Huang et al., 2022). Last, we used linear interpolation and the Savitzky-Golay filter methods (Chen et al., 2004) to fill the missing values and smooth the NDVI series to reduce the contamination from cloud, rain and snow (Zheng et al., 2022). The above processes were run on the GEE platform."

We did not use harmonization techniques to combine Landsat and Sentinel data. There are differences in band wavelengths among different sensors of Sentinel-2 and Landsat, but the difference between NDVI calculated by Landsat and Sentinel products is small (Claverie et al., 2018). Moreover, some studies (You and Dong., 2020; Dong et al., 2020) have successfully classified different crops using unharmonized vegetation index from Landsat and Sentinel products.

Here, we have also added some contents to discuss the difference in the Discussion section:

"Besides, the wavelength difference between Sentinel-2 and Landsat sensors may affect the quality of synthesized NDVI. It is still a challenge to completely eliminate the impact from this difference (He et al., 2018)."

References:

Claverie, M., Ju, J., Masek, J. G., Dungan, J. L., Vermote, E. F., Roger, J. C., Skakun, S. V., Justice, C.: The Harmonized Landsat and Sentinel-2 surface reflectance data set, Remote Sens. Environ., 219, 145–161, https://doi.org/10.1016/j.rse.2018.09.002, 2018.

He, M., Kimball, J. S., Maneta, M. P., Maxwell, B. D., Moreno, A., Beguería, S., Wu, X.: Regional crop gross primary productivity and yield estimation using fused landsat-MODIS data, Remote Sens., 10(3), 372, https://doi.org/10.3390/rs10030372, 2018.

Ju, J. C., Roy, D. P.: The availability of cloud-free Landsat ETM+ data over the conterminous United States and globally, Remote Sens. Environ., 112(3), 1196-1211, https://doi.org/10.1016/j.rse.2007.08.011, 2008.

You, N., Dong, J.: Examining earliest identifiable timing of crops using all available Sentinel 1 / 2 imagery and Google Earth Engine, ISPRS J. Photogramm. Remote Sens., 161, 109–123, https://doi.org/10.1016/j.isprsjprs.2020.01.001, 2020.

**8). The section on Data should really come before the explanation on the methodology**

Response: Yes, we have moved the data section to the front of the method section.

**Data comments:**

**9). GeoTIF files are called e.g. F*rance_classify_2021_WTCI_Bline20_Vline05* or**

***Belgium_classify_2021_WTCI_Bline60_Vline80.* This naming convention needs to be explained. Two neighboring countries where *Vline* value changes from *05* to *80*? What are these values and why do they differ so much?**

Response: Thank you for your suggestion. We have modified the file name in our dataset. In order to make the WTCI method more stable in different regions, we use an automatic method to determine the optimal identification result through different percentile combinations of V and B lines in each identification unit. The details can be found in section 2.3.3 of the revised manuscript. Here, The NDVI value corresponding to Vline05 in France is 0.62, which is the *v* value in the WTCI method, and the NDVI value corresponding to Vline80 in Belgium is 0.9. We discussed the potential reasons for the differences in percentile combinations of V and B lines in different regions in the Discussion section of the original manuscript. We also display the differences of F1 score in different percentile combinations of V and B lines in France and Belgium (Fig. R1). The results indicate that the different percentile combinations of V and B lines have little impact on F1 score in France where winter-triticeae is dominant crop (Fig. R1a), while in Belgium with less winter-triticeae crops planting area, there is a significant difference in F1 score under different percentile combinations of V and B lines (Fig. R1b). We consider this is one of the potential reasons for the differences in percentile combinations of V and B lines among different regions. This conclusion is consistent with the study of Xu et al. (2023), who analysed the reasons for the differences in rice identification accuracy in different regions based on similar method. We will continue to explore more reasons of this phenomenon in future work.

[Figure]

**Figure R1: Identification accuracy under different percentile combinations of V line (v) and B line (b). V05 and B05 represent the 5% percentile of the V and B lines, respectively.**

**10). For the files I checked I was unable to get a correct georeferencing, cfr. this screenshot. More checks on the data format are required so a user can actually make use of them.**

[Figure]

Response: We have checked these GeoTIFF files and found that the reason for the above situation is a problem with spatial projection. After converting it to GCS 1984 (EPSG:4326),

we obtained the correct display in the following picture. In addition, we have added a documentation with the dataset to introduce the spatial projection, so that users can use the data correctly.

[Figure]

**11). Even within single files, strange spatial artefacts appear, like this one in Northern France. Where does this come from?**

[Figure]

Response: This reason is the same as question 10. The following picture shows the enlarged details in the same area after converting the GeoTIFF file to GCS 1984 (EPSG:4326).

[Figure]

**Other comments:**

**12). L28: no area estimates follow naturally from crop maps. Suggest to use another term than "crop area"**

Line28: Crop mapping can monitor crop information by providing detailed location and nearreal time crop area (Skakun et al., 2017).

Response: Thank you for your suggestion. We have revised this sentence as follows:

"Crop mapping can provide detailed location and analyse spatiotemporal dynamics of crops (Skakun et al., 2017)."

**13). L33: Rephrase "warning global food security"**

Line33: Closely monitoring the spatial distribution information of winter-triticeae crops is therefore beneficial for evaluating yield, optimizing land use, and warning global food security (Fu et al., 2021; Nelson and Burchfield, 2021; Wardlow et al., 2007).

Response: Thanks, we have revised this sentence as follows:

"Closely monitoring the spatial distribution information of winter-triticeae crops is therefore beneficial for evaluating yield, optimizing land use, and assessing food security (Fu et al., 2021; Nelson and Burchfield, 2021; Wardlow et al., 2007)."

**14). L35-36: Please have a look at the recently released WorldCereal global 10m maps of winter triticeae. Some of the statements in the manuscript are outdated. Please revise the introduction accordingly.**

Line 35-36: Few studies have attempted to produce global triticeae crop maps (You et al., 2014), but efforts have been limited to coarse resolutions.

Response: Thank you for your reminder. We have added some new contents in the Introduction section of the revised manuscript, and the details are as follows:

"The WorldCereal project proposed by European Space Agency (ESA) has released a global crop maps with a spatial resolution of 10 m for 2021, addressing the limitations of spatial resolution in global-scale crop mapping (Van Tricht., 2023). However, this product is currently only available for one year, which will affect the demand for long time-scale."

Reference:

Van Tricht, K., Degerickx, J., Gilliams, S., Zanaga, D., Battude, M., Grosu, A., Brombacher, J., Lesiv, M., Bayas, J. C. L., Karanam, S., Fritz, S., Becker-Reshef, I., Franch, B., Mollà-Bononad, B., Boogaard, H., Pratihast, A. K., Koetz, B., and Szantoi, Z.: WorldCereal: a

dynamic open-source system for global-scale, seasonal, and reproducible crop and irrigation mapping, Earth Syst. Sci. Data, 15, 5491–5515, https://doi.org/10.5194/essd-15-5491-2023, 2023.

**15). L67: see main comments: why these 65 countries. It's a bit strange that because CDL is available in US, this country is not mapped. This would actually be an excellent reason to do it nonetheless and compare your results with the CDL.**

Line 67: The study area covers 65 countries, including 36 European countries, 15 Asian countries, 8 African countries, 1 North American country, 4 South American countries, and 1 Oceania country (Fig.1).

Response: Thank you for your comments. We have added the US as our study area, and made some modifications to 2.1 section of the revised manuscript. In addition, we validated and compared the results of the proposed method using the CDL dataset. Detailed information can be seen in 2.1, 2.2.2 and 3.3 sections in the revised manuscript, also can be found in responses to Q3, Q4 and Q5.

**16). L72: see main comments. Please provide more information on where this number comes from.**

Line 72: with the winter-triticeae crops area occupying about 99% of the global (excluding the United States) winter-triticeae crops area (FAO, 2020).

Response: Yes, we have revised some contents and listed the source of this number, and the details can be found in response to Q5.

**17). L91: what with evergreen trees? And grassland not necessarily exhibits this continuous decrease in NDVI.**

Line 91: In contrast, natural vegetation types (i.e., forest and grassland) are in the deciduous stage, and exhibit a continuous decrease in NDVI during this period (Fig. 2).

Response: Thank you for your suggestion. We have added land cover types (include evergreen trees) to analyze the characteristics of NDVI time series, and the details can be found in

response to Q1. Xu et al. (2023) have demonstrated that in mid-latitude, where is the main distribution areas of winter-triticeae crops worldwide (Zhao et al., 2018), the NDVI time series of grassland in most areas showed a decrease trend from autumn to winter. Most important, the WTCI method is depends on the NDVI characteristics of land cover types in spring and summer. During this period, the variation trend of NDVI of grassland and winter-triticeae is opposite (Figure 2 in the revised manuscript), which will not affect the identification of winter-triticeae crops.

References:

Xu, X. F., Tang, J. K., Zhang, N., Zhang, A. A., Wang, W. H., Sun, Q.: Remote Sensing Classification of Temperate Grassland in Eurasia Based on Normalized Difference Vegetation Index (NDVI) Time-Series Data, Sustain., 15, 14973, https://doi.org/10.3390/su152014973, 2023.

**18). L120: See main comments. Max before min with respect to what? How do you enforce this? How do you treat calendar years? A maximum will always be before some minimum later on. What's your reference period in which you make this analysis?**

Line 120: the maximum NDVI should appear before the minimum NDVI.

Response: Thank you for your suggestion. The design concept of the WTCI method is to use the NDVI time series characteristics of winter-triticeae crops from the heading to harvesting stages to distinguish other land cover types. This stage occurs in the spring and summer of a year and the NDVI of winter-triticeae crops reaches its maximum during the heading stage of winter-triticeae crops, then gradually decreases until harvest (Figure 2 in the revised manuscript). Thus, we could use the condition that whether the maximum NDVI occur before the minimum NDVI during the stage from the heading to harvesting in a year. More details analysis can be found in response to Q1 or 2.3.1 section in the revised manuscript. Second, the process of winter-triticeae crops identification and calendar years or reference period used in this study can be found in response to Q2.

**19). L119: how is this done in practice for the southern hemisphere, i.e. how do you take**

**into account cyclical dates?**

Line 119: It should be noticed that Eq. (1) was used to identify the winter- triticeae crops only when *n1<n2*,

$$WTCI = f(D) \times f(V) \times f(B), n1 < n2 \quad, \tag{1}$$

Response: We referred to crop calendar data provided by the United States Department Agriculture (https://ipad.fas.usda.gov/ogamaps/cropcalendar.aspx) to determine the growth season of winter-triticeae crops in southern hemisphere. The details can be found in response to Q2.

**20). L157: which accuracy is meant here based on which to decide the optimal threshold?**

L157: the optimal combination was decided solely based on the accuracy at the pixel scale.

Response: Thank you for your reminder. We have modified this sentence:

"In addition, we determined the optimal combination of V and B lines according to the identification accuracy at the pixel scale (F1 score) and the relative mean absolute error (RMAE) between identified and agricultural statistical areas. For countries lacking agricultural statistical data, the optimal combination was decided solely based on the F1 score."

**21). L160: why only these two crops?**

L160: The identification of winter-triticeae crops in the study area may be affected by winter rapeseed and garlic.

Response: Because winter rapeseed and garlic have similar growth season and spectral characteristics to winter-triticeae crops. We have revised this sentence:

"The identification of winter-triticeae crops in the study area may be affected by winter rapeseed and garlic, as these crops have similar growth season and spectral characteristics with winter-triticeae crops (Fu et al., 2023b; Tian et al., 2021)."

Reference:

Tian, H. F., Wang, Y. J., Chen, T., Zhang, L. J., Qin, Y. C.: Early-Season Mapping of Winter Crops Using Sentinel-2 Optical Imagery, Remote Sens., 13, 3822, https://doi.org/10.3390/rs13193822, 2021.

**22). L172-176: this lacks detail. The tresholds seem arbitrary and are computed on what exactly?**

L172-176: Therefore, the VH thresholds set by these studies were further employed to distinguish winter rapeseed and winter-triticeae crops. Specifically, in regions of Asia where winter rapeseed is planted, this study provides smaller WTCI values for pixels with VH values greater than -15.5 in March or April. In some European countries, pixels with VH values greater than -15.5 in May were assigned smaller WTCI values to reduce their probability of becoming winter-triticeae crops.

Response: Thank you for your suggestion. The threshold is derived from the study of Dong et al. (2020), which was obtained through the analysis of filed samples. The details can be found in response to Q1.

**23). L366: Or other land covers**

Line 366: The product is provided in Geotiff format with pixel values of 1 for winter-triticeae crops and 0 for non-winter-triticeae crops.

Response: Yes, we have revised this sentence:

    "The product is provided in GeoTIFF format with pixel values of 1 for winter-triticeae crops and 0 for other land covers."

---

## Referee Report (RR1)

I understand this is revised submission, and hence my comments are limited to considering that status.

There is no methodology mentioned in abstract. "First global 30 m resolution distribution maps was produced from what and how? After this, add one to two sentences about how the study was carried out?

Line 50: clarify what is 'long-term distribution maps"

Line 64 -70: these are about method and better fits in section 2 Data and method not in section 1.

Methodology needs better explanation because results can be trusted based on methodology. Line 95: how many data sets used of Landsat, Sentinel-1. Similarly, procedure is too short, for example what was the process of noise removal, radiometric calibration, terrain correction – there are no theorical foundation nor procedure explained.

Line 171-172. " On the contrary, the NDVI of natural vegetation peaks". Incomplete sentence.

Section 2.4 what was references used for accuracy assessment.

Texts are often confusing and readability is hampered. Even in results section, the sentences are composed as if it is method section still describing the method, e.g. line 261-266, line 311, 321,…. Sentences could be better composed to reflect the contents of the respective section. Another example, when you say, in conclusion, the study proposed a new method, then follow-up with a sentence describing salient feature of that method WTCI, what and how to do. Also introduce the limitations, of the method if any, and the recommendation. The contents in conclusion section mostly reads as findings.

---

## Referee Report (RR2)

**Reviewer comments to author**

The paper presents significant contributions to the field and demonstrates substantial effort. However, there are still a few areas for improvement in the methodology and accuracy evaluation sections.

**Methodology**: While the paper provides additional explanations on the calculation and application of the WTCI, the threshold calculation for the index remains unclear. The authors mention relying on provincial or state-level statistics to set the threshold but fail to detail how these statistics are used. It is unclear how the consistency between statistical data and threshold products is determined, whether a single year's data is used to set thresholds for other years, or if yearly statistical data is directly used for threshold determination. Detailed explanations are needed for the index threshold determination. Additionally, the paper uses NDVI>0.4 to identify potential crop areas, based on a study from a small region in Sichuan, China. It is questionable if this threshold can be globally applicable. The authors should provide more references or experiments to validate the reliability of this threshold.

**Accuracy assessment**: The paper dedicates significant space to demonstrating high agreement between the product and statistical data. However, since the methodology relies on statistical areas to set thresholds, the evaluation does not convincingly reflect the product's accuracy. The authors should elaborate on the index threshold determination method and consider using crop reference layers in regions with such data to further verify the product's accuracy through consistent area distribution.

**Figures and Tables**: The design of figures and tables requires improvement. For instance, Fig. 3(a) shows texture features of different land types, which is not mentioned anywhere in the paper. The authors should reconsider the necessity of this figure. In Fig. 6, adding results from crop reference layers could provide a clearer and more intuitive presentation of data quality to the readers.

Overall, the paper is well-conceived and the work is substantial. However, there are issues in the presentation of the methodology and accuracy evaluation that need addressing. Therefore, I recommend major revisions before reconsidering the paper for publication.

---

## Referee Report (RR3)

**Summary**

I have performed an earlier review of the initial manuscript, and will therefore limit myself to remaining questions based on author feedback and additional comments based on the latest version of the manuscript.

Overall, the current version of the manuscript has improved from a content point of view, even though I still have some remaining questions listed below. However, from a data point of view, I still see the same issues I raised before. I will give some examples below which for me still hamper the use of the published data.

Therefore, I recommend (major) revisions, especially to the data provision.

**Manuscript comments:**

L23: terminology of these datasets has changed. LPIS only contains the parcel geometries. GSAA contains the crop type declarations which is what was used here (ref: https://wikis.ec.europa.eu/download/attachments/86968605/JRC133145_lpisgsa_v05_finalb.pdf?version=1&modificationDate=1691571477191&api=v2). I suggest to update throughout the manuscript to be in line with official terminology.

L104-105: please explain why the nearest neighbor method is preferred over another resampling method that would be closer to the aggregated effect of several Sentinel-2 pixels embedded in one Landsat pixel

L114: similar question: why nearest neighbor resampling?

L135-151: sampling from CDL and LPIS/GSAA is only done for cropland. How can the method be validated for commission errors in other non-crop land covers?

L171-172: why not shrubland or wetland?

L324: coming back to my earlier comment in the first review, I remain reluctant to accept that computing area statistics from pixel counting is a good approach here. Such area statistics are biased (see Olofsson et al., 2014). Please comment on this.

**Data comments:**

In general, I still have issues with understanding the projection of the individual files. In case standardized projection information is encoded in the files, visualizing them in software such as QGIS should be straightforward. However, for some files I checked this is still not the case. Files like the Belgian and France ones are still offsetting by default when being imported in QGIS. How does a user correctly visualize these?

Some other comments after checking some files:

*Uzbekistan_2017*

The method seems to be triggered in certain plantations (first picture) and also larger regions that seem not to be related to winter triticeae. What is causing this?

[Figure]

[Figure]

*India_West_Bengal_2021*

When checking this file, I stumbled upon an artefact on the west side of the product which contains a stripe of 1 (winter triticeae) values which is clearly an artefact.

[Figure]

*France_2021*

Example of the projection issue that I still encounter:

[Figure]

In previous review round I mentioned a strong artefact which the authors replied to be related to the projection issue I was facing. I'm not convinced by this however. There seems to be another reason which really causes this difference and artefact. Please investigate and explain.

[Figure]

**Technical corrections:**

L120: great -> greater

---

## Author Response (AR2)

*Earth System Science Data*

Paper # essd-2023-432

May. 25, 2024

Dear editor and reviewers:

We are very grateful to you for your constructive comments and suggested amendments on our manuscript entitled: "**High-resolution mapping of global winter-triticeae crops using a sample-free identification method**" (essd-2023-432). The comments have improved the paper quite tremendously. We have carefully studied the comments and revised our manuscript accordingly.

Here are our detailed responses to your comments. Please note that the comments from you are in **bold font** followed by our responses in regular font, changes/additions to the manuscript are underlined.

Sincerely yours,

Wenping Yuan on behalf of all co-authors

Corresponding author: Wenping Yuan, Ph.D., Professor

School of Atmospheric Sciences,

Sun Yat-sen University

135 West Xingang Road, Guangzhou 510275, China

E-mail address: yuanwp3@mail.sysu.edu.cn

**Detailed responses to reviewers' comments**

**Reviewer #1**

**1). There is no methodology mentioned in abstract. "First global 30 m resolution distribution maps was produced from what and how? After this, add one to two sentences about how the study was carried out?**

Response: Thanks for your comments. In fact, we have described the method used to produce the distribution map of winter -triticeae crops before this sentence. Here, we have revised and refined this sentence:

"In this study, we propose a new method based on the Winter-Triticeae Crops Index (WTCI) for global winter-triticeae crops mapping. This is a new sample-free method for identifying winter-triticeae crops based on differences in their normalized difference vegetation index (NDVI) characteristics from the heading to the harvesting stages and those of other types of vegetation. We considered state (or province) or country as an identification unit and employed WTCI to produce the first global 30 m resolution distribution maps of winter-triticeae crops from 2017 to 2022 using Landsat and Sentinel images."

**2). Line 50: clarify what is 'long-term distribution maps"**

Line 50: Therefore, it is necessary to produce long-term distribution maps of winter-triticeae crops with a high-spatial resolution for these countries.

Response: We have revised this sentence:

"Therefore, it is necessary to produce distribution maps of winter-triticeae crops with high-spatial resolution and continuous years for these countries."

**3). Line 64 -70: these are about method and better fits in section 2 Data and method not in section 1.**

Line 64 -70: Here, based on Landsat 7, Landsat 8 and Sentinel-2 satellite data, this study developed the Winter-Triticeae Crops Index (WTCI), a sample-free method for identifying the global distribution of winter-triticeae crops. Specifically, we first designed the WTCI based on

the NDVI differences between winter-triticeae crops and other vegetation types. Then, we applied this method to identify the winter-triticeae crops in 66 countries worldwide. Finally, we assessed the accuracy and spatiotemporal transferability of the WTCI method based on field survey samples, visual interpretation samples from high-resolution images on Google Earth, CDL dataset, the Land Parcel Identification System (LPIS) dataset and agricultural statistical data. Ultimately, we produced 30 m spatial resolution distribution maps of winter-triticeae crops from 2017 to 2022 in 66 countries (2020 for US, see 2.2.2 for details) worldwide to fill such product gaps, providing a data basis for yield estimation and crop management.

Response: Thanks for your suggestion. We have revised this paragraph:

"This study aims to develop a new sample-free method, i.e., Winter-Triticeae Crops Index (WTCI), to identify global winter-triticeae crops based on Landsat 7, Landsat 8, Sentinel-1 and Sentinel-2 satellite data. The main goals are to (1) assess the accuracy and spatiotemporal transferability of the new method using field survey samples, visual interpretation samples from high-resolution images on Google Earth, CDL dataset, the Land Parcel Identification System (LPIS) dataset and agricultural statistical data, (2) produce 30 m spatial resolution distribution maps of winter-triticeae crops in 66 countries worldwide from 2017 to 2022 to fill such product gaps, providing a data basis for yield estimation and crop management."

**4). Methodology needs better explanation because results can be trusted based on methodology. Line95: how many data sets used of Landsat, Sentinel-1. Similarly, procedure is too short, for example what was the process of noise removal, radiometric calibration, terrain correction – there are no theorical foundation nor procedure explained.**

Line 95: In this study, we used Landsat 7 collection 2 data and Landsat 8 collection 2 data, as well as Sentinel-2 data on the Google Earth Engine (GEE) platform to obtain NDVI from 2016 to 2022, all of which were surface reflectance (SR) products and have undergone atmospheric correction.

Response: Thank you for your suggestion. We used Landsat 7, Landsat 8, Sentinel-2 and Sentinel-1 datasets in our study. We obtained all available images during the study period from

these datasets, and processed these data pixel by pixel (2.2.1 section in the original manuscript), therefore, we did not calculate how many scenes were used in this study. In addition, the sentinel-1 data for each scene provided on the GEE platform has been pre-processed with thermal noise removal, radiometric calibration, and terrain correction ([https://developers.google.com/earth-engine/datasets/catalog/COPERNICUS_S1_GRD](https://developers.google.com/earth-engine/datasets/catalog/COPERNICUS_S1_GRD)). Each procedure of processing NDVI and VH data is described in section 2.2.1 of the original manuscript. Here, we revised some contents, and the details are as follows:

"In this study, we used all available Landsat 7 collection 2 data (USGS Landsat 7 Level 2, Collection 2, Tier 1) and Landsat 8 collection 2 data (USGS Landsat 8 Level 2, Collection 2, Tier 1), as well as Sentinel-2 data (Harmonized Sentinel-2 MSI: MultiSpectral Instrument, Level-2A) on the Google Earth Engine (GEE) platform to obtain NDVI from 2016 to 2022, all of which were surface reflectance (SR) products and have undergone atmospheric correction."

"The data provided on GEE platform has undergone thermal noise removal, radiometric calibration, and terrain correction. We applied a refined Lee filter (Abramov et al., 2017) to alleviate the impact of speckle noise caused by the interferences between adjacent backscatter returns, and finally obtained the monthly maximum composite values of VH from 2016 to 2022 and resampled them to 30 m using the nearest neighbour method to keep consistency with NDVI. These operations were also run on the GEE platform."

**5). Line 171-172. "On the contrary, the NDVI of natural vegetation peaks". Incomplete sentence.**

Line 171-172: On the contrary, the NDVI of natural vegetation peaks.

Response: Thank you for your reminder. We have modified this sentence:

"On the contrary, the NDVI of natural vegetation approaches its peak in a year."

**6). Section 2.4 what was references used for accuracy assessment.**

Response: We added some references for accuracy assessment, and the details are as follows:

"The producer's accuracy (PA), user's accuracy (UA), overall accuracy (OA) and F1 score (Congalton, 1991; Hripcsak and Rothschild, 2005; Lin et al., 2022) were employed to validate

the identification accuracy at the pixel scale."

"The correlation coefficient ($R^2$) and relative mean absolute error (RMAE) were used to examine the consistency between the identified area and the statistical area (Shen et al., 2023; Zheng et al., 2022)."

References:

Lin, C. X., Zhong, L. H., Song, X. P., Dong, J. W., Lobell, D. B., Jin, Z. N.: Early- and in-season crop type mapping without current-year ground truth: Generating labels from historical information via a topology-based approach, Remote Sens. Environ., 274, 112994, https://doi.org/10.1016/j.rse.2022.112994, 2022.

Shen, R. Q., Pan, B. H., Peng, Q. Y., Dong, J., Chen, X. B., Zhang, X., Ye, T., Huang, J. X., and Yuan, W. P.: High-resolution distribution maps of single-season rice in China from 2017 to 2022, Earth Syst. Sci. Data., 15, 3203–3222, https://doi.org/10.5194/essd-15-3203-2023, 2023.

Zheng, Y., dos Santos Luciano, A. C., Dong, J., Yuan, W. P.: High-resolution map of sugarcane cultivation in Brazil using a phenology-based method, Earth Syst. Sci. Data., 14, 2065–2080, https://doi.org/10.5194/essd-14-2065-2022, 2022.

**7). Texts are often confusing and readability is hampered. Even in results section, the sentences are composed as if it is method section still describing the method, e.g. line 261-266, line 311, 321,…. Sentences could be better composed to reflect the contents of the respective section. Another example, when you say, in conclusion, the study proposed a new method, then follow-up with a sentence describing salient feature of that method WTCI, what and how to do. Also introduce the limitations, of the method if any, and the recommendation. The contents in conclusion section mostly reads as findings.**

line 261-266: This study first identified the spatial distribution of winter-triticeae crops in 66 countries in 2020 based on the WTCI. Fig. 4 shows that winter-triticeae crops are mainly distributed in mid-latitude regions, including most countries in Europe (Fig. 4b), the plains of Asia (Fig. 4c), northern Africa (Fig. 4d), the southern edge of Australia (Fig. 4e), middle of US (Fig. 4f) and the southeast regions of South America (Fig. 4g). To display the detailed

information of the winter-triticeae crops map produced by this study, we selected twelve typical areas in different countries to zoom in and compared them with high-resolution images from Google Earth (Fig. 5).

Line 311: To examine the temporal transferability of the WTCI method, this study applied the optimal percentile of the V and B lines in 2020 to other years.

Line 321: Based on CDL and LPIS datasets, we further validated the performance of the WTCI method in the US and Europe.

Response: Thank you for your suggestion. We have revised and refined some contents in results and conclusions sections, the details are as follows:

"The spatial distribution map of winter-triticeae crops in 66 countries in 2020 was first produced based on the WTCI method (Fig. 4), which effectively presented the distribution of winter-triticeae crops in the study area. Specifically, the winter-triticeae crops were mainly distributed in most European countries and Asian plains (Fig. 4b and 4c). To display the detailed information of the map of winter-triticeae crops, we selected twelve typical areas in different countries to zoom in and compared them with high-resolution images from Google Earth (Fig. 5). In general, despite some noise, the identification map clearly displays the fields planted with winter-triticeae crops and effectively distinguishes roads and rivers between the fields."

"The comparison between the identified and statistical areas of winter-triticeae crops indicates that the WTCI method can be effectively applied to other years."

"The distribution map of winter-triticeae crops exhibited high consistency with CDL and LPIS datasets."

"This study proposed a new sample-free method (WTCI) for mapping winter-triticeae crops and examined its performance in 66 countries worldwide. The new method exhibits high accuracy and strong spatiotemporal transferability by comparing the produced maps with field survey and Google Earth samples, the CDL and LPIS datasets, and agricultural statistical data. Overall, the OA and F1 score were more than 80% and 75% in most of identification units, respectively. The $R^2$ between identified and statistical areas in most of regions was greater than 0.6 in all years, and RMAE less than 30%. These satisfactory results indicate that the WTCI method can be used for long-term and large-scale crop mapping. At the same time, the first 30

m spatial resolution distribution maps of winter-triticeae crops from 2017 to 2022 produced by the WTCI method fills the current product gaps, which can be further served for the harvest area monitoring, yield estimation and agricultural management."

**Reviewer #2**

**The authors responded to my comments and improved the study substantially. The main improvement includes the validation of obtained results with CDL and LPIS datasets. However, I still have some comments.**

Response: Thank you for your comments and affirmation of our revised manuscript. Here, we have revised our manuscript based on your comments this time, and we also attached a point-by point letter to you. The detailed responses are listed below.

**1). Thank you for addressing the issue of (lack of) harmonization of Sentinel-2 and Landsat data, as well as Landsat-7 failure-related striping issue. But I think it should be already mentioned in the methods/data description. And maybe it would be good to discuss more deeply the limitations and errors, such as maquis mentioned in my previous review. Maybe also adding one figure with examples of errors/uncertainties would be beneficial.**

Response: Thank you for your deep thought and suggestion. We have added some contents in data and discussions sections, and the details are as follows:

"We choose Landsat 7 satellite to obtain more available data although a malfunction in its scan line corrector. To ensure the data quantity and quality, we first removed the pixels with clouds."

"Additionally, due to the scan line corrector failed of the Landsat 7 sensor, the striping issues and reduced data availability may also impact the accuracy of NDVI time series (Ju and Roy., 2008), resulting in the errors in identification results. In our study, there were some striping issues in the distribution map of winter-triticeae crops in a few regions (Fig. S1a), which may lead to errors in winter-triticeae crops identification and the differences in identification results between different years (Fig. S1)."

[Figure]

**Figure S1: Comparison of distribution maps of winter-triticeae crops between different years. (a) and (b) show the zoomed-in maps of subregion in France in 2018 and 2021, respectively.**

**2). Furthermore, I still cannot fully understand how the authors selected samples based on Google Earth imagery and how they could say that these were winter triticeae and not other types of crops. I understand that they were later again checked with the NDVI curves. However, what you can see for example in Figure 5 – some of the identified winter triticeae crops are represented with vegetation and some with bare soils on the corresponding Google Earth high resolution images. What are the dates high resolution images from Google Earth were collected? I think this is important information and needs further clarification, because the types of crops may change from year to year.**

Response: Thank you for your meticulous thinking and suggestion. We have added some contents and pictures to illustrate how we selected samples based on Google Earth image. The new pictures in Fig .2a are the Google Earth image corresponding to the filed survey. The details are as follows:

"We first chose regions with available images during the growing season of winter-triticeae crops (section 2.3.3), and selected samples from these regions based on the texture features and colors. Winter-triticeae crops have deeper color or stronger texture than winter rapeseed and grassland, and their roughness is lower than that of forest, which can be used to

distinguish winter-triticeae crops from other land cover types (Fig .2a). Crops with different growing season (such as, maize, rice, and soybean) will not affect the visual interpretation. To ensure the accuracy of the samples, we then validated the selected samples on GEE platform by checking whether the NDVI temporal features of these samples matched the characteristics of winter-triticeae crops, and finally obtained 7,029 winter-triticeae crops samples and 8,897 non-winter-triticeae crops samples (Fig. 1)."

[Figure]

**Figure 2: Example of the (a) textures and colors on the high-resolution images from © Google Earth and (b) NDVI time series characteristics of different land cover types. The red five-pointed stars represent the different phenological stages of winter-triticeae crops.**

In addition, we have added dates for each image in Figure 5 and updated Fig. 5a and 5f to ensure that the displayed images are during the growing season of winter-triticeae crops. The identified winter-triticeae crops appears as vegetation or bare soil on the corresponding Google Earth high-resolution image may be related to the date of the image or the planting habits of

farmers, such as the early or late planting or harvesting time. Besides, as described in the discussion section, the quality of satellite data can also lead to some identification errors. In general, the identification map clearly displays the winter-triticeae crops fields and effectively distinguishes other land covers.

[Figure]

**Figure 5: Comparison between the identification maps of winter-triticeae crops and high-resolution images from © Google Earth in the study area. (A1-L1) represent the high-resolution images from Google Earth of different regions; (A2-L2) represent the zoomed-in maps of area A-L in Figure 4.**

**Other comments:**

**3). Line 23 and others: F1 should be also reported in %, similarly as producer's and user's accuracy**

Line 23: the overall accuracy and F1 score in most regions of the United States and Europe

were more than 80% and 0.75.

Response: Thank you for your suggestion. We have revised F1 score in the manuscript to be expressed in %, and the details are as follows. Meanwhile, we have modified the corresponding figures (Figure 6, Figure 7 and Figure 10), and the details can be found in the revised manuscript.

"Moreover, compared with the Cropland Data Layer (CDL) and the Land Parcel Identification System (LPIS) datasets, the overall accuracy and F1 score in most regions of the United States and Europe were more than 80% and 75%."

"the overall accuracy (OA), producer's accuracy (PA), and user's accuracy (UA) of the winter-triticeae crops identification maps in 65 countries (except US) were 87.7%, 81.12% and 87.85%, respectively, and the F1 score was 84.04% (Fig. 6). PA and UA varied between 52% and 97.73%, 63.64% and 97.83% over the various countries, and OA and F1 ranged from 70.86% to 96.05% and 65.63% to 96.09%, respectively. At state (province) scale, the variation range of OA and F1 score in China were 77.68% to 95.9% and 71.79% to 94.47%, respectively (Fig. 7a). In Brazil, the OA and F1 score were in the range of 76.99%-94.74% and 78.26%-96.24% (Fig. 7b). The OA in India was between 67.53% and 92.07%, and the F1 score was between 65.24% and 92.05% (Fig. 7c). The OA and F1 score in Australia lied in the range of 79.21% to 91.67% and 69.23% to 91% (Fig. 7d). In general, the F1 score in most of the identification units was greater than 75%, indicating that the WTCI method shows satisfactory accuracy in identifying winter-triticeae crops. The regions with F1 scores less than 75% were mainly found in small winter-triticeae crops planting areas and complex winter crop types, such as Croatia (HRV), Albania (ALB), Sichuan (SC) province in China, and Bihar (BR) state in India."

"In 2020, the OA and F1 score in the US were 86.84% and 82.09%."

"For all states planting winter-triticeae crops, the OA varied from 70.42% to 94.24%, and the F1 score ranged from 66.67% to 91.01% (Fig. 10a-10c)."

"In major planting states, such as Kansas, Oklahoma and Texas, the planting area of winter-triticeae crops accounted for approximately 50% of the total area of winter-triticeae crops in the US, with OA and F1 score over 92% and 85%, respectively (Fig. 10)"

"Among the 10 European countries from LPIS datasets, the OA, F1 score, PA and UA ranged from 71.22% to 94.79%, 67.67% to 90.14%, 63.68% to 84.77% and 71.43% to 96.24%,

with the mean value of 83.88%, 78.87%, 73.18% and 86% (Fig. 10d), respectively."

"In general, the OA and F1 score in most of regions of US and Europe were higher than 80% and 75%, implying that the WTCI method exhibited satisfactory performance compared to the CDL and LPIS datasets."

"Overall, the OA and F1 score were more than 80% and 75% in most of identification units, respectively."

**4). Line 55: ability to what?**

Line 55: For example, Ge et al. (2021) combined Landsat images with the CDL production of Arkansas to train a classifier and then assessed the ability of the classifier in California, USA, and Liaoning, China.

Response: Thank you for your reminder. we have revised this sentence:

"For example, Ge et al. (2021) combined Landsat images with the CDL production of Arkansas to train a classifier and then assessed the spatial transferability of the classifier in California, USA, and Liaoning, China."

**5). Line 73: I think you should use words for cardinal numbers less than 10**

Line 73: The study area covers 66 countries, including 36 European countries, 15 Asian countries, 8 African countries, 2 North American country, 4 South American countries, and 1 Oceania country (Fig.1).

Response: Thank you for your suggestion. we have modified the representation of numbers less than 10:

"The study area covers 66 countries, including 36 European countries, 15 Asian countries, eight African countries, two North American country, four South American countries, and one Oceania country (Fig.1)."

**6). Figure 1: validation samples are not very well visible, maybe use more distinct colors**

Response: Thank you for your suggestion. We have modified the Figure 1:

[Figure]

**Figure 1: Distribution of the study area and validation samples. The study area is the region covered in grey; The legend indicates the winter-triticeae (WT) crops samples and non-winter-triticeae (Non-WT) crops samples from Cropland Data Layer (CDL) dataset of the United States, the Land Parcel Identification System (LPIS) dataset of Europe, and field survey in China, as well as visual interpretation base on Google Earth images, respectively.**

**7). Line 95: Please add the names of the used collections in GEE**

Line 95: In this study, we used Landsat 7 collection 2 data and Landsat 8 collection 2 data, as well as Sentinel-2 data on the Google Earth Engine (GEE) platform to obtain NDVI from 2016 to 2022, all of which were surface reflectance (SR) products and have undergone atmospheric correction.

Response: We have added the names of the used collections in GEE, and the details are as follows:

"In this study, we used all available Landsat 7 collection 2 data (USGS Landsat 7 Level 2, Collection 2, Tier 1) and Landsat 8 collection 2 data (USGS Landsat 8 Level 2, Collection 2, Tier 1), as well as Sentinel-2 data (Harmonized Sentinel-2 MSI: MultiSpectral Instrument, Level-2A) on the Google Earth Engine (GEE) platform to obtain NDVI from 2016 to 2022, all of which were surface reflectance (SR) products and have undergone atmospheric correction."

**8). Line 284: plating -> planting**

Line 284: The regions with F1 scores less than 0.75 were mainly found in small winter-triticeae

crops planting areas and complex winter crop types.

Line 286: On the contrary, the identification accuracy of regions with larger planting areas of winter-triticeae crops was significantly higher than that of regions with smaller plating areas.

Response: Thank you for your reminder. We speculate that you are referring to line 286, and we have revised this word:

"On the contrary, the identification accuracy of regions with larger planting areas of winter-triticeae crops was significantly higher than that of regions with smaller planting areas."

**9). Figure 6: what a-e represent? Maybe you can sort countries by descending/ascending accuracy?**

Response: Thank you for your suggestion. Figure 6a-6e represent the producer's accuracy (PA), user's accuracy (UA), overall accuracy (OA) and F1 score of the identification maps of winter-triticeae crops at each country in 2020. It is a good idea to sort countries by descending/ascending accuracy. But we think sorting them in ascending order of country abbreviations is more convenient for readers to match the identification results of these countries in the figure with the full names of the countries in the supplement, therefore, we modified Figure 6 according to this rule. Meanwhile, we have also made modifications to Figures 7-11 according to the same rules to ensure consistency throughout the entire manuscript. The details can be found in the revised manuscript.

[Figure]

**Figure 6: The producer's accuracy (PA), user's accuracy (UA), overall accuracy (OA) and F1 score of the identification maps of winter-triticeae crops at national scale in 2020. The abbreviations of countries are shown in Table S2 in the supplement.**

10). Figures 8 and 9: I would use reversed color palette for RMAE (green – low error, red – high error).

Response: Thank you for your suggestion. We have revised Figure 8 and 9, and the details are as follows:

[Figure]

**Figure 8: Comparison between identified and statistical areas of winter-triticeae crops at national scale from 2017 to 2022. (a) and (b) show the correlation coefficient and RMAE between identified and statistical areas, respectively.**

[Figure]

**Figure 9: Comparison between identified and statistical areas of winter-triticeae crops at state (province) scale from 2017 to 2022. (a1-d1) represent the correlation coefficient at state (province) scale in China, Brazil, India, and Australia, respectively; (a2-d2) represent the RMAE at state (province) scale in China, Brazil, India, and Australia, respectively.**

**11). Figure 12: Why is LPIS shown as points and CDL as polygons?**

Response: The CDL product is a crop classification map with a spatial resolution of 30 meters, so we would like to compare the spatial consistency between the identification map produced using the WTCI method and the CDL. The LPIS data we collected is the location information of land cover types, so we compared it with our identification map using coordinate points.

**12). Figures 4 &13: are coordinates necessary here? You don't use them in the Figure 12.**

Response: Thank you for your reminder. We have removed the coordinates to keep these figures consistent, and the details are as follows:

[Figure]

**Figure 4: Spatial distribution of winter-triticeae crops in the study area in 2020. (a) shows the distribution of winter-triticeae crops in 66 countries; (b-g) show the zoomed-in maps of Europe, Asia, Africa, Australia, North America and South America, respectively.**

[Figure]

**Figure 13: Harvest time of winter-triticeae crops in the study area in 2020.**

---

## Author Response (AR3)

*Earth System Science Data*

Paper # essd-2023-432

Jun. 11, 2024

Dear editor:

We would like to thank you for your constructive comments and suggested amendments on our manuscript. The comments are very useful for us to improve our research. We have carefully studied the comments and revised our manuscript accordingly.

Here are our detailed responses to your comments. Please note that the comments from you are in **bold font** followed by our responses in regular font, changes/additions to the manuscript are underlined.

Sincerely yours,

Wenping Yuan on behalf of all co-authors

Corresponding author: Wenping Yuan, Ph.D., Professor

School of Atmospheric Sciences,

Sun Yat-sen University

135 West Xingang Road, Guangzhou 510275, China

E-mail address: yuanwp3@mail.sysu.edu.cn

**Detailed responses to editors' comments**

**We hope this message finds you well. Thank you for your efforts in revising the manuscript entitled "High-resolution mapping of global winter-triticeae crops using a sample-free identification method." We appreciate the substantial improvements made based on the previous round of feedback.**

**However, we believe there are still a few areas that would benefit from further clarification and enhancement. Specifically, the methodology section, particularly the calculation and application of the Winter-Triticeae Crops Index (WTCI), could be more detailed (section. 2.3 Method before Fig. 2a, 2b). Adding a visual representation, such as a flowchart, would significantly aid in understanding the steps involved in the WTCI calculation. In addition, while the validation process using various datasets is well-documented in the revised version, many if not all explanations were provided with little or no justification. Please provide more details on the selection criteria and the rationale behind using specific datasets. The selection criteria and the rationale would enhance the robustness and transparency of the study.**

**Considering these points, we suggest that the revised manuscript undergo another round of review to ensure that these critical aspects are fully addressed. This will help in confirming that the methodology is comprehensively understood and that the validation processes are thoroughly documented.**

**We appreciate your attention to these details and look forward to receiving the revised version of the manuscript.**

Response: Thank you for your deep thought and suggestions. We have revised and added some details in method section, and a flowchart was drawn to represent the calculation and application process of the WTCI method. The details are as follows:

[revised manuscript text omitted]

In addition, we have added more details on the selection criteria and the rationale behind using various validation samples, and the details are as follows:

[revised manuscript text omitted]

---

## Author Response (AR4)

*Earth System Science Data*

Paper # essd-2023-432

Sep 10, 2024

Dear editor and reviewers:

Thank you for taking time out of your busy schedule to review the manuscript entitled: "**High-resolution mapping of global winter-triticeae crops using a sample-free identification method**" (essd-2023-432). Your comments provide valuable insights for improving the contents and analysis. We have carefully studied the comments and revised our manuscript accordingly.

Here are our detailed responses to your comments. Please note that the comments are in **bold font** followed by our responses in normal font, changes/additions to the manuscript are underlined.

Sincerely,

Wenping Yuan on behalf of all co-authors

Corresponding author: Wenping Yuan, Ph.D., Professor

School of Atmospheric Sciences,

Sun Yat-sen University

135 West Xingang Road, Guangzhou 510275, China

E-mail address: yuanwp3@mail.sysu.edu.cn

**Detailed responses to editor and reviewers' comments**

**# Editor**

Thank you for your efforts in revising your manuscript titled "Global Winter-Triticeae Crops Mapping Using a Sample-Free Method." The reviewers acknowledge the improvements made but have highlighted several concerns that need to be addressed. We invite you to carefully revise your manuscript based on their detailed comments.

Reviewer 1 notes the progress made but still has several concerns. They suggest providing a map showing the spatial pattern of WTCI thresholds to illustrate variations among identification units. Additionally, they recommend discussing how multiple winter-triticeae crops in an administrative unit might impact threshold suitability and data accuracy. Clarification is needed on whether using USDA's national-level crop calendar affects the reliability of WTCI thresholds at the province/state level. Reviewer 1 also requests an explanation of whether the NDVI change curves can represent all winter-triticeae crops and surrounding vegetation. They advise including a comparison statistical analysis of WTCI values between winter-triticeae and non-winter-triticeae crops. Finally, they suggest revising the titles and content focus of Sections 3.4 and 4.3.

Reviewer 2 appreciates the significant contributions of your work but points out areas for improvement in methodology and accuracy evaluation. Detailed explanations are needed on how provincial or state-level statistics are used to set the WTCI threshold and how consistency is determined across years. The global applicability of the NDVI>0.4 threshold should be validated with more references or experiments. They also recommend elaborating on the index threshold determination method and using crop reference layers to verify accuracy. Reviewer 2 suggests improving the design of figures and tables, particularly reconsidering Figure 3(a) and adding results from crop reference layers in Figure 6.

Reviewer 3, while acknowledging improvements, still has concerns, especially regarding data provision. They advise updating terminology to reflect official terms, such as using GSAA instead of LPIS. Justification is needed for the choice of the nearest neighbor resampling method. Reviewer 3 also raises questions about validating

**commission errors in non-crop land covers and addressing projection issues in data files to ensure correct visualization in software like QGIS.**

**Please refer to the reviewers' detailed reports for specific comments and suggestions. We look forward to receiving your revised manuscript and believe these revisions will significantly enhance the quality of your work.**

Response: We appreciate you for giving us this opportunity to revise this paper. We try to address the issues raised as best as possible and have responded to them one by one. The detailed responses are listed below.

**Reviewer #1**

**Winter-triticeae crops are among the most important grain crops in the world, thus mapping its distribution is helpful for crop yield estimation, crop planting pattern optimization, and food security assessment. This study developed a new global winter-triticeae crops map by using a sample-free method, and has a relative high accuracy validated by using the field samples, CDL, LPIS data, and agricultural statistical data. I have reviewed the revised manuscript and the point-by-point responses to the comments from the other reviewers. The authors worked well in addressing the comments. However, I still have some concerns for the revised manuscript and provides as follows:**

Response: Thank you for your comments and affirmation of our revised manuscript. We deeply appreciate your time for reviewing the manuscript. Your suggestions are very useful for us to improve our manuscript. Here, we have revised our manuscript based on your comments, and we also attached a point by point letter to you. The detailed responses are listed below.

**1). This study used the planted area to determine the threshold of WTCI at administrative units. So, are there large variations in the thresholds among all the identification units? I suggest providing a map in the supplementary materials to show the spatial pattern of WTCI threshold.**

Response: Thank you for your suggestion. We have added a figure (Figure. S2) in the supplementary to display the spatial pattern of WTCI threshold. Overall, the spatial differences between the WTCI thresholds of all identification units are relatively small, and these thresholds mainly range from 0.3 to 0.6.

[Figure]

**Figure S2:** The spatial distribution of WTCI thresholds in all identification units in 2020.

**2). In this study, winter-triticeae crops include winter wheat, winter barley, winter rye, and triticale. If there are multiple types of winter-triticeae crop in an administrative unit, the threshold may not suitable for some crops and further impact on the data accuracy.**

Response: Thank you for your deep thought. Previous studies indicated that winter-triticeae crops, including winter wheat, winter barley, winter rye and triticale, have similar seasonal change curves of NDVI and phenological characteristics (Huang et al., 2022; Xu et al., 2017). Therefore, we identified them as a whole. There may be differences between these crops, leading to differences in threshold and affecting identification accuracy. We have added some discussion in the Discussion section of the revised manuscript, and the details are as follows:

"Besides, this study ignored the internal differences between winter wheat, winter barley, winter rye and triticale due to their similar NDVI time series and phenological characteristics (Huang et al., 2022; Xu et al., 2017), which may affect the identification accuracy." (Line 471-473)

"In the future, identifying useful bands or vegetation indexes that eliminate interferences from other land covers, further subdividing each winter-triticeae crop, as well as increasing the availability and quality of satellite data, will further promote the performance of the WTCI method." (Line 476-478)

**3). Line 239-241: USDA only provides national-level crop calendar, would it influence the**

**reliability of WTCI threshold at province/state level?**

Line 239-214: Specifically, this study referred to crop calendar data provided by the United States Department Agriculture (USDA) (https://ipad.fas.usda.gov/ogamaps/cropcalendar.aspx) to determine the growth season of winter-triticeae crops in each country.

Response: Thank you for your careful consideration. We use the national level crop calendar as a reference for phenological periods, which only defines the range of phenological periods in a country. Importantly, our method can consider the phenological differences in different regions within a country. Specifically, the parameters of WTCI method are determined automatically during the winter-triticeae crops growing season, for example, the maximum and minimum values of NDVI and their occurrence times are automatically searched during the regreening to harvesting stages of winter-triticeae crops. It should be noted that the time when the maximum and minimum values of NDVI appear is not fixed, but is flexibly determined based on the NDVI curve characteristics of each pixel, which considering the phenological differences between different regions. Therefore, although we used the national level crop calendar as a reference to determine the WTCI threshold at province (or state) level in some countries, the advantage of the WTCI method can effectively balance the phenological differences between regions. Moreover, our results further demonstrate the reliability of WTCI threshold at province (or state) level, despite the lack of detailed crop calendar information in these regions.

**4). Figure 3 shows the time series of winter-triticeae crops and other natural vegetation, can these NDVI change curves represent all winter-triticeae crops, other crops and surrounding natural vegetation around the study area (i.e., 66 countries)?**

Response: According to the record of statistical data and prior knowledge, the main winter crops are winter-triticeae and winter rapeseed in the study area. The variety of summer crops is relatively abundant, but their phenological period is significantly different from that of winter crops, which will not affect the identification of winter-triticeae crops. Therefore, the crops shown in the Figure 3 are representative and widely planted in the study area, and the natural vegetation types are also typical and widely distributed. In addition, the key point of the WTCI method is to distinguish winter-triticeae crops based on the phenological characteristics of

different land cover types. The NDVI time series in Figure 3 can accurately reflect the phenological characteristics of different land cover types during their growing seasons. Most importantly, they can clearly distinguish winter-triticeae crops from them. We believe that this is the main message that Figure 3 intends to deliver. We also added the NDVI times series of wetland and shrub based on the suggestion from another reviewer to further support our study.

[Figure]

**Figure 3:** Example of the (a) textures and colours on the high-resolution images from © Google Earth and (b) NDVI time series characteristics of different land cover types. The red five-pointed stars represent the different phenological stages of winter-triticeae crops.

**5). The WTCI is the key variable to identify the winter-triticeae crops, a comparison statistical analysis (e.g., a box plot of WTCI values for different vegetation type or frequence distribution map) in WTCI between winter-triticeae crops and non-winter-**

**triticeae crops should be provided in the manuscript.**

Response: Thank you for your suggestion. We have plotted a figure to compare the WTCI values between winter-triticeae crops and non-winter-triticeae crops. Except for winter rapeseed, there are significant differences in WTCI values between other land cover types and winter-triticeae crops, and this study used VH to exclude winter rapeseed when identifying winter-triticeae crops. Due to the fact that the calculation of WTCI values requires the use of phenological period of winter-triticeae crops, and the phenological period of summer crops is obviously different from that of winter-triticeae crops, they do not participate in the calculation of WTCI. Therefore, the figure only displays the WTCI values of some land cover types that overlap with the phenological period of winter-triticeae crops, and does not show the WTCI values of summer crops.

[Figure]

**Figure S1:** WTCI values of different land cover types. Letters represent statistically significant differences in WTCI values for different land cover types (Tukey's Test, P < 0.05).

**6). Section 3.4. The title should be "Harvest time of global winter-triticeae crops" rather than "Harvest dynamics of global winter-triticeae crops", because there is no temporal analysis.**

Response: Thank you for your reminder. We have revised the title of section 3.4.

**7). The winter-triticeae crops dataset refers planted area rather than harvested area. While, Section 4.3 emphasized too much on the harvested area rather than the spatial variations of harvested time.**

Response: Thank you for your reminder. In fact, the winter-triticeae crops dataset refers to harvested area, as we used NDVI during the harvest period when identifying winter-triticeae crops. In addition, we speculate that you are referring to section 3.4. We described the harvested area in this section to show and compare the proportion of winter-triticeae crops harvested area at different harvest times to the global winter-triticeae crops harvested area. Here, we have added some contents to describe the spatial variations of harvested time, and the details are as follows:

"Overall, the harvest time of winter-triticeae crops is delayed with increasing latitude (Fig. 14). In the Northern Hemisphere, winter-triticeae crops in East and South Asia were harvested in May and June (Fig. 14c), and the harvested area accounted for about 35.64% of the total harvested area in the study area (Fig. 15). The harvest time in Central Asia, Europe, North Africa and North America was concentrated between July and August (Fig. 14b, 14c 14d and 14f), and the proportion of harvested area to the total area was around 47.05% (Fig. 15). The regions with harvest time in September were mainly distributed in high latitude areas of Russia (Fig. 14b). In the Southern Hemisphere, the harvest time of winter-triticeae crops was mainly from November to January of the following year (Fig. 14e and 14g), with the harvested area accounting for 13.7% of the total harvested area (Fig. 15). These areas with the harvest time occurring from November to January were mainly located in high latitude regions of Australia and South America (Fig. 14e and 14g), and the harvest time in October only occurred in some areas of low latitude regions of South America (Fig. 14g)." (Line 393-402)

**Specific comments:**

**8). Line 74-75: 278.87 and 209.15 refer planting area, harvested area, or physical area?**

Response: Thank you for your reminder. 278.87 and 209.15 refer harvested area. Here, we have revised these contents that:

"The harvested area of global triticeae crops (including spring and winter varieties) is

278.87 million ha in 2020 (https://www.fao.org/faostat/en/#data), with winter-triticeae crops accounting for about 75% (i.e., 209.15 million ha) of the global triticeae crops harvested area (Zhao et al., 2018). According to the statistics of the winter-triticeae crops area provided on official websites of various countries (Table S1), the total harvested area of winter-triticeae crops in our study area in 2020 is 207.45 million ha, occupying 99.19% of the global winter-triticeae crops harvested area." (Line 73-78)

**9). Line 77: 207.45 million ha. Dot not comma.**

Response: Thank you for your meticulous discovery. We have revised this punctuation, and the details can be found in the response to Q8.

**10). Line 349-351: How do you get this conclusion? Delete this sentence or give some examples.**

Line 349-351: Similar to the results of 2020, the regions with a higher error are concentrated in areas with small planting areas of winter-triticeae crops and diverse planting types of winter crops.

Response: Thank you for your suggestion. We have deleted this sentence.

**11). Line 360: I didn't find the $R^2$ of any state was 1.**

Line 360: At state scale, the $R^2$ varied between 0.52 to 1, and the RMAE was in 9.01%-57.84% (Fig. 12b-12w).

Response: Thanks for pointing this out. We have modified this sentence as follows:

"At state scale, the $R^2$ varied between 0.52 to 0.96, and the RMAE was in 9.01%-57.84% (Fig. 12b-12w)." (Line 369-370)

**12). Line 410: Why does the US winter-triticeae crops data not include year of 2017-2019 and 2021-2022? The reason should be explained in manuscript.**

Response: Thank you for your comments. The United States already has high-accuracy and annually updated Cropland Data Layer (CDL) product, while other countries where wintertriticeae crops are planted widely still lack high-accuracy distribution maps of winter-triticeae crops. To avoid duplication of work, this study focuses on producing distribution maps of winter-triticeae crops for other countries. On the other hand, this study developed the WTCI method based on 2020, therefore only the winter-triticeae crops data in 2020 in the US was produced and compared with CDL product to validate the performance of the WTCI method. In fact, we have explained the reason on Line 47-52 and Line 135-142 of the original manuscript.

**13). Line 450: Revise "Sichuan (SC)" as "Sichuan (SC) province of China".**

Line 450: First, the commission error is higher in regions where winter-triticeae crops are not dominant crops, such as in Sichuan (SC), West Bengal (WB), Bihar (BR), Karnataka (KA) and few countries in Mediterranean Sea region indicating that here non-winter-triticeae crops are misclassified as winter-triticeae crops.

Response: Thank you for your advice. We have revised this sentence, and the details are as follows:

"First, the commission error is higher in regions where winter-triticeae crops are not dominant crops, such as Sichuan (SC) province of China, West Bengal (WB), Bihar (BR), Karnataka (KA) and few countries in Mediterranean Sea region, indicating that here non-winter-triticeae crops are misclassified as winter-triticeae crops." (Line 458-461)

**14). Line 453: What does "…large differences in the available images…" mean? quantity or quality of satellite images?**

Line 453: Second, although we used synthetized images from Landsat and Sentinel productions to increase the amount of effective data, there are still large differences in the available images among the study area.

Response: Thanks for pointing this out. We have revised this sentence:

"Although we used synthetized images from Landsat and Sentinel productions to increase the amount of effective data and conducted linear interpolation and the Savitzky-Golay filter to further improve data quality, there are still differences in the quantity and quality of satellite data among the study area." (Line 461-464)

**15). Line 457: The cloud pixels in satellite images have been removed as the descriptions in the methods, why the cloud and rain contaminations still introduce noise in the NDVI?**

Line 457: For example, cloud and rain contaminations introduce noise in the NDVI data and consequently dampen the winter-triticeae crops detection signal (Song et al., 2017; Xiao et al., 2014).

Response: Thank you for your reminder. We have deleted this sentence.

**16). Line 476: "Google Earth samples" is weird. Revise it as "visual interpretation samples from Google Earth images".**

Line 476: The new method exhibits high accuracy and strong spatiotemporal transferability by comparing the produced maps with field survey and Google Earth samples, the CDL and LPIS datasets, and agricultural statistical data.

Response: Thank you for your suggestion. We have revised this sentence, and the details are as follows:

"The new method exhibits high accuracy and strong spatiotemporal transferability by comparing the produced maps with field survey samples and visual interpretation samples from Google Earth images, the CDL and EuroCrops datasets, and agricultural statistical data." (Line 485-487)

References:

Huang, X. J., Fu, Y. Y., Wang, J. J., Dong, J., Zheng, Y., Pan, B. H., Skakun, S., Yuan, W. P.: High–resolution mapping of winter cereals in Europe by time series Landsat and Sentinel images for 2016–2020, Remote Sens., 14(9), 2120, https://doi.org/10.3390/rs14092120, 2022.

Xu, X. M., Conrad, C., Doktor, D.: Optimising phenological metrics extraction for different crop types in Germany using the moderate resolution imaging Spectrometer (MODIS). Remote Sens., 9(3), 254, https://doi.org/10.3390/rs9030254, 2017.

**Reviewer #2**

**The paper presents significant contributions to the field and demonstrates substantial effort. However, there are still a few areas for improvement in the methodology and accuracy evaluation sections.**

Response: Thank you for your comments and suggestions. Your suggestions are very valuable for us to improve our research. Here, we have revised our manuscript based on your suggestions, and we also attached a point by point letter to you. The detailed responses are listed below.

**1). Methodology: While the paper provides additional explanations on the calculation and application of the WTCI, the threshold calculation for the index remains unclear. The authors mention relying on provincial or state-level statistics to set the threshold but fail to detail how these statistics are used. It is unclear how the consistency between statistical data and threshold products is determined, whether a single year's data is used to set thresholds for other years, or if yearly statistical data is directly used for threshold determination. Detailed explanations are needed for the index threshold determination. Additionally, the paper uses NDVI>0.4 to identify potential crop areas, based on a study from a small region in Sichuan, China. It is questionable if this threshold can be globally applicable. The authors should provide more references or experiments to validate the reliability of this threshold.**

Response: Thank you for your suggestions. We have added some contents to explain the calculation of WTCI threshold, and the details are as follows:

"The potential pixels (_N_th) with high WTCI values are considered winter-triticeae crops in a given identification unit, and the total area of all _N_ potential pixels should be equal to the agricultural statistical area of the identification unit." (Line 234-236)

"In this study, we considered each state (or province) as an identification unit in China, Brazil, India, Australia and US, and the threshold of WTCI was determined based on statistical area at state (or province) scale. For the remaining countries, we treated each country as an identification unit, and the threshold of WTCI was calculated relied on statistical area at national scale. The annual statistical area was used to determine the threshold of WTCI for each

identification unit in the current year." (Line 245-248)

In addition, we have added some references to support the reliability of the threshold (NDVI>0.4), and the details are as follows:

"Some regional and global scale studies have reported that NDVI greater than 0.4 usually indicates vegetation cover (Ma et al., 2022; Peng et al., 2019; Xu et al., 2023; Yang et al., 2024; Yang et al., 2024)." (Line 175-176)

**2). Accuracy assessment: The paper dedicates significant space to demonstrating high agreement between the product and statistical data. However, since the methodology relies on statistical areas to set thresholds, the evaluation does not convincingly reflect the product's accuracy. The authors should elaborate on the index threshold determination method and consider using crop reference layers in regions with such data to further verify the product's accuracy through consistent area distribution.**

Response: Thank you for your suggestion. In fact, the statistical data used to determine the threshold and the statistical data used for accuracy validation are independent of each other. Specifically, this study used province (or state) scale statistical area to determine the thresholds for China, Brazil, India, Australia and the United States, and evaluated the accuracy of each province (or state) using the statistical area of low-level administrative regions, such as, municipal or county scale. A province (or state) can contain dozens or hundreds of municipalities or counties. The national scale statistical area was used to determine the WTCI thresholds for other counties, and the statistical area of all states or provinces or municipalities or counties included in each country was used to evaluate accuracy. The method of accuracy assessment using agricultural statistical area was described in the section 2.4 of the original manuscript. Here, we randomly selected some regions to verify the relationship between national scale statistical area and province scale identification area (Fig. 1a), as well as province scale statistical area and municipal scale identification area (Fig. 1b). It can be seen that there is almost no correlation between the two variables, indicating that our method of setting thresholds using statistical area is reliable.

[Figure]

**Figure 1:** Comparison between (a) national scale statistical area and province scale identification area of winter-triticeae crops in Romania and Ukraine and (b) province scale statistical area and municipal scale identification area of winter-triticeae crops in Shandong and Henan provinces of China.

In addition, we have added some contents to explain the process of determining the WTCI threshold, and the details can be found in the response to Q1. Moreover, we validated the product's accuracy using CDL and EuroCrops datasets in section 3.3 of the original manuscript, which have high recognition and are widely used for accuracy assessment of data products.

**3). Figures and Tables: The design of figures and tables requires improvement. For instance, Fig. 3(a) shows texture features of different land types, which is not mentioned anywhere in the paper. The authors should reconsider the necessity of this figure. In Fig. 6, adding results from crop reference layers could provide a clearer and more intuitive presentation of data quality to the readers.**

Response: Thank you for your suggestions. Fig 3(a) is to explain how to select validation samples based on the features of Google Earth images. In fact, we have described the Fig 3(a) in section 2.2.2 of the original manuscript. According to the suggestion of another reviewer, we have further improved the content and Figure 3. The details are as follows:

"We first chose regions with available images during the growing season of winter-triticeae crops (section 2.3.3), and selected samples from these regions based on the texture features and colours. Winter-triticeae crops have deeper colour or stronger texture than winter rapeseed and grassland, and their roughness is lower than that of forest, which can be used to

distinguish winter-triticeae crops from other land cover types (Fig .3a). The images of wetland and shrub show obvious differences from those of winter-triticeae crops. Wetland have dual characteristics of water and vegetation, and without regular texture features. Shrub have lower vegetation coverage and stronger graininess. These features make them easy to distinguish from winter-triticeae crops (Fig .3a). Crops with different growing season (such as maize, rice, and soybean) will not affect the visual interpretation." (Line 125-132)

In addition, we have added comparisons with crop reference layers produced by other studies, and the details are as follows:

"In addition, we compared the spatial distribution map of winter-triticeae crops in this study with some existing products in Europe (Huang et al., 2022) and China (Dong et al., 2020), which also have a spatial resolution of 30 m. The spatial distribution of winter-triticeae crops fields in the maps produced in this study was similar to other studies, and the maps generated by WTCI method had less noise and clearer boundaries of roads and rivers (Fig. S3)." (Line 304-308)

[Figure]

**Figure 3:** Example of the (a) textures and colours on the high-resolution images from © Google Earth and (b) NDVI time series characteristics of different land cover types. The red five-pointed stars represent the different phenological stages of winter-triticeae crops.

[Figure]

**Figure S3:** Comparison between the identification maps of this study and other studies. (a1-d1) represent the high-resolution images from © Google Earth in the study area; (a2-d2) represent the zoomed-in maps of the identification results based on WTCI method; (a3-d3) represent the zoomed-in maps of the identification results of other studies. Area a-d can be found in Figure 5.

References:

Ma, Z., Dong, C., Lin, K., Yan, Y., Luo, J., Jiang, D., Chen, X.: A Global 250-m Downscaled NDVI Product from 1982 to 2018, Remote Sens., 14(15), 3639, https://doi.org/10.3390/rs14153639, 2022.

Xu, S., Zhu, X. L., Chen, J., Zhu, X. L., Duan, M. J., Qiu, B. W., Wang, L. M., Tan, X. Y., Xu, Y. N., Cao, R. C.: A robust index to extract paddy fields in cloudy regions from SAR time series, Remote Sens. Environ., 285, 113374, https://doi.org/10.1016/j.rse.2022.113374, 2023.

Yang, J. Y., Wu, T. X., Sun, X. Y., Liu, K., Farhan, M., Zhao, X., Gao, Q. S., Yang, Y. Y., Shao,

Y. H., Wang, S. D.: Global 24 solar terms phenological MODIS normalized difference vegetation index dataset in 2001–2022, Geosci. Data J., 00, 1–12, https://doi.org/10.1002/gdj3.268, 2024.

Yang, J., Yan, D. M., Yu, Z. L., Wu, Z. N., Wang, H. L., Liu, W. M., Liu, S. M., Yuan, Z.: NDVI variations of different terrestrial ecosystems and their response to major driving factors on two side regions of the Hu-Line, Ecol Indic., 159, 111667, https://doi.org/10.1016/j.ecolind.2024.111667, 2024.

**Reviewer #3**

**Summary**

I have performed an earlier review of the initial manuscript, and will therefore limit myself to remaining questions based on author feedback and additional comments based on the latest version of the manuscript.

Overall, the current version of the manuscript has improved from a content point of view, even though I still have some remaining questions listed below. However, from a data point of view, I still see the same issues I raised before. I will give some examples below which for me still hamper the use of the published data.

Therefore, I recommend (major) revisions, especially to the data provision.

Response: Thank you for your comments and affirmation of our revised manuscript. We also appreciate your clear and detailed feedback. Here, we have revised our manuscript based on your comments this time and attached a point by point letter to you. The detailed responses are listed below.

**Manuscript comments:**

**1). L23: terminology of these datasets has changed. LPIS only contains the parcel geometries. GSAA contains the crop type declarations which is what was used here (ref: https://wikis.ec.europa.eu/download/attachments/86968605/JRC133145_lpisgsa_v05_fin alb.pdf?version=1&modificationDate=1691571477191&api=v2). I suggest to update throughout the manuscript to be in line with official terminology.**

Line 23: Moreover, compared with the Cropland Data Layer (CDL) and the Land Parcel Identification System (LPIS) datasets, the overall accuracy and F1 score in most regions of the United States and Europe were more than 80% and 75%.

Response: Thank you for your suggestion. In fact, the LPIS dataset we use comes from a publicly available EuroCrops dataset that includes detailed crop types. Specifically, the EuroCrops project manually collected all publicly available self-declared crop reporting datasets from countries of the European Union, with the LPIS dataset being an important component. Then they developed a new version of the Hierarchical Crop and Agriculture

Taxonomy (HCAT) in order to organize all crops that are cultivated within the EU into a common hierarchical representation scheme. The detailed information can be found in https://doi.org/10.1038/s41597-023-02517-0. In this study, we have chosen EuroCrops data that winter-triticeae crops were clearly labelled as validation data. We believe that the EuroCrops dataset is reliable for accuracy assessment. Here, we have refined some content, and the details are as follows:

"The EuroCrops dataset, supported by the German Space Agency at DLR on behalf of the Federal Ministry for Economic Affairs and Climate Action (BMWK), is combines all publicly available self-declared crop reporting datasets from countries of the European Union. Importantly, this dataset utilizes a new version of Hierarchical Crop and Agriculture Taxonomy (HCAT) to provide a unified hierarchical representation scheme for all crops within the European Union (Schneider et al., 2023). We collected 10 countries (Austria, Belgium, Germany, Denmark, Estonia, France, Netherlands, Slovakia, Slovenia and Sweden) with winter-triticeae crops clearly labelled in EuroCrops dataset, including winter spelt, winter barley, winter durum hard wheat, winter common soft wheat, winter triticale, winter rye and winter oats (https://zenodo.org/records/10118572), and these data cover the period from 2018 to 2021." (Line 144-151)

**2). L104-105: please explain why the nearest neighbor method is preferred over another resampling method that would be closer to the aggregated effect of several Sentinel-2 pixels embedded in one Landsat pixel.**

Line 104-105: Then, based on nearest neighbour method, we resampled the NDVI of Sentinel-2 to 30 m to keep the same spatial resolution as Landsat data.

Response: Thank you for your deep thought. We randomly selected two groups pixels of 10 m × 10 m (9 pixels per group) from the 10 m resolution image of Sentinel-2, and extracted the NDVI curves of each pixel during the winter-triticeae crops growing season. Furthermore, we used the nearest neighbor method to resample Sentinel-2 image to 30 m, and searched the corresponding 30m × 30m pixels in the resampled image, and extracted their NDVI curves separately. There are slight differences in NDVI values between pixels with 30 m resolution

and pixels with 10 m resolution, pixels with 30 m resolution can still accurately reflect the trend of NDVI changes over time (Fig. 2). The WTCI method also focuses on the trend of NDVI changes. In addition, even if the nearest neighbor method may have some impact on data quality, this study used linear interpolation and the Savitzky-Golay filter to further improve the data quality. Some studies have demonstrated the available and valuable of the nearest neighbor method, and indicated that this seemingly simple method remains competitive in some cases against the state-of-the-art techniques (Boiman et al., 2008; Chen and Shah, 2018; Weinberger and Saul, 2009).

[Figure]

**Figure 2:** Comparison of NDVI time series between 10 m and 30 m pixels.

**3). L114: similar question: why nearest neighbor resampling?**

Line 114: and finally obtained the monthly maximum composite values of VH from 2016 to 2022 and resampled them to 30 m using the nearest neighbour method to keep consistency with NDVI.

Response: The reason can be found in response to Q2.

**4). L135-151: sampling from CDL and LPIS/GSAA is only done for cropland. How can the method be validated for commission errors in other non-crop land covers?**

Response: Thank you for your suggestion. In fact, we conducted sampling both in cropland and non-cropland based on CDL and EuroCrops datasets, and the details can be found in lines 140

and 148-149 of the original manuscript. Here, we have added some details information about sampling from CDL and EuroCrops datasets:

"Non-winter-triticeae crops samples were randomly generated in the remaining pixels, including other crops pixels in cultivated land and non-cultivated land pixels." (Line 141-142)

"We first convert the polygon file into point file using Acrmap 10.2, then randomly extracted winter-triticeae crops samples from the point file labelled with winter-triticeae crops in each country, and selected non-winter-triticeae crops samples from other land cover types, such as forest, grassland or other crops." (Line 151-154)

In addition, we have added tables in the supplement to display the validation results of commission errors for non-winter triticeae crops (or other non-crop land covers), and the details are as follows:

**Table S4:** The confusion matrix of the identification maps of winter-triticeae crops based on CDL dataset.

| Country | CDL samples | Map | | Producer's accuracy (%) | User's accuracy (%) | Overall accuracy (%) | F1 score (%) |
|---|---|---|---|---|---|---|---|
| | | Winter-triticeae crops | Non-Winter-triticeae crops | | | | |
| Alabama (AL) | Winter-triticeae crops | 99 | 38 | 72.26 | 88.39 | 88.54 | 79.52 |
| | Non-Winter-triticeae crops | 13 | 295 | 95.78 | 88.59 | | |
| Arkansas (AR) | Winter-triticeae crops | 101 | 18 | 84.87 | 90.18 | 90.76 | 87.45 |
| | Non-Winter-triticeae crops | 11 | 184 | 94.36 | 91.09 | | |
| California (CA) | Winter-triticeae crops | 34 | 20 | 62.96 | 79.07 | 75.21 | 70.10 |
| | Non-Winter-triticeae crops | 9 | 54 | 85.71 | 72.97 | | |
| Colorado (CO) | Winter-triticeae crops | 204 | 31 | 86.81 | 90.67 | 93.59 | 88.70 |
| | Non-Winter-triticeae crops | 21 | 555 | 96.35 | 94.71 | | |
| Delaware | Winter-triticeae | 308 | 62 | 83.24 | 90.59 | 90.12 | 86.76 |

| | | | | | | | |
|---|---|---|---|---|---|---|---|
| (DE) | crops | | | | | | |
| | Non-Winter-triceae crops | 32 | 549 | 94.49 | 89.85 | | |
| Georgia (GA) | Winter-triticeae crops | 64 | 31 | 67.37 | 84.21 | 81.47 | 74.85 |
| | Non-Winter-triticeae crops | 12 | 125 | 91.24 | 80.13 | | |
| Idaho (ID) | Winter-triticeae crops | 174 | 66 | 72.50 | 80.18 | 82.36 | 76.15 |
| | Non-Winter-triticeae crops | 43 | 335 | 88.62 | 83.54 | | |
| Illinois (IL) | Winter-triticeae crops | 247 | 49 | 83.45 | 92.51 | 91.68 | 87.74 |
| | Non-Winter-triticeae crops | 20 | 513 | 96.25 | 91.28 | | |
| Indiana (IN) | Winter-triticeae crops | 287 | 95 | 75.13 | 96.31 | 89.21 | 84.41 |
| | Non-Winter-triticeae crops | 11 | 589 | 98.17 | 86.11 | | |
| Kansas (KS) | Winter-triticeae crops | 344 | 46 | 88.21 | 93.99 | 93.13 | 91.01 |
| | Non-Winter-triticeae crops | 22 | 578 | 96.33 | 92.63 | | |
| Kentucky (KY) | Winter-triticeae crops | 130 | 25 | 83.87 | 90.28 | 90.30 | 86.96 |
| | Non-Winter-triticeae crops | 14 | 233 | 94.33 | 90.31 | | |
| Maryland (MD) | Winter-triticeae crops | 216 | 78 | 73.47 | 85.04 | 87.14 | 78.83 |
| | Non-Winter-triticeae crops | 38 | 570 | 93.75 | 87.96 | | |
| Michigan (MI) | Winter-triticeae crops | 345 | 80 | 81.18 | 97.46 | 91.32 | 88.58 |
| | Non-Winter-triticeae crops | 9 | 591 | 98.50 | 88.08 | | |

| State | Crop type | | | | | | |
|---|---|---|---|---|---|---|---|
| Missouri (MO) | Winter-triticeae crops | 148 | 50 | 74.75 | 85.06 | 88.74 | 79.57 |
| | Non-Winter-triticeae crops | 26 | 451 | 94.55 | 90.02 | | |
| Mississippi (MS) | Winter-triticeae crops | 73 | 30 | 70.87 | 83.91 | 82.26 | 76.84 |
| | Non-Winter-triticeae crops | 14 | 131 | 90.34 | 81.37 | | |
| Montana (MT) | Winter-triticeae crops | 122 | 51 | 70.52 | 81.88 | 84.46 | 75.78 |
| | Non-Winter-triticeae crops | 27 | 302 | 91.79 | 85.55 | | |
| North Carolina (NC) | Winter-triticeae crops | 108 | 30 | 78.26 | 89.26 | 86.73 | 83.40 |
| | Non-Winter-triticeae crops | 13 | 173 | 93.01 | 85.22 | | |
| North Dakota (ND) | Winter-triticeae crops | 63 | 45 | 58.33 | 77.78 | 70.42 | 66.67 |
| | Non-Winter-triticeae crops | 18 | 87 | 82.86 | 65.91 | | |
| Nebraska (NE) | Winter-triticeae crops | 263 | 51 | 83.76 | 88.55 | 89.68 | 86.09 |
| | Non-Winter-triticeae crops | 34 | 476 | 93.33 | 90.32 | | |
| New Jersey (NJ) | Winter-triticeae crops | 203 | 70 | 74.36 | 91.44 | 85.02 | 82.02 |
| | Non-Winter-triticeae crops | 19 | 302 | 94.08 | 81.18 | | |
| New Mexico (NM) | Winter-triticeae crops | 79 | 32 | 71.17 | 84.95 | 79.46 | 77.45 |
| | Non-Winter-triticeae crops | 14 | 99 | 87.61 | 75.57 | | |
| New York (NY) | Winter-triticeae crops | 167 | 70 | 70.46 | 84.77 | 77.43 | 76.96 |
| | Non-Winter- | 30 | 176 | 85.44 | 71.54 | | |

| State | Crop type | | | | | | |
|-------|-----------|---|---|---|---|---|---|
| Ohio (OH) | Winter-triticeae crops | 315 | 49 | 86.54 | 94.59 | | |
| | Non-Winter-triticeae crops | 18 | 510 | 96.59 | 91.23 | 92.49 | 90.39 |
| Oklahoma (OK) | Winter-triticeae crops | 159 | 27 | 85.48 | 90.34 | | |
| | Non-Winter-triticeae crops | 17 | 561 | 97.06 | 95.41 | 94.24 | 87.85 |
| Oregon (OR) | Winter-triticeae crops | 244 | 36 | 87.14 | 91.73 | | |
| | Non-Winter-triticeae crops | 22 | 489 | 95.69 | 93.14 | 92.67 | 89.38 |
| Pennsylvania (PA) | Winter-triticeae crops | 162 | 34 | 82.65 | 90.00 | | |
| | Non-Winter-triticeae crops | 18 | 379 | 95.47 | 91.77 | 91.23 | 86.17 |
| South Carolina (SC) | Winter-triticeae crops | 91 | 28 | 76.47 | 89.22 | | |
| | Non-Winter-triticeae crops | 11 | 152 | 93.25 | 84.44 | 86.17 | 82.35 |
| South Dakota (SD) | Winter-triticeae crops | 147 | 61 | 70.67 | 84.00 | | |
| | Non-Winter-triticeae crops | 28 | 313 | 91.79 | 83.69 | 83.79 | 76.76 |
| Tennessee (TN) | Winter-triticeae crops | 99 | 33 | 75.00 | 90.83 | | |
| | Non-Winter-triticeae crops | 10 | 234 | 95.90 | 87.64 | 88.56 | 82.16 |
| Texas (TX) | Winter-triticeae crops | 113 | 24 | 82.48 | 89.68 | | |
| | Non-Winter-triticeae crops | 13 | 353 | 96.45 | 93.63 | 92.64 | 85.93 |
| Utah (UT) | Winter-triticeae crops | 76 | 29 | 72.38 | 83.52 | 81.12 | 77.55 |

| Country | Samples | Map Winter-triticeae crops | Map Non-Winter-triticeae crops | Producer's accuracy (%) | User's accuracy (%) | Overall accuracy (%) | F1 score (%) |
|---|---|---|---|---|---|---|---|
| | Non-Winter-triticeae crops | 15 | 113 | 88.28 | 79.58 | | |
| Virginia (VA) | Winter-triticeae crops | 124 | 27 | 82.12 | 90.51 | 90.85 | 86.11 |
| | Non-Winter-triticeae crops | 13 | 273 | 95.45 | 91.00 | | |
| Washington (WA) | Winter-triticeae crops | 275 | 80 | 77.46 | 84.88 | 86.28 | 81.00 |
| | Non-Winter-triticeae crops | 49 | 536 | 91.62 | 87.01 | | |
| Wisconsin (WI) | Winter-triticeae crops | 214 | 64 | 76.98 | 89.54 | 85.71 | 82.79 |
| | Non-Winter-triticeae crops | 25 | 320 | 92.75 | 83.33 | | |
| Wyoming (WY) | Winter-triticeae crops | 100 | 42 | 70.42 | 89.29 | 84.66 | 78.74 |
| | Non-Winter-triticeae crops | 12 | 198 | 94.29 | 82.50 | | |

**Table S5:** The confusion matrix of the identification maps of winter-triticeae crops based on EuroCrops dataset.

| Country | EuroCrops samples | Map | | Producer's accuracy (%) | User's accuracy (%) | Overall accuracy (%) | F1 score (%) |
|---|---|---|---|---|---|---|---|
| | | Winter-triticeae crops | Non-Winter-triticeae crops | | | | |
| Austria (AUT) | Winter-triticeae crops | 240 | 96 | 71.43 | 89.22 | 85.05 | 79.34 |
| | Non-Winter-triticeae crops | 29 | 471 | 94.20 | 83.07 | | |
| Belgium (BEL) | Winter-triticeae crops | 139 | 50 | 73.54 | 76.37 | 86.50 | 74.93 |
| | Non-Winter-triticeae crops | 43 | 457 | 91.40 | 90.14 | | |
| Denmark (DNK) | Winter-triticeae crops | 185 | 60 | 75.51 | 84.09 | 83.76 | 79.57 |
| | Non-Winter-triticeae crops | 35 | 305 | 89.71 | 83.56 | | |

| Country | | | | | | | |
|---|---|---|---|---|---|---|---|
| Estonia (EST) | Winter-triticeae crops | 128 | 73 | 63.68 | 92.75 | 82.96 | 75.52 |
| | Non-Winter-triticeae crops | 10 | 276 | 96.50 | 79.08 | | |
| France (FRA) | Winter-triticeae crops | 285 | 57 | 83.33 | 89.34 | 87.53 | 86.23 |
| | Non-Winter-triticeae crops | 34 | 354 | 91.24 | 86.13 | | |
| German (DEU) | Winter-triticeae crops | 128 | 23 | 84.77 | 96.24 | 94.79 | 90.14 |
| | Non-Winter-triticeae crops | 5 | 381 | 98.70 | 94.31 | | |
| Netherlands (NLD) | Winter-triticeae crops | 62 | 27 | 69.66 | 93.94 | 87.98 | 80.00 |
| | Non-Winter-triticeae crops | 4 | 165 | 97.63 | 85.94 | | |
| Slovakia (SVK) | Winter-triticeae crops | 161 | 78 | 67.36 | 80.90 | 71.22 | 73.52 |
| | Non-Winter-triticeae crops | 38 | 126 | 76.83 | 61.76 | | |
| Slovenia (SVN) | Winter-triticeae crops | 108 | 30 | 78.26 | 85.71 | 84.26 | 81.82 |
| | Non-Winter-triticeae crops | 18 | 149 | 89.22 | 83.24 | | |
| Sweden (SWE) | Winter-triticeae crops | 45 | 25 | 64.29 | 71.43 | 74.71 | 67.67 |
| | Non-Winter-triticeae crops | 18 | 82 | 82.00 | 76.64 | | |

**5). L171-172: why not shrubland or wetland?**

Line 171-172: After applying these steps, the main remaining land cover types in the potential pixels were forest, grassland, and cultivated land.

Response: Thank you for your reminder. We have modified Figure 3 and added NDVI time

series of shrub and wetland for comparison with winter-triticeae crops. Meanwhile, we have also made modifications to the corresponding content. The details are as follows:

"After applying these steps, the main remaining land cover types in the potential pixels were forest, grassland, cultivated land, wetland and shrub." (Line 178-179)

"There are significant differences in the temporal variations of NDVI between winter-triticeae crops and natural vegetation types (i.e., deciduous forest, evergreen forest, shrub and grassland) as well as wetland during the growing season of winter-triticeae crops (Fig. 3b). Specifically, in the period from seedling to tillering stages, winter-triticeae crops are in a state of slow growth, with their NDVI gradually increasing. In contrast, natural vegetation types are in the deciduous stage and exhibit a continuous decrease in NDVI during this period, and wetland also exhibit the similar characteristics (Fig. 3b). From the regreening to the heading stages, the NDVI of winter-triticeae crops rapidly increases and reaches its maximum value, while the increase of NDVI of natural vegetation types and wetland tends to lag behind that of winter-triticeae crops (Fig. 3b). Furthermore, the NDVI of winter-triticeae crops show a downward trend and reach their lowest value during the harvesting stage. However, the NDVI values of natural vegetations and wetland rapidly increase at this time (Fig. 3b)." (Line 180-188)

**Figure 3:** Example of the (a) textures and colours on the high-resolution images from © Google Earth and (b) NDVI time series characteristics of different land cover types. The red five-pointed stars represent the different phenological stages of winter-triticeae crops.

**6). L324: coming back to my earlier comment in the first review, I remain reluctant to accept that computing area statistics from pixel counting is a good approach here. Such area statistics are biased (see Olofsson et al., 2014). Please comment on this.**

Line 324: In addition, compared to the agricultural statistical area in different administrative units in 2020, the WTCI method can effectively estimate the planting area of winter-triticeae crops.

Response: Thank you for your deep thought. We have carefully read the paper you suggested. The "good practices" recommendations mentioned in the paper can be used to obtain more accurate areas by using sample-based approach to calculate the area to compensate for the bias

introduced by area estimation based on map (e.g., pixel counting). In this study, our main goal is to develop the identification method of winter-triticeae crops and produce the high-resolution distribution maps of winter-triticeae crops, and the area estimation is only a part of accuracy assessment. Even if we adjusted the area using sample-based approach estimation, we could not change the mapping results on the pixel scale. More importantly, the distribution map produced in our study is a simple binary (1 represents winter-triticeae crops and 0 represents non-winter-triticeae crops), and each pixel has a regular shape of 30 m × 30 m. Therefore, we tend to believe that the method of calculating area in the study is applicable for this situation. Previous studies (Shen et al., 2023; Zheng et al., 2022) have also used the same method for area validation. Of course, we highly appreciate the comprehensiveness and rigor of "good practice" methodology in area estimation, and we are willing to use this method in our further studies.

**Data comments:**

**7). In general, I still have issues with understanding the projection of the individual files. In case standardized projection information is encoded in the files, visualizing them in software such as QGIS should be straightforward. However, for some files I checked this is still not the case. Files like the Belgian and France ones are still offsetting by default when being imported in QGIS. How does a user correctly visualize these?**

Response: Sorry for any confusion caused. Maybe we didn't explain it clearly in our last reply. Although we have added a documentation with the dataset to introduce the spatial projection, users still need to convert the projection according to own needs to match the reference layer data. For the convenience of users, we have unified the spatial reference of all maps as WGS84 (EPSG:4326), a commonly used coordinate system, and update the dataset (https://doi.org/10.57760/sciencedb.12361).

**Some other comments after checking some files:**

**8). Uzbekistan_2017**

**The method seems to be triggered in certain plantations (first picture) and also larger regions that seem not to be related to winter triticeae. What is causing this?**

[Figure]

[Figure]

Response: Thank you for your carefully check. First, we cannot confirm whether the year between the image and distribution map of winter-triticeae crops matches, and whether the time of the image is during the winter-triticeae growing season. In addition, although the distribution maps of winter-triticeae crops have achieved good results in most regions, the global mapping accuracy has not yet reached 100%. Therefore, as shown in the pictures you presented, there

may be errors in some areas. We have discussed the reasons for errors caused by the WTCI method in the Discussion section of the original manuscript and added some new contents, and the details as below. These discussions on errors will promote us to improve the WTCI method in future work or provide references for other researchers.

"Besides, this study ignored the internal differences between winter wheat, winter barley, winter rye and triticale due to their similar NDVI time series and phenological characteristics (Huang et al., 2022; Xu et al., 2017), which may affect the identification accuracy. We referred to previous studies (Dong et al., 2020; Huang et al., 2022) on winter crop mapping and only distinguished winter rapeseed to reduce its impact on the identification of winter-triticeae crops. Other winter crops with smaller planting area that have not been discovered or overlooked may also interfere with the identification and lead to errors in the identification map. In the future, identifying useful bands or vegetation indexes that eliminate interferences from other land covers, further subdividing each winter-triticeae crop, as well as increasing the availability and quality of satellite data, will further promote the performance of the WTCI method." (Line 471-478)

**9). India_West_Bengal_2021**

**When checking this file, I stumbled upon an artefact on the west side of the product which contains a stripe of 1 (winter triticeae) values which is clearly an artefact.**

[Figure]

Response: Thank you very much for your meticulous inspection. We have investigated the reason of the above issue and suspect that there may be an error in the output of winter-triticeae crops identification map and we have re-output the identification map for this state (Fig. 3).

Meanwhile, the dataset has been checked and updated.

[Figure]

**Figure 3:** The distribution map of winter-trirticeae crops in the West Bengal state of India in 2021.

**10). France_2021**

**Example of the projection issue that I still encounter:**

[Figure]

Response: Sorry for the inconvenience caused to you. We have unified the spatial references of all identification maps to WGS84 (EPSG:4326) for user convenience. We have carefully checked to ensure that these data can be displayed correctly on QGIS, for example:

[Figure]

**11). In previous review round I mentioned a strong artefact which the authors replied to be related to the projection issue I was facing. I'm not convinced by this however. There seems to be another reason which really causes this difference and artefact. Please investigate and explain.**

[Figure]

Response: Thank you for your reminder. We have rechecked this data and compared it with the distribution maps of winter-triticeae crops in other years (in the figure below), and found that the reason for the above phenomenon is the striping issues of the satellite data. This issue has been discussed in the Discussion section of the original manuscript.

[Figure]

**Technical corrections:**

**12). L120: great -> greater**

Line 120: In the fieldwork, we only selected large winter-triticeae crops fields with an area great than 900 m$^2$, and used GPS (G120, UniStrong, Beijing, China) (Fu et al., 2023b) to mark the locations inside the fields.

Response: Thank you for pointing this out. We have revised this word:

"In the fieldwork, we only selected large winter-triticeae crops fields with an area greater than 900 m$^2$, and used GPS (G120, UniStrong, Beijing, China) (Fu et al., 2023b) to mark the locations inside the fields." (Line 118-119)

References:

Boiman, O., Shechtman, E., Irani, Michal.: In defense of nearest-neighbor based image classification, In 2008 IEEE Conference on Computer Vision and Pattern Recognition, 10.1109/CVPR.2008.4587598, 2008.

Chen, G. H., Shah, Devavrat.: Explaining the success of nearest neighbor methods in prediction, Foundations and Trends® in Machine Learning, 10(5–6), 337–588, 10.1561/2200000064, 2018.

Shen, R. Q., Pan, B. H., Peng, Q. Y., Dong, J., Chen, X. B., Zhang, X., Ye, T., Huang, J. X., and Yuan, W. P.: High-resolution distribution maps of single-season rice in China from 2017 to 2022, Earth Syst. Sci. Data., 15, 3203–3222, https://doi.org/10.5194/essd-15-3203-2023, 2023.

Weinberger, K. Q., Saul, L. K.: Distance metric learning for large margin nearest neighbor classification, Journal of Machine Learning Research, 10, 207–244, https://dl.acm.org/doi/10.5555/1577069.1577078, 2009.

Zheng, Y., dos Santos Luciano, A. C., Dong, J., Yuan, W. P.: High-resolution map of sugarcane cultivation in Brazil using a phenology-based method, Earth Syst. Sci. Data., 14, 2065–2080, https://doi.org/10.5194/essd-14-2065-2022, 2022.